# Identifying, characterizing and predicting spatial patterns of lacustrine groundwater discharge

**-** Christina Tecklenburg[1], Theresa Blume[1]

[1]Helmholtz Centre Potsdam, GFZ German Research Centre for Geosciences, Section Hydrology, Potsdam, Germany

*Correspondence to*: Christina Tecklenburg (christina.tecklenburg@gfz-potsdam.de)

**Abstract.** Lacustrine groundwater discharge (LGD) can significantly affect lake water balances and lake water quality. However, quantifying LGD and its spatial patterns is challenging because of the large spatial extent of the aquifer-lake interface and pronounced spatial variability. This is the first experimental study to specifically study these larger scale

patterns with sufficient spatial resolution to systematically investigate how landscape and local characteristics affect the spatial variability in LGD. We measured vertical temperature profiles around a 0.49km² lake in north-eastern Germany with a needle-thermistor, which has the advantage of allowing for rapid (manual) measurements and thus, when used in a survey, high spatial coverage and resolution. Groundwater inflow rates were then estimated using the heat transport equation. These near-shore temperature profiles were complemented with sediment temperature measurements with a fibre-optic cable along

6 transects from shoreline to shoreline and radon measurements of lake water samples to qualitatively identify LGD patterns in the offshore part of the lake. As the hydrogeology of the catchment is sufficiently homogeneous (sandy sediments of a glacial outwash plain, no bedrock control) to avoid patterns being dominated by geological discontinuities, we were able to test the common assumptions that spatial patterns of LGD are mainly controlled by sediment characteristics and the groundwater flow field. We also tested the assumption that topographic gradients can be used as a proxy for gradients of the

groundwater flow field. Thanks to the extensive data set these tests could be carried out in a nested design, considering both small and large-scale variability in LGD. We found that LGD was concentrated in the near shore area, but along-shore variability was high, with specific regions of higher rates and higher spatial variability. Median inflow rates were 44 L m$^{-2}$ d$^{-1}$ with maximum rates in certain locations going up to 169 L m$^{-2}$ d$^{-1}$. Offshore LGD was negligible except for two local hotspots on steep steps in the lake bed topography. Large-scale groundwater inflow patterns were correlated with topography

and the groundwater flow field whereas small-scale patterns correlated with grain size distributions of the lake sediment. These findings confirm results and assumptions of theoretical and modelling studies more systematically than was previously possible with coarser sampling designs. However, we also found that a significant fraction of the variance in LGD could not be explained by these controls alone and that additional processes need to be considered. While regression models using these controls as explanatory variables had limited power to predict LGD rates, the results nevertheless encourage the

use of topographic indices and sediment heterogeneity as an aid for targeted campaigns in future studies of groundwater discharge to lakes.

## 1 Introduction

By linking groundwater with the surface water body, lacustrine groundwater discharge (LGD) can strongly control lake water quality and lake water budgets. Hence, all processes affecting quantity and quality of groundwater could also affect

lake water quantity and quality (Winter et al., 1998; Rosenberry et al., 2015). To understand the vulnerability of groundwater dominated lakes it is not only important to know the total volume of groundwater lake exchange, but also the spatial patterns of LGD (Meinikmann et al., 2013; Lewandowski et al., 2015).

### 1.1 Spatial patterns of lacustrine groundwater discharge and their potential controls

In an isotropic homogenous aquifer, the exchange between groundwater and lake is expected to follow a distinct pattern along a 2D transect: as sloping groundwater water tables meet the flat surface of the lake, groundwater inflow is strongest in close proximity to the shoreline and decreases exponentially with distance to shore (McBride and Pfannkuch, 1975). However, isotropic and homogenous conditions rarely exist and spatial distribution of groundwater inflow differs strongly from lake to lake (Rosenberry et al., 2015). Experimental studies highlighted a large variety of observed exchange patterns including decreasing seepage with distance from shoreline (McBride and Pfannkuch, 1975; Brock et al., 1982; Cherkauer & Nader, 1989; Kishel & Gerla 2002), increasing seepage with distance from shoreline (Cherkauer and Nader, 1989; Schneider et al., 2005; Vainu et al., 2015), local hotspots of offshore seepage (Fleckenstein et al., 2009; Ono et al., 2013) and a high small-scale variability in near shore zones (Kishel and Gerla, 2002; Blume et al., 2013; Neumann et al., 2013; Sebok et al., 2013). Most often complex hydrogeological settings are the reason for deviations from the theoretical pattern of LGD (Rosenberry et al., 2015). For example, it was found that offshore LGD was caused by local connections with a deeper aquifer (Fleckenstein et al., 2009; Ono et al., 2013) or resulted from local thinning of low permeable lake sediment (Cherkauer & Nader 1989).

In general, the position of a lake in its regional groundwater flow system determines if a lake receives groundwater, loses water towards the groundwater or both (Born et al., 1974). As the groundwater flow field is often not well known, landscape topography can help to determine the groundwater flow field in humid regions and homogenous aquifers, where groundwater tables are assumed to follow the topography (Toth, 1963). However, Haitjema and Mitchell-Bruker (2005) found that the groundwater table is only topographically controlled if the ratio of groundwater recharge over hydraulic conductivity is sufficiently large and that often groundwater tables are indeed not topography, but recharge controlled (for a US wide classification of these groundwater table controls see also Gleeson et al. 2011).

Little is known about controls of small-scale variability of LGD. LGD is driven by the hydraulic gradients between lake and aquifer and controlled by the hydraulic conductivity. So far, there is no clear picture about the role of lake sediment characteristics in controlling LGD patterns and observations seem to be very site specific. For example, Kidmose et al. (2013) found that low permeable lacustrine sediments can completely prevent groundwater upwelling, whereas Vainu et al. (2015) observed LGD through low permeable lacustrine sediments. Kishel and Gerla (2002) associated small-scale variabilities in LGD with small-scale heterogeneities in hydraulic conductivities (Kishel & Gerla, 2002), Schneider et al., (2005) found no correlation between seepage rates and sediment characteristics. Methods used in these studies include seepage meters (e.g. Kidmose et al. 2010, 2013, Kishel and Gerla, 2002, Schneider et al. 2005, Vainu et al. 2015), and piezometers (Kishel & Gerla, 2002).

## 1.2 Pattern identification

Although several methods exist to quantify groundwater-lake exchange (Rosenberry et al., 2015), measuring LGD patterns is challenging as we are faced with pronounced spatial variability across the large extent of the aquifer-lake interface. Heat as a natural tracer of groundwater-surface water interactions has received increasing attention in the last decade (Rau 2014).
When groundwater and surface water temperatures differ significantly, sediment surface temperature can be used to localize groundwater inflows in lakes (Blume et al.; 2013; Sebok et al., 2013). Using a fibre optic distributed temperature sensing (FO-DTS) system for this purpose has the advantage of providing precise temperature measurements with a high spatial resolution along fibre optic cables up to a length of several kilometres (Selker et al., 2006). Although sediment surface temperatures do not allow a direct estimation of water exchange fluxes, sediment temperature anomalies can be taken as an
indicator for groundwater inflows (Blume et al., 2013; Sebok et al., 2013). Another method which uses heat as a tracer and which has been used successfully to investigate exchange patterns between lakes and groundwater are vertical temperature profiles (VTP) (Blume et al., 2013; Meinikmann et al., 2013; Neumann et al., 2013; Sebok et al., 2013). In contrast to the FO-DTS method, VTP measurements allow the calculation of exchange rates by using the analytical solution of the heat transport equation (Schmidt et al., 2006). Temperature profiles can be measured manually or continuously using profile
probes. The measurement of radon activity can also help to identify groundwater inflows (Kluge et al., 2012; Ono et al., 2013; Shaw et al., 2013). Elevated radon activities in surface water indicate groundwater inflow as radon is naturally enriched in groundwater but degasses quickly in surface water.

Existing lake studies have investigated LGD patterns with either a high spatial resolution (1–2 m²) but a local focus (10 m × 17 m – 25 m × 6 m) (Kishel & Gerla, 2002; Blume et al.; 2013; Sebok et al., 2013) or focused on the entire lake, but used a
relatively low spatial resolution (measurements along the shoreline: every 200 m – 3000m) (Schneider et al., 2005; Meinikmann et al., 2013; Shaw et al., 2013). However, the experimental effort required rarely allows their extension to cover the lateral, along-shore dimension in sufficient extent and detail to identify the spatial variability and patterns of LGD along the shore-line.

## 1.3 Objectives

Identifying the processes and structures controlling LGD patterns is the key to predicting them reliably (Grimm 2005). The aim of this study is the characterization of inflow patterns as well as the identification of their controls. The ability to identify patterns and their controls strongly depends on the spatial resolution and the extent of the applied experimental methods. By taking measurements with a high spatial resolution over large parts of the lake we are closing the observational gap between high-resolution "plot"-scale studies (focusing on a small shore line segment) and low resolution larger-scale studies (see
section 1.2) and open the possibility to truly investigate not only shore-lake transects or plots, but along-shore spatial variability and patterns of LGD.

The study design aimed at answering the following research questions:

- How variable is LGD in space?
- Can we identify patterns?
- Can we link the patterns to external factors/controls?
- Can we use LGD patterns to test if groundwater tables are topography controlled?
- Can we predict LGD patterns?

To study these research questions we chose Lake Hinnensee, a typical post-glacial lake located in the intensively monitored TERENO observatory in the lowland landscape of northeast Germany. Strong water level declines observed in the last decades at this lake as well as at others in the region are currently under investigation. This lake system has the additional advantage that the upper unconfined aquifer in which the lake rests can be considered as largely homogeneous and isotropic (sandy sediments of a glacial outwash plain, no bedrock control). Therefore LGD patterns are unlikely to be dominated by geological discontinuities, and we were able to test the common assumptions that spatial patterns of LGD are controlled by sediment characteristics and topography as a proxy for gradients of the groundwater flow field.

To identify LGD patterns, we measured VTPs in the near shore area and used FO-DTS measurements and radon sampling in the offshore area. VTPs were used to quantify LGD rates, whereas FO-DTS measurements and radon sampling were used as qualitative tracers to detect the presence or absence of offshore LGD.

## 2. Methods

### 2.1 Study site

Lake Hinnensee is located in northeast Germany in the Müritz National Park (53°19'30.6"N, 13°11'16.2"E) and is one of the focus areas of the TERENO observatory Northeast Germany. The landscape of the Müritz National Park was shaped by the last glaciation and is dominated by lakes. Lake Hinnensee was formed as a glacio-fluvial tunnel valley and is located within the outwash plain. Borelogs of the 16 observation wells installed around the lake show largely homogeneous sandy sediments. The terminal moraine is situated north of the lake (Figure 1). Lake Hinnensee has a mean depth of 7 m with a maximum depth of 14 m and is connected to Lake Fürstenseer See in the south. The two lakes together cover an area of 2.68 km², the size of Lake Hinnensee is 0.49 km². The lake system has no surface water inflow or outflow, apart from a two minor ditches connected with the Lake Fürstenseer See that only become active at very high lake level. Since 2011, when first LGD measurements were conducted at Lake Hinnensee, the ditches were only active for a period of four months (maximum observed inflow: 0.0083 m³ s$^{-1}$, 22 February 2012, maximum observed outflow: 0.0030 m³ s$^{-1}$, 11 May 2012). The connection to Lake Fürstenseer See is not assumed to influence LGD patterns of Lake Hinnensee, as the general flow direction of the groundwater flow field is from north to south with water leaving the lake system at the southern end of Lake Fürstenseer See. The relief of the lake catchment is hilly in the north, with steep slopes down to the lake, and more gentle slopes and lower elevations towards the south (Figure 1). Elevations range between 63. and 115 m a.s.l.. The lake is

surrounded by forest. The mean annual precipitation is 610 mm and the mean annual temperature is 8.1 °C (1901–2005 Neustrelitz, DWD-German Weather Service).

## 2.2 Estimating lacustrine groundwater discharge (LGD)

We applied three different methods to determine LGD patterns: VTPs in the near shore region and FO-DTS and radon in the off shore area. The VTPs allowed us to determine LGD rates, while the other methods were only used as indicators for the presence and absence of LGD. As the main body of the study focusses on the temperature-based methods, the radon methodology and results are described in the appendix.

### 2.2.1 Near-shore LGD derived from vertical temperature profiles (VTPs)

VTPs were used to estimate the spatial variability of LGD rates along the shoreline. Profiles were measured 50 cm away from the shoreline every 10 meters along 2.39 km of the shoreline. The dataset covers 62 % of the total shoreline (Figure 1). The VTPs were measured during five field campaigns in August 2011, June and July 2012, January and July 2013 (Table 1, Figure 1). In July 2013, sediment temperatures were additionally measured in 150 cm distance from the shoreline in order to analyse the trend of LGD with increasing distance to shore. Measurements in August 2011 and January 2013 were conducted only on a 350 m long subsection of the shoreline in the north east of the lake in order to analyse the temporal stability of the observed patterns (Figure 1).

One VTP consisted of six temperature measurements: one at the sediment–water interface and five in the saturated sediment at 5, 10, 20, 30, and 40 cm depth. Temperatures were measured with a high-precision digital thermometer (Greisinger GMH 3750) and a corresponding Pt100 thermistor with an accuracy of ± 0.03 °C. The needle had a length of 45 cm and a diameter of 3 mm.

LGD was calculated from the measured VTP using the analytical solution of the 1-D heat flow equation from Bredehoeft and Papaopulos (1965). Assuming a vertical water flux along the temperature profile and steady state temperatures at the sediment–water interface, sediment temperature at a specific depth is calculated as follows:

$$Tmod(z) = \frac{exp^{\frac{qz*pfcf*z}{kfs}} - 1}{exp^{\frac{qz*pfcf*L}{kfs}} - 1} * (T_L - T_0) + T_0, \tag{1}$$

where $qz$ is the vertical water flux [m s$^{-1}$] (positive for groundwater gaining), $pfcf$ is the volumetric heat capacity of the fluid [J m$^{-3}$ K$^{-1}$], $z$ is the depth below the upper boundary [m], $kfs$ is the thermal conductivity of the sediment [J s$^{-1}$ m$^{-1}$ K$^{-1}$], $L$ is the extent of the exchange zone and the depth of the lower boundary [m], and $T_L$ is the temperature of the lower and $T_0$ of the upper boundary (°C). The values of $pfcf$ of water and $kfs$ of lake sediment were taken from Stonestrom and Constantz

(2003). *Pfcf* of water was set to $4.19 \times 10^6$ J m$^{-3}$ K$^{-1}$ and *kfs* to 2 J s$^{-1}$ m$^{-1}$ K$^{-1}$, a typical value for sandy sediment, which was the dominant grain size in the upper meter of lake sediment.

Usually the upper boundary is the sediment-water interface (Schmidt et al. 2006; Blume et al., 2013; Meinikmann et al., 2013). But at locations with shallow water depths in lakes, temperatures in the near-surface sediments can be strongly affected by daily temperature variations and thus violate the upper boundary condition of the steady state model. To avoid unreliable LGD calculations due to biased temperatures at the upper boundary, we instead used the temperatures measured at 10 cm sediment depth. A shift of the upper boundary from the sediment-water interface to a depth of 10 cm had a negligible effect on the estimation of the LGD rate assuming steady state conditions. This was validated with theoretical temperature profiles. A shift of the boundary condition to a depth of 10 cm caused a maximal deviation in the estimation of exchange rates of 1 L m$^{-2}$ d$^{-1}$ and the error decreased with increasing flowrates.

For the lower boundary, we used the shallow groundwater temperature measured in close vicinity of the lake (Figure 1c). For the length of the exchange zone, we tested different values. The quality of LGD estimation increased with increasing L, but was insensitive for values larger than 3 m. Thus, L was set to 3 m.

The exchange rate was optimized by minimizing the root mean square error (RMSE) between measured and calculated sediment temperatures as described in Schmidt et al. (2006):

$$RMSE = \sqrt{\frac{1}{n}\Sigma\left(T_{meas(z)} - T_{mod(z)}\right)^2}, \tag{2}$$

The quality of the fit between measured and modelled sediment temperature was also visually checked using plots of the measured and modelled VTPs. Fits with RMSE greater than or equal to 0.4 °C were not used for further analyses as differences between modelled and measured data were considered too large.

Estimated LGD values were analysed for their lateral spatial variability using VTPs measured at a distance of 50 cm from shoreline, for their trend with increasing distance from shore using VTPs measured at 50 cm and 150 cm distance from shore and for their temporal stability using VTPs measured in the different years (Table 1).

Spatial variability and correlation of LGD along different distances along the shoreline were analysed using autocorrelation plots and autocorrelation values ($|\rho|$) as described in Caruso et al. (2016). High autocorrelation indicates that LGDs along a given stretch of the shoreline are correlated (i.e. if LGD is high in a certain location it is also likely to be high at 10m distance), whereas $|\rho| < 0.2$, indicate that LGDs are uncorrelated and strong spatial variability exists.

In order to analyse the temporal stability of spatial patterns we used the differences between the LGD rates measured at different points in time and calculated the correlation between the VTP surveys using the Spearman's rank correlation coefficient ($\rho$). Correlations were regarded as significant for p-values smaller than 0.05

To test if single sediment temperature measurements instead of profiles could be used as a quickly measureable qualitative indication for LGD spatial patterns, we determined the correlations between sediment temperatures at all individual depths with LGD rates determined from the full profiles.

### 2.2.2 Lake sediment temperature anomalies as indicators for offshore-LGD based on fibre optic distributed temperature sensing (FO-DTS)

To identify offshore groundwater inflow patterns, we measured sediment surface temperatures with a 500 m long FO-DTS cable installed permanently along 6 transects through the northern part of the lake (Figure 1c). Two divers ensured good contact of the cable to the lake sediment and also tracked the location of the cable with a differential GPS system (Topcon GR-3) installed on a buoy.

The technology of the FO-DTS is based on the detection of the Raman scattering of a laser pulse through the optical fibre (Ukil et al. 2012). For our measurements, we used a Sensornet Halo device with a sampling resolution of 2 m and a measurement precision of 0.05 °C.

We carried out two measurement campaigns, in February and in August 2014 (Table 2). In February the DTS measurements were taken between 18:49 and 19:17 CEST with a temporal resolution of 2 minutes. We used a single ended set-up with a double ended mode (four channels, two in each direction) and two calibration baths, a warm bath (25.5°C) at one end and a cold bath (0°C) at the other end.

In the second campaign from the 27–28 August 2014 measurements extended over 24 hours, from 18:43 on the 27[th] until 18:45 CEST the next day with a temporal resolution of 2 min. The setup was the same as in February, but additionally both cable ends were run through the cold bath (warm bath: 20.9°C, cold bath 0.1°C).

The trend and offset in the DTS temperature data were corrected using external temperature loggers in the calibration baths (February: Greisinger GMH 3750 (accuracy: ± 0.03 °C); August: HOBO TidbiT v2 Water Temperature Data Logger (accuracy: ± 0.21 °C)).

All four channels showed the same pattern with only small differences in absolute temperature values and further analyses were based on one of the four traces. Sediment temperatures in August were strongly affected by solar radiation. Analysis of 24 hour amplitude or daily minimum temperature did not provide useful information of groundwater inflow patterns as the impact of solar radiation was too strong and spatially variable. Sebok et al. (2013) recommended using night time data to avoid the uncertainties caused by solar radiation. However, at night shallow near shore water cooled down and it was not easy to distinguish if temperature shifts resulted from groundwater inflow or resulted from a decrease of water temperature due to decreasing air temperature. We thus chose a time window in which the temperature at the near shore shallow region and in the deeper region of the lake were very similar and groundwater inflow induced temperature shifts were easy to identify. This time window was from 18:43 to 19:11 CEST on August 27[th].

Temperature depth profiles of the lake were available with 1 m spatial resolution (HOBO Water Temperature Pro v2 Data Logger, accuracy: $\pm 0.21$ °C). In winter we had only one profile in the central part of the lake, but in August a second profile further north was available (Figure 1a, b). The groundwater temperature was measured in a piezometer (OTT Orpheus Mini, accuracy: $\pm 0.5$ °C) close to the lake (Figure 1c) and air temperature data were available from a weather station 1.5 km away and in August an additional air temperature data logger (of the same type as used for the water profiles) was installed directly at the lake.

## 2.3 Identifying controls of LGD patterns

In order to identify the controls of the observed LGD patterns, we characterized both the near-and the far-field conditions and correlated these characteristics with LGD rates using Spearman's rank correlation coefficient. At the local scale (near-field conditions) this includes sediment characteristics, while at the larger scale (far-field conditions) we considered topographic indices and the groundwater flow field as the most likely controls.

### 2.3.1 Sediment heterogeneity as a small-scale control on LGD patterns

Hydraulic conductivity from slug tests

At 37 VTP positions (Figure 1b), slug tests were performed to estimate hydraulic conductivity ($k_{sat}$) of the near-surface sediment. Slugtests were carried out in piezometers with an inner diameter of 36.4 mm. The screen placed on the lower end of the piezometer had a length of 10 cm and consisted of 4 mm diameter perforations in the PVC tube wrapped with fine mesh. The midpoint of the screen was 50 cm below the sediment surface. To minimize interference with the temperature profile measurements, the piezometers were installed at 50 cm distance. For the rising-head tests water was quickly removed out of the piezometer using a peristaltic pump. Recovery of the water table was measured with automatic pressure logger (HOBO 13-Foot Fresh Water Level Data Logger, accuracy: $\pm 0.3$ cm) with a temporal resolution of 1 second. Recovery data were then analysed using the approach of Hvorslev (1951):

$$k_{sat} = \frac{\pi r^2}{T_0 c}, \tag{3}$$

where $r$ is the radius of the piezometer, $T_0$ the time needed to recover 37 % of initial water level and $c$ a shape factor. The shape factor depends on the ratio of screen length and radius. The piezometer had a screen length radius ratio of 5.5 and thus we used a shape factor introduced by Chapuis (1989), valid for wells with ratio smaller than 8:

$$c = 4\pi r \sqrt{\frac{L}{2r} + \frac{1}{4}} \ , \tag{4}$$

where $L$ is the length of the screen.

Grain size distributions from sediment cores

Sediment cores were taken from 30 selected slug test positions (Figure 1b). Sediment cores were taken with a transparent tube with an inner diameter of 32 mm. Length of cores varied between 42 cm and 145cm, with the majority of core lengths between 80 cm and 128 cm. Each core was split into samples according to the visible sediment layers. The samples were oven-dried at a temperature of 105 °C and sieved with a vibratory sieve shaker (Retsch AS 200). The sieving setup included the following mesh sizes: 0.063, 0.125, 0.2, 0.3, 0.5, 0.63, 2, 5, 10 mm. Grain sizes smaller than 0.063 mm were classified as silt, grain sizes larger than 0.063 mm but smaller than 0.2 mm as fine sand, larger than 0.2 mm but smaller than 0.63 mm as medium sand, larger than 0.63 mm but smaller than 2 mm as coarse sand larger than 2 mm as gravel.

For the correlation analyses between sediment characteristics and LGD, we used only samples taken from the upper 100 cm of the lake sediment. The mean of each grain size fraction was calculated for each sampling location from all single samples of the upper 100 cm in which the core was split. In addition to the correlation analyses, simple and multiple linear regression models were calculated between LGD and each grain size fraction. For the calculation of the multiple linear regression models, correlations between explanatory variables were checked before and variables were regarded to be independent from each other if $\rho$ was below 0.7. Models were regarded as significant if p-values were below 0.05. The goodness-of-fit of the models was estimated with the coefficient of determination ($R^2$).

## 2.3.2 Topographic indices as controls on large-scale LGD patterns

In order to analyse the effect of far-field conditions on LGD patterns the following topographic indices were calculated using SAGA GIS: average elevation, average slope and the percentage of area with low topographic gradient in direct vicinity to the lake shore. To determine the topographic indices we used a digital elevation model (DEM) of the area with a resolution of 1 m. The topographic indices were estimated for representative areas, i.e. upslope areas for shoreline sections of 100 m length. Therefore the shoreline was split into 46 subsections of 100 m length with an overlap of 50 m. As upslope areas can only be determined for points, not for lines, we calculated upslope areas every meter along the shoreline and aggregated them to one upslope area for each subsection. The upslope areas were determined using the multiple flow direction approach. To investigate the topographical zone of influence (zi), four different extents of the upslope areas were considered: 25, 50,100 and 200 m distance from shoreline. These zones of influence will in the following be called: $zi_{25m}$, $zi_{50m}$, $zi_{100m}$, $zi_{200m}$. The indices slope and elevation were averaged over each upslope area (arithmetic mean). The percentage of area with low topographic gradient was here defined as the percentage of the upslope area not to be higher than 50 cm above lake level in direct vicinity to the lake shore. This threshold was chosen as this was the area flooded at maximum lake levels known within the last 25 years. Indices were correlated with median LGD rates for the 100 m long subsections derived from VTPs using the Spearman's rank correlation coefficient. Each subsection included 10 VTP measurement locations. Furthermore simple and multiple linear regression models were calculated between LGDs and topographic indices derived for $zi_{25m}$ and $zi_{50m}$, as in these zones correlation between LGD and far-field conditions were strongest. Correlations between explanatory variables were checked and regarded as independent from each other if $\rho$ was below 0.7. Predictors were regarded as

significant if p-values were below 0.05. The goodness-of-fit of the models was estimated based on the coefficient of determination R².

### 2.3.3 Groundwater flow field as control on large-scale LGD patterns

The groundwater flow field was the second far-field variable assumed to affect the LGD patterns. The groundwater flow field is generally assumed to be largely controlled by topography. We used two approaches to test this assumption: the water table ratio (Haitjema and Mitchell-Bruker 2005) and a comparative analysis of flow fields determined based on measured groundwater levels alone (ordinary kriging) or including topographic effects (regression kriging). The simple dimensionless water-table ratio: $WTR= (RL^2)/(mkHd)$ with R as annual recharge rate [m/d], L as mean distance between surface waters [m], m=8 or 16 [-] for either 1D or radial flow, k as average hydraulic conductivity [m/d], H as average aquifer thickness [m] and d being the maximum terrain rise [m] (Haitjema and Mitchell-Bruker, 2005, Gleeson et al. 2011) allows a first test of the potential influence of topography on the groundwater flow field, with WTR>1 indicating topography controlled water tables and WTR<1 indicating recharge controlled water tables. The average hydraulic conductivity was determined from 92 undisturbed cores taken during observation well installation and were measured in the lab using a permeameter.

In order to derive the groundwater flow field, measured groundwater heads from 16 observation wells located around Lake Hinnensee (Figure 1a) were spatially interpolated. 12 of the 16 bore wells were drilled in 2012, three in 2014 and one existed already before installation of the TERENO monitoring network. Groundwater levels were measured every seven to nine weeks since 2012 using an electric contact meter (SEBA Hydrometrie, electric contact meter type KLL, accuracy: ± 1 cm). To determine the groundwater flow field we used groundwater levels measured in 2014, when all wells were completed. In 2014 groundwater levels were generally lower than during the VTP measurement campaigns (2011-2013), but the spatial patterns of groundwater heads of the 12 wells already installed in 2012 remained stable. Groundwater levels from March 2014 had the smallest differences (mean difference 5.55 cm) to available groundwater data around the time of the VTP measurement campaigns and were thus chosen to derive the groundwater flow field. For the interpolation of the groundwater measurements, we used both ordinary kriging and regression kriging. In regression kriging a linear regression between an external variable and the depending variable is included. This allowed us to incorporate the potential effect of topography on the groundwater flow field. In order to minimize the effect of small-scale heterogeneities in the topography the DEM was smoothed by reducing the resolution from 1 m to 10 m and rescaling to a resolution of 1 m to maintain a consistent resolution of the results. The groundwater gradients were then calculated from the interpolated groundwater surface.

To analyse the correlation of the groundwater flow field with the LGD patterns, we averaged groundwater gradients in each of the subcatchments for each zone of influence (arithmetic mean) and correlated these with the median LGD rates of the subsections using the Spearman's rank correlation coefficient as described in *2.3.2*. In addition to the correlation analyses, simple linear regression models were calculated between LGD and groundwater gradients. Furthermore groundwater gradients were also included in multiple linear regression models with topographic indices.

All analyses were carried out in the statistic software R (R Development Core Team, 2011). For the geographical analyses we used the geographical information system SAGA GIS, and the package "rsaga" (Brenning, 2008).

## 3. Results

### 3.1 Estimating lacustrine groundwater discharge (LGD)

**3.1.1 Near-shore LGD derived from vertical temperature profiles (VTP)**

At 216 locations along the shoreline of Lake Hinnensee (Figure 1) a total of 520 VTPs were measured to analyse a) spatial patterns of near shore LGD, b) the trend of LGD with increasing distance from shore and c) the temporal stability of LGD patterns. These 520 profiles thus include repeated measurements in time as well as measurements at two distances to shore. At the western lake section, 150 m to 290 m from the northern tip, VTP measurements could only be taken every 20 m

instead of every 10 m as the lake shore could either not be accessed or the sediment was unsuitable for measuring due to a thick layer of muddy organic material. However, as lake sediments in this lake section were quite homogeneous, we assume that despite the wider spacing we still captured the spatial variability of LGD. The same reasons also precluded measurements at 11 other locations around the lake. These other 11 locations were irregularly distributed so that gaps were small and we do not expect a strong influence of these gaps on overall spatial patterns. 22 profiles (4%) were excluded from

the analyses as no satisfying fit to the heat transport equation could be achieved. The quality of all remaining LGD estimations was satisfying (median(RMSE) = 0.06 °C, n = 498).

Spatial patterns along the shore line

LGD rates determined from VTPs every 10 m along 2.39 km of the shoreline (216 locations) ranged from -12 L m$^{-2}$ d$^{-1}$ (losses) to 169 L m$^{-2}$ d$^{-1}$ (gains) with a median of 44 L m$^{-2}$ d$^{-1}$ and an interquartile range (IQR) of 26 L m$^{-2}$ d$^{-1}$. Occurrence of

very strong LGD rates of more than 94 L m$^{-2}$ d$^{-1}$ (positive outliners of LGD distribution), was limited to the northern most 140 m on both the western and the eastern shore of the lake (between "a" and "b" and "f" and "g" in Figure 2) and to one single spot at the western shore 470 m to the south ("i" in Figure 1&2). The northern most 140 m on both the west and the east shore (between "a" and "b" and "f" and "g" in Figure 1&2) are in the following called "the northern part" and the adjacent region in the south (between "b" and "e" and "g" and "j" in Figure 1&2) will be called "the southern part". Negative

rates were only observed at the eastern shore, between 480 m and 530 m from the northern tip ("c" in Figure 1&2). In the northern part of the lake, LGD was stronger and spatially more variable (median = 74 L m$^{-2}$ d$^{-1}$) than in the southern part (median = 41 L m$^{-2}$ d$^{-1}$) (Figure 2, Figure 3). In the northern part LGD was statistically uncorrelated for all lag distances, while in the southern part it was auto correlated up to a lag distance of 50 m with |ρ| ranging between 0.62 and 0.23 (Figure 3). Autocorrelation in the southern part was stronger on the eastern than on the western shore.

Spatial patterns perpendicular to the shore line

Between 660 m and 1520 m along the eastern shore and 300 m and 830 m south of the northern tip along the western shore, VTPs were measured at 50 cm and 150 cm distance from shoreline to analyse the trend of LGD with increasing distance from shore.

In more than two thirds of all cases (71 %), LGD measured at 150 cm from the shoreline was lower than the rate measured at 50 cm distance (Figure 2). The reduction of LGD rate was on average 20 % (median). However, in 29 % of all cases, LGD increased with distance to the shore (Figure 2) with an average increase of 15 % (median). The patterns of LGD along the shore line measured 50 cm and 150 cm apart from shore were very similar ($\rho = 0.81$ (p-value $< 2\times10^{-16}$), Figure 2).

Temporal stability of spatial patterns

The annual repetitions of LGD rates measured at 43 VTP positions (Figure 1) highlight that the observed LGD patterns were correlated between the individual measurement campaigns (Figure 4). The correlation coefficient was 0.71 (p-value $= 5\times10^{-6}$) between summer 2011 and 2012 (n = 34), 0.82 (p-value $= 10^{-3}$) between 2012 and summer 2013 (n = 13), 0.70 (p-value $< 4\times10^{-6}$) between 2012 and winter 2013 (n = 34), and 0.66 (p-value $< 3\times10^{-5}$) between 2011 and winter 2013 (n = 33). The differences between LGD rates measured in different years were lowest comparing rates from summer 2011 and summer 2012 (median = -6 L m$^{-2}$ d$^{-1}$) and strongest comparing rates from summer 2011 and winter 2013 (median = 27 L m$^{-2}$ d$^{-1}$).

Single sediment temperature measurements as a qualitative indicator for LGD spatial patterns

Sediment temperatures from the top of the sediment down to a depth of 10 cm were not well correlated with LGD rates, but strong correlations were found between LGD rates and sediment temperatures measured 20 cm below surface and deeper (correlation coefficients range between 0.46 and 0.96). While the correlation is generally high, the slope of the regression line varies in time (Figure 5).

### 3.1.2 Lake sediment temperature anomalies as indicators for LGD based on fibre optic distributed temperature sensing (FO-DTS)

We measured lake sediment temperature with a FO-DTS cable installed in 6 transects through the northern part of the lake (Figure 1c) during two measurement campaigns (20.02.2014, 27.08.2014) to identify offshore groundwater inflow. The complementary results of the radon measurement campaigns are described in the appendix.

The winter and summer measurements consisted of 15 measurements in 2-minute intervals. The repetitions resulted in very similar measurements (median range of temperature differences among repetitions in winter: 0.19 °C; maximum range in winter: 0.44 °C, median range in summer: 0.22 °C; maximum range in summer: 0.38 °C) (Figure 6a). In winter sediment temperature ranged between 3.4 °C and 5.3 °C and in summer between 17.0 °C and 18.4 °C (Figure 6). In summer, groundwater was 7 °C cooler than lake water and in winter 3 °C warmer than the lake. The air temperature, groundwater temperatures and temperature depth profiles of the lake measured during the DTS measurements are presented in Table 2.

The spatial patterns of sediment temperature anomalies, i.e. the shifts of sediment temperatures towards groundwater temperatures, were similar in both campaigns. Strongest deviations from sediment temperatures towards the groundwater temperatures (positive in winter and negative in summer) were located near the shoreline at corners two and three (Figure 6). A slight shift towards groundwater temperatures was also observed near corner 5, but only along the DTS cable north of the

corner. These hot and cold spots in winter and summer respectively were not located nearest to the shoreline, but between 2 m and 14 m offshore, where lake depth steeply increased (Figure 6b).

## 3.2 Identifying controls of LGD patterns

### 3.2.1 Sediment heterogeneity as a small-scale control

Hydraulic conductivity from slug tests

The $k_{sat}$ values estimated from the slugtests ranged between $2.03 \times 10^{-6}$ m s$^{-1}$ and $4.25 \times 10^{-5}$ m s$^{-1}$ with an IQR of $1.41 \times 10^{-5}$ m s$^{-1}$ (Figure 7). Points with $k_{sat}$ values lower than the 25 % quartile ($9.20 \times 10^{-6}$ m s$^{-1}$) were mainly located at the western shoreline, while points with values higher than the 75 % quartile ($2.33 \times 10^{-5}$ m s$^{-1}$) were located on the eastern shore. Instead of a positive correlation between $k_{sat}$ values and LGD rates, there was a slight negative, but statistically insignificant (p-value = 0.05), correlation of -0.36 (Figure 7).

Grain size distributions from sediment cores

The sediment samples taken from 30 VTP measurement locations were dominated by sand with a small fraction of gravel and silt. The median percentages of sand, gravel and silt were 92.3 %, 6.8 % and 0.6 %, respectively. Within the sand fraction, medium sand dominated (median = 59.2 %), followed by fine sand (median = 23.3 %) and coarse sand (median =

13.7 %). No consistent layering or trends of grain sizes with depth could be identified. Only the fraction of medium sand decreased slightly with increasing sampling depth by 2 % every 10 cm, but also here the correlation was weak ($\rho = 0.3$, p-value = $6 \times 10^{-4}$).

Relating grain size distributions averaged over the upper meter of the lake sediment to the strength of LGD showed that low LGD rates occurred at locations dominated by fine sand and stronger LGD rates occurred at locations with higher fraction of

larger grain sizes (Figure 8a). LGD was positively correlated with the percentage of gravel and coarse sand and negatively correlated with the percentage of fine sand (Table 3, Figure 8 b-c), but LGD rates were uncorrelated with the grain size fractions medium sand and silt (Table 3). LGD varied by a factor of three across these grain size fractions. For the multiple regression model considering all grain sizesonly coarse sand and fine sand were significant variables (p-values < 0.05). The model had an R² of 0.54 (Table 3). The absolute residuals were on average 21 L m$^{-2}$ d$^{-1}$, with largest positive residuals

(observed < calculated) at a distance of 50 m and 10 m from the northern tip at the eastern and western shore, respectively and largest positive residuals (observed > calculated) at a distance of 70 m and 90 m from the northern tip at the eastern shore (Figure 9a).

### 3.2.2 Topographic indices as controls on large-scale LGD patterns

Subcatchments derived from the 46 shoreline sections differed significantly in size. While subcatchments in the flatter areas of the south were larger, elevations were higher and slopes generally steeper in the north. There was no clear correlation between LGD rates and the size of the subcatchment in each topographical zone of influence (Table 4). Percentages of area with low topographic gradient in direct vicinity to the lake shore area were also not correlated with LGD rates, except for $zi_{25m}$, where a weak negative correlation was found (Table 4). The correlation between LGD rates and the indices elevation and slope were both positive (Table 4, Figure 10 right) and the strength of correlation was strongest for $zi_{50m}$ and decreased with increasing zone of influence (Table 4).

### 3.2.3 Groundwater flow field as control on large-scale LGD patterns

The water table ratio (Haitjema and Mitchell-Bruker, 2005) as an indicator for either recharge or topography controlled groundwater tables was determined based on conservative estimates of the input variables: annual recharge rate R=0.00351 m/d (based on values from Müller et al. (2009) determined in the same region), mean distance to the next surface waters L being approximately 2000 m, m=8 [-] for 1D flow (however, changing this to 16 for radial flow does not change the outcome), the average hydraulic conductivity k=7.776 m/d (based on laboratory analyses of undisturbed cores obtained during observation well installation), average aquifer thickness H=15 m (from bore-logs) and the maximum terrain rise d=52 m. The water table ratio in this case amounts to 0.029 which is << 1 and thus indicates that water tables in the study area are not topography- but instead recharge controlled.

The interpolated groundwater table based on the water tables in the observation wells showed groundwater flow towards Lake Hinnensee from all directions (Figure 10). In general, groundwater gradients were stronger in the north than in the south.

The maximum deviation between interpolated and measured groundwater levels was below 1 cm for both interpolations, but in comparison to the ordinary kriging, the regression kriging (which included topographical information) resulted in significantly stronger gradients: median groundwater gradient in the $zi_{200m}$, for example, was 0.24 cm m$^{-1}$ using regression kriging and 0.09 cm m$^{-1}$ using ordinary kriging. While the correlation between LGD and groundwater gradients derived from ordinary kriging was weak with correlation coefficients ranging between 0.28 and 0.37 (Table 2), stronger correlation was found for LGD and gradients derived from regression kriging (between 0.55 and 0.64) (Table 2). The linear predictor function used for the regression kriging was $gw = 62.5 + 0.02e$ where $gw$ is the groundwater level in meter and $e$ the smoothed surface elevation in meter. The topography was a significant model predictor with a p-value of 10$^{-5}$.

### 3.2.4 Linear regression models between LGD and far-field predictors

The linear regression models describing the correlation between LGD and far-field conditions were estimated for all predictors of $zi_{25m}$ and $zi_{50m}$. Significant predictors of LGD large-scale patterns were elevation, slope, percentage of area with low topographic gradient (only in $zi_{25m}$) and groundwater gradients (Table 5). No significant linear regression model was found with the potential predictors size of subcatchment and percentage of area with low topographic gradient in $zi_{50m}$. The R² of all models were not larger than 0.37 (Table 5), but were below 0.12 for the predictors "percentage of area with low topographic gradient" and "groundwater gradients derived from ordinary kriging" (Table 5). Therefore these predictors were excluded for further analyses.

The calculation of multiple regression models revealed no significant relations between LDG and topographic indices in both zones of influence. Reducing stepwise the most insignificant predictor until all predictors became significant resulted in the single linear regression models as presented above and in Table 5.

Calculated large-scale LGD patterns along the shoreline using the best linear regression model (based on groundwater gradients derived from regression kriging in $zi_{25m}$, Table 5), are shown in Figure 9b. The general spatial pattern was captured by the model, but absolute deviations were on average 10.4 L m$^{-2}$ d$^{-1}$. Strongest overestimations occurred at the distances 500 m at the eastern and 300 m at the western shore and strongest underestimation by the model were found at the distances 450 m and 150 m at the western shore.

## 4. Discussion

### 4.1 LGD patterns along the shoreline and potential controls

The here employed experimental design based on extensive field campaigns provided exceptionally detailed information on both small-scale variabilities and large-scale patterns of LGD rates along a lake shore line. This data set thus bridges the gap between the detailed local and low resolution larger scale investigations of previous LGD studies.

While the employed method of measuring VTPs with a needle-thermistor is sufficiently rapid to make this large high resolution data set possible it nevertheless takes considerable effort. We therefore investigated if measuring temperature at just a single depth instead of measuring an entire depth profile (thus reducing measurement time even further) would already supply at least qualitative information on LGD patterns. We found that temperatures measured at depths larger than 20cm generally correlated well with LGD rates, thus reproducing the general pattern of groundwater inflow. However, to convert these temperatures to LGD rates it would be necessary to measure at least some complete profiles for a large enough range of LGD rates to obtain a decent calibration. As sediment temperatures and their gradients change over time this calibration needs to be repeated at each survey date.

### 4.1.1 Large-scale patterns

As expected, the observed large-scale patterns of LGD at Lake Hinnensee correlated with mean groundwater gradients. Interestingly, even though the system classifies as recharge controlled and not topography controlled according to the water table ratio (Haitjema and Mitchell-Bruker, 2005, Gleeson et al. 2011), correlation was stronger between LGD and groundwater gradients derived from regression kriging (which includes topographic information) than between LGD and groundwater gradients derived from ordinary kriging. This suggests that the groundwater surface was more realistically interpolated using regression kriging and indicates at least some influence of topography on groundwater movement, which is consistent with the theory of topography-controlled groundwater flow described by Toth (1963), The predictors based on surface slopes and surface elevation, so purely topographical information, also correlated with LGD rates (with a correlation coefficient of 0.6), another indication of at least some topographic control on the groundwater flow field. However, even the best regression model (based on the groundwater table gradients from regression kriging) did not capture all of the observed variability in LGD, thus suggesting the existence of additional controls. In contrast to observations of streamflow generation in mountain catchments (Jencso et al., 2009, 2010), no positive correlation was found between the size of the subcatchment and LGD rates (Table 4) indicating that surface catchments derived for lake subsections were not a good estimator for the amount of subsurface water flowing into the corresponding lake sections. We assume our result indicates that subsurface catchments differ from surface catchments which is not unusual in low land areas. The weak negative correlation between LGD and the topographic index "percentage of area with low topographic gradient in direct vicinity to the lake shore" in $zi_{25m}$ indicated that low topographic gradients at the shoreline could buffer groundwater flow towards the lake.

Even though the interpolated groundwater surface showed groundwater flow towards Lake Hinnensee from all directions (Figure 10), we measured negative LGD rates at one small subsection of the lake (Figure 2). Reasons for this flow reversal are unclear. However, the neighbouring stretches of shoreline were characterized by very low LGD rates (Figure 2), even though $k_{sat}$ values at this section were comparably high (Figure 7) and thus we assume that very low hydraulic gradients are the cause for the low LGD rates. While transpiration is likely to cause diurnal fluctuations in groundwater levels all around the lake, it can result in a temporary local inversion of the groundwater – lake gradients at locations where these gradients are very low (Winter et al., 1998). This could be a potential explanation for the negative LGD rates measured at this location.

### 4.1.2 Small-scale patterns

Our measurements revealed strong small-scale spatial variability in LGD along the shoreline (Figure 2). Absolute amounts and spatial variability of LGD were within the range of previous studies (Rosenberry et al., 2015; Blume et al.; 2013; Neumann et al., 2013). Measuring VTPs with a high measurement resolution along large parts of the shoreline and analysing LGD values with autocorrelation plots highlighted furthermore that the strength of small-scale variability also varies along the shoreline. This type of information is likely to be overlooked in studies focusing either on entire lake systems but using a low spatial resolution (Schneider et al., 2005; Meinikmann et al., 2013; Shaw et al., 2013) or using a high spatial resolution

but only on a very local scale (Kishel & Gerla, 2002; Blume et al.; 2013; Sebok et al., 2013). The repetitions of VTP measurements revealed that the observed patterns were stable in time and are thus likely controlled by static characteristics. Differences in LGD rates measured in different years are likely the result of annual differences in groundwater recharge and thus gradients of the flow field.

5 Surprisingly no positive correlation was found between LGD rates and $k_{sat}$ values derived from slugtest at Lake Hinnensee. The relationship between $k_{sat}$ and LGD could be confounded as a result of strong differences in hydraulic gradients. However, even when only adjacent measurement locations with similar gradients were taken into account no clear positive correlations appeared (Figure 7). Slug test were found to be the most accurate method to determine $k_{sat}$ values of sandy stream beds (Landon et al., 2001), but estimation of hydraulic conductivity is always subject to high uncertainties (Landon et 10 al., 2001; Kalbus et al., 2006). Even though the slug tests were carried out carefully, we cannot exclude that pore structure was altered during the piezometer installation and thus the $k_{sat}$ values of lake sediments were changed. In contrast to $k_{sat}$ values, LGD rates clearly correlated with both the finest and the coarsest grain size fractions. Grain sizes give no direct information on hydraulic conductivity, but coarse sand and gravel is associated with higher hydraulic conductivity values than well sorted fine sand (Bear, 1972). As LGD rates correlated positively with percentages of gravel and coarse sand and 15 negatively with the percentage of fine sand, this corroborates the assumption that sediment heterogeneities do at least partially control small scale variability in LGD. We furthermore see that a variation of LGD of up to a factor of three can be due to grain size variability alone.

### 4.2 LGD patterns with increasing distance to shore and their potential controls

To identify LGD patterns with increasing distance to shore, we analysed VTP profiles measured in 50 cm and 150 cm 20 distance from the shoreline in the southern part. Results showed a prevailing decrease of LGD with distance to shore (median decrease of 20%). This observation corresponds to the theoretical pattern of LGD found by McBride and Pfannkuch (1975) and other experimental studies (Brock et al., 1982; Cherkauer & Nader 1989, Kishel & Gerla 2002; Blume et al.; 2013). The study from Blume et al. (2013), conducted at small shore line section of 20 m length and 4 m width in the northern part of Lake Hinnensee, indicated that the strongest decrease of LGD occurred in the first 1.5 m distance from 25 shore. However, LGD increased with increasing distance from shore in 29 % of the locations. The locations with anomalously increasing LGDs did not show any obvious anomalies with respect to local bathymetry, density of vegetation or organic top-layers, which could have been used to explain the observed patterns. The hypothesis that differences in sediment characteristics cause these anomalies could unfortunately not be tested as no information on sediment characteristics is available for distances from shore larger than 50cm.

### 30 4.3 Offshore-LGD patterns and potential controls

We investigated the presence and absence of offshore LGD with two natural tracers: radon and heat, and the two methods lead to similar results (for the radon results and discussion see appendix). The FO-DTS measurements showed no shifts in

sediment temperatures towards the groundwater temperature in the flatter and deeper parts of the lake (Figure 6b), and we assume that groundwater inflow is insignificant here. Low radon activities measured offshore at the lake bottom across the entire lake also support this assumption (see appendix). Insignificant groundwater inflow in the flatter and deeper part of the lake is in correspondence with the theory of exponentially decreasing groundwater inflow with distance from the shore

introduced by McBride & Pfannkuch (1975). Furthermore we know from observations by divers that the lake bottom was covered with fine-grained organic sediments, which typically accumulates in the deep, flat parts, where wave action cannot re-suspend the fine sediments (Rosenberry et al., 2015). A layer of fine-grained sediment could significantly decrease hydraulic conductivity of the lake bed sediment and can even totally prevent groundwater lake exchange (Kidmose et al., 2013).

FO-DTS showed local hotspots of groundwater inflow at three locations, all on steep steps between the near shore part and the flatter central basin. These locations with sediment temperature anomalies coincided with high near shore LGD rates (estimated with VTPs), while corners where no temperature anomalies were found, coincided with low near shore LGD rates (Figure 6b). As hotspots in near shore LGD occurred locally and mainly in the northern part of the lake, we assume that the occurrence of hotspots of LGD at steep steps is also a local phenomenon limited to the northern part. As the occurrence of

the local sediment temperature anomalies is temporally stable (Figure 6a), we assume that static characteristics are responsible – at this locations, where near shore LGD was already strong, the morphology of steep steps might force a local offshore increase of LGD: at steep slopes fine sediment is prone to be re-disturbed by turbidity current activities (Håkanson, 1977), locally increasing hydraulic conductivity and thus also LGD.

## 4.4 Prediction of LGD patterns

Using linear regression models based on topographic characteristics or the groundwater flow field to predict large-scale patterns of LGD at Lake Hinnensee roughly reproduced the observed patterns, but locally strongly over- or underestimated the observed LGD rates (Figure 9b). However, regression models considering sediment heterogeneities were able to explain more than 50 % of the observed small-scale variability in LGD ($R^2 = 0.55$). We calculated linear regression models separately for topographic indices and sediment heterogeneities, because sediment cores were only taken from a fraction of

the lake covering not more than one lake subsection used for far-field analysis. But as LGD is driven by both, the hydraulic gradients between lake and aquifer and by the hydraulic conductivity, combining the information would likely explain more of the observed variability. Another possible influence are currently unknown local heterogeneous features within the adjoining aquifer (Winter 1999; Cherkauer & Nader, 1989; Fleckenstein et al., 2009). Such small-scale structures may influence the groundwater flow paths and cause variability in LGD.

## 5. Summary and Conclusion

As LGD can significantly contribute to a lake water budgets and could furthermore significantly influence lake water quality by transporting large loads of nutrients or contaminants, quantifying LGD rates and determining LGD patterns can be essential for a sustainable lake management (Meinikmann et al., 2013; Lewandowski et al., 2015). While LGD is known to be spatially variable, it is also not easily measured, especially with an extent and spatial resolution that allows for the characterisation of LGD patterns. Furthermore, causes and controls of these patterns are not well understood. Our aim was the characterization of LGD patterns at Lake Hinnensee based on a unique high resolution data set that extends along most of the shoreline, and to use this data set to test common assumptions. Identifying external (and easily measureable) controls as a means for pattern prediction would greatly reduce experimental effort. By using VTPs in the near shore area and FO-DTS measurements and radon sampling in the offshore area, we identified the following pattern in LGD for Lake Hinnensee: LGD was concentrated in the near shore area and generally decreased with distance to shore; some local hotspots of LGD were identified in locations of steep steps towards the lake bottom; overall, offshore-LGD was insignificant. LGD was generally stronger and more variable in the northern part than in the southern part of the lake. Repetitions of LGD measurements indicated that the observed patterns in LGD remained stable in time. As the hydrogeology of the catchment is sufficiently homogeneous to avoid patterns being dominated by geological discontinuities, we were able to test the common assumptions that spatial patterns of LGD are controlled by sediment characteristics and the groundwater flow field (here interpolated from observation wells) and the potential of topographic indices as proxies for gradients of the groundwater flow field. We identified the following links between LGD patterns and external factors at Lake Hinnensee: Even though classified not as topography-, but as recharge controlled based on the water table ratio, large-scale LGD patterns were linked to the local topographic gradient and also to groundwater gradients derived from regression kriging, which also included topographic information. The explanatory power of these indices was strongest when derived locally up to a distance of 50 m from shoreline, and decreased with increasing distance from the lake. Small-scale LGD patterns in the north were linked to sediment heterogeneities: LGD patterns correlated positively with percentages of gravel and coarse sand and negatively with the percentage of fine sand. However, LGD patterns did not correlate with $k_{sat}$ values derived from slug tests. We assume that our findings are transferrable to similar lowland landscapes with quasi-homogeneous aquifers. As the water table ratio at this site indicated recharge control, we assume topography to have an even greater influence on LGD patterns in areas where groundwater tables classify as topography controlled. However, more complex hydrogeological settings which include discontinuities can override and mask the topographic signal.

Our results furthermore showed that predictions of LGD rates using regression models derived from correlation with external controls were associated with high uncertainties, but nevertheless allowed a rough estimation of LGD patterns. Topographic indices, such as elevation or slope, are often readily available. Analysing grain size distributions of lake sediment is labour intensive, but sediment cores taken with transparent sampling tubes can be easily analysed at least qualitatively. This information combined with information of topographic gradients can then be used to develop an effective and efficient

measurement design for more a detailed characterisation of spatial patterns of LGD that goes beyond the rough estimate that the linear regression models can provide.

Measuring VTPs with a 45 cm long needle-thermometer, is a fast and inexpensive method to determine LGD rates without disturbing the sediment. Correlation between LGD rates and temperature values measured in 30 cm depth also show that even single depth temperature measurements can provide at least some rough qualitative information on LGD patterns. While installing a fiber optic cable along the shore line would have the advantage of providing a large high resolution spatial data set as well as continuous measurements, the cable would need to be installed at a fixed depth > 20cm to provide reliable data, uninfluenced by solar radiation and boundary effects. Such an installation of a kilometre-long cable at a fixed depth is challenging and would significantly disturb the sediment, potentially causing preferential flow paths and changing the very fluxes we want to measure. We therefore favour the manual measurements using the thermistor-needle, as this method has furthermore the added advantage of providing temperature profiles and thus enabling us to obtain reliable quantitative estimates of LGD rates at a large number of locations. While seepage meters provide flux data directly they are less suitable for high numbers of measurements. Piezometers provide vertical hydraulic gradients, however, for the determination of LGD rates from this data, the necessary information on saturated hydraulic conductivities is often subject to high uncertainties.

From the experience gained in this study we would suggest the following protocol for future studies of LGD patterns and controls:

1) Determine topographic indices for the 50m region around the lake (broken down in partially overlapping subsections of a length representative of the variability in topography – in our study 100m with 50 m overlap) and combine this with groundwater flow field information from observation wells. Determine if system classifies as groundwater or topography controlled by using the water table ratio, if only limited well information exists, to get a better idea of the ground water table characteristics. 2) Predict LGD patterns based on this information. 3) Test predictions at 8 locations (more if feasible) by measuring VTPs and estimating LGD rates based on the heat transport equation. If possible perform additional single depth temperature measurements and compare the obtained qualitative LGD patterns with the ones predicted from topographic indices to further evaluate the reliability of the prediction. 4) Characterise small-scale variability at 2 of these locations (covering high and low inflow regions) by additional measurements at higher spatial resolution. 5) Use clear plastic tubes to sample sediment cores for quick visual inspection and rough classification according to grain size/permeability and relate this to the corresponding LGD at each of these locations. 6) If interested in the temporal stability of the LGD patterns, repeat step 3 at different points in time.

**Acknowledgments**

This study is a contribution to the Virtual Institute of Integrated Climate and Landscape Evolution Analysis –ICLEA– of the Helmholtz Association. We would like to thank Lisei Köhn, Christian Budach, Sigfried Tusche, Philip Müller, Markus Morgner, Henriette Wilke and Knut Günther for their help in the field, the divers Silke Oldorff and Frank Kroll for their help laying out the FO-DTS cable and the Mueritz National Park authorities for their cooperation. Furthermore we thank Matthias

Munz and Jana Carus for helpful discussions in the early stage of the manuscript. Bathymetry data were provided by the Ministerium für Landwirtschaft, Umwelt und Verbraucherschutz Mecklenburg-Vorpommern. We also thank two anonymous referees, whose suggestions improved the manuscript considerably. The study took place in the TERENO observatory of north-east Germany funded by the Helmholtz Association.

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

**Table 1: Dates, boundary conditions and use of vertical temperature profile surveys.**

| *Date* | *Groundwater temperature [°C]* | *Lake water temperature [°C]* | *Data analysis* |
|---|---|---|---|
| 24–25 August 2011 | 10.7 | 22.7 | Temporal stability of LGD patterns |
| 12–14 June 2012 | 8.9 | 18.2 | Spatial patterns of LGD along the shoreline |
| 16–17 July 2012 | 10.1 | 20.1 | Spatial patterns of LGD along the shoreline |
| 21–23 January 2013 | 7.0 | 0.0 | Spatial patterns of LGD along the shoreline |
| 17–25 July 2013 | 10.1-10.3 | 23.2 | Spatial patterns of LGD along the shoreline and LGD patterns with increasing distance from shore |

**Table 2: Groundwater temperatures and lake water temperatures depth profile measured during FO-DTS campaigns. \*1 measured at the nearby weather station, \*2 measured close to the lake**

| Date | Air temperature [•C] | Groundwater temperature [•C] | Temperature depth profile [•C] |
|---|---|---|---|
| 20 February 2014 | 6.8[*1] | 6.3 | 0m: 1.8 <br> -1m: 3.4 <br> -2m: 3.4 <br> -3m: 3.4 <br> -4m: 3.4 <br> -5m: 3.4 |
| 27 August 2014 | 19.4[*1] <br> 16.1[*2] | 11.6 | 0m: 18.9 <br> -1m: 18.7 <br> -2m: 18.7 <br> -3m: 18.5 <br> -4m: 18.4 <br> -5m: 18.4 |

**Table 3: Correlation coefficients (ρ), linear models describing the correlation between LGD and predictors and the coefficient of determination ($R^2$). ρ coloured in black indicate significant correlation (p-value < 0.05), light red indicate insignificant correlations (p-value > 0.05)**

| Predictor (x) | ρ | Models (LGD = ...) | $R^2$ |
|---|---|---|---|
| gravel | 0.61 | 49.84 + 1.78x | 0.25 |
| coarse sand | 0.62 | 14.03 + 5.27x | 0.46 |
| medium sand | 0.01 | - | - |
| fine sand | -0.7 | 127 - 2.12x | 0.48 |
| Silt | 0.01 | - | - |
| Coarse + fine sand | | $72.72 + 3.06x_1 - 1.35x_2$ | 0.54 |

**Table 4: Correlation coefficients (ρ) between LGD and far-field predictors calculated for upslope areas in certain topographical zones of influence (zi). ρ coloured in light red indicate insignificant coefficients (p-value > 0.05). ρ coloured in grey to black in dependence of the strength of correlation indicate significant coefficients (p-value < 0.05). gg is the abbreviation for groundwater gradients and ltg the abbreviation for low topographic gradient in direct vicinity to the lake shore.**

| $z_i$/ Predictors | Size | Mean elevation | Mean slope | Percentage of ltg | Mean gg -ordinary kriging- | Mean gg -regression kriging- |
|---|---|---|---|---|---|---|
| 25 m | 0.15 | 0.61 | 0.58 | -0.44 | 0.33 | 0.64 |
| 50 m | 0.03 | 0.62 | 0.64 | -0.30 | 0.36 | 0.65 |
| 100 m | -0.19 | 0.45 | 0.58 | 0.00 | 0.32 | 0.59 |
| 200 m | -0.31 | 0.23 | 0.54 | -0.02 | 0.33 | 0.55 |

**Table 5: Linear regression models describing the correlation between LGD and far-field predictors and the coefficient of determination ($R^2$). gg is the abbreviation for groundwater gradients and ltg the abbreviation for low topographic gradient in direct vicinity to the lake shore.**

| Predictor (x) | $zi_{25m}$ Models (LGD = ...) | $R^2$ | $zi_{50m}$ Models (LGD = ...) | $R^2$ |
|---|---|---|---|---|
| Elevation | -591.62 + 9.65x | 0.35 | -273.14 + 4.65 | 0.34 |
| Slope | 21.66 + 2.04x | 0.33 | 20.74 + 2.2x | 0.32 |
| Percentage of ltg | 52.61 - 1.29x | 0.11 | - | - |
| gg -ordinary kriging- | 30.12 + 141.11x | 0.09 | 27.18 + 167.90x | 0.12 |
| gg -regression kriging- | 21.16 + 62.30x | 0.37 | 21.08 + 64.36x | 0.36 |

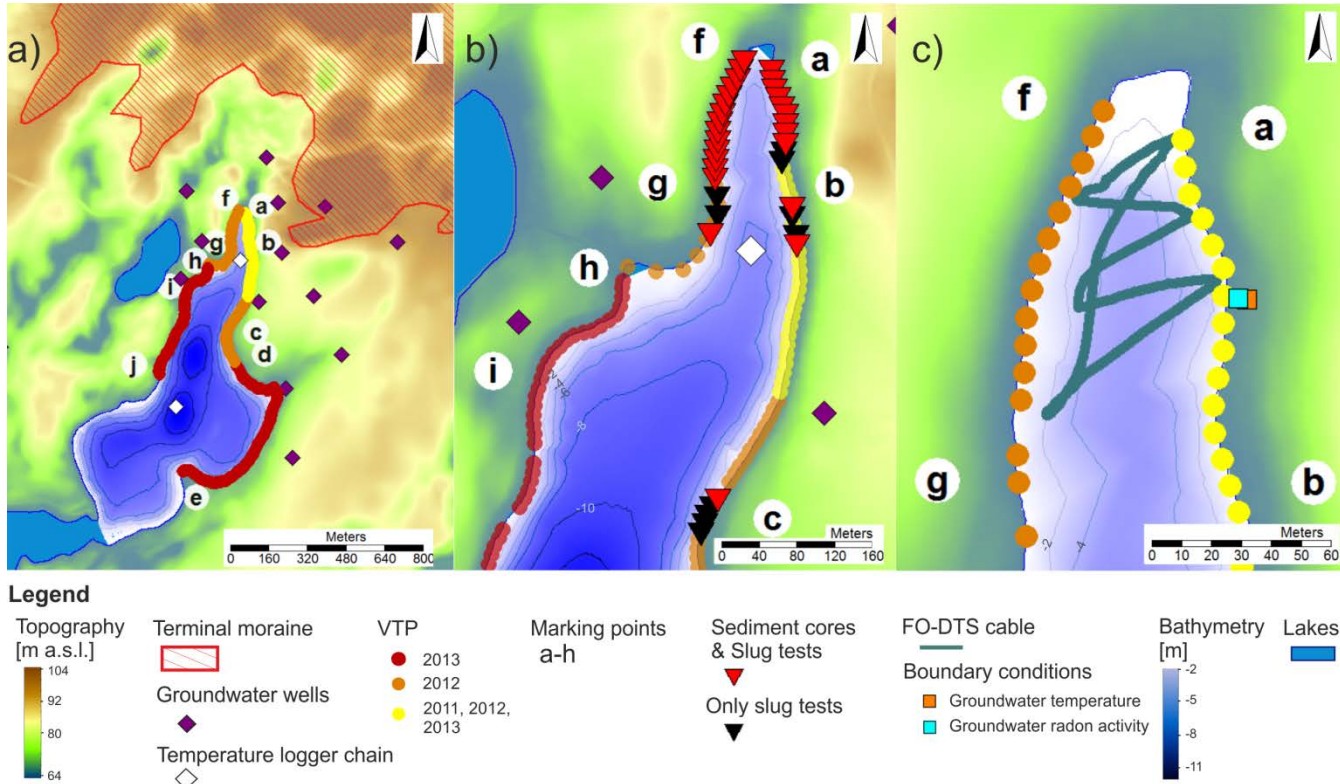

**Figure 1: Study site and experimental infrastructure. a) Overview of the study site with VTP measurement and locations of groundwater wells and temperature logger chains, b) Slug test and sediment core sampling locations, c) FO-DTS cable installation in the northern part of the lake.**

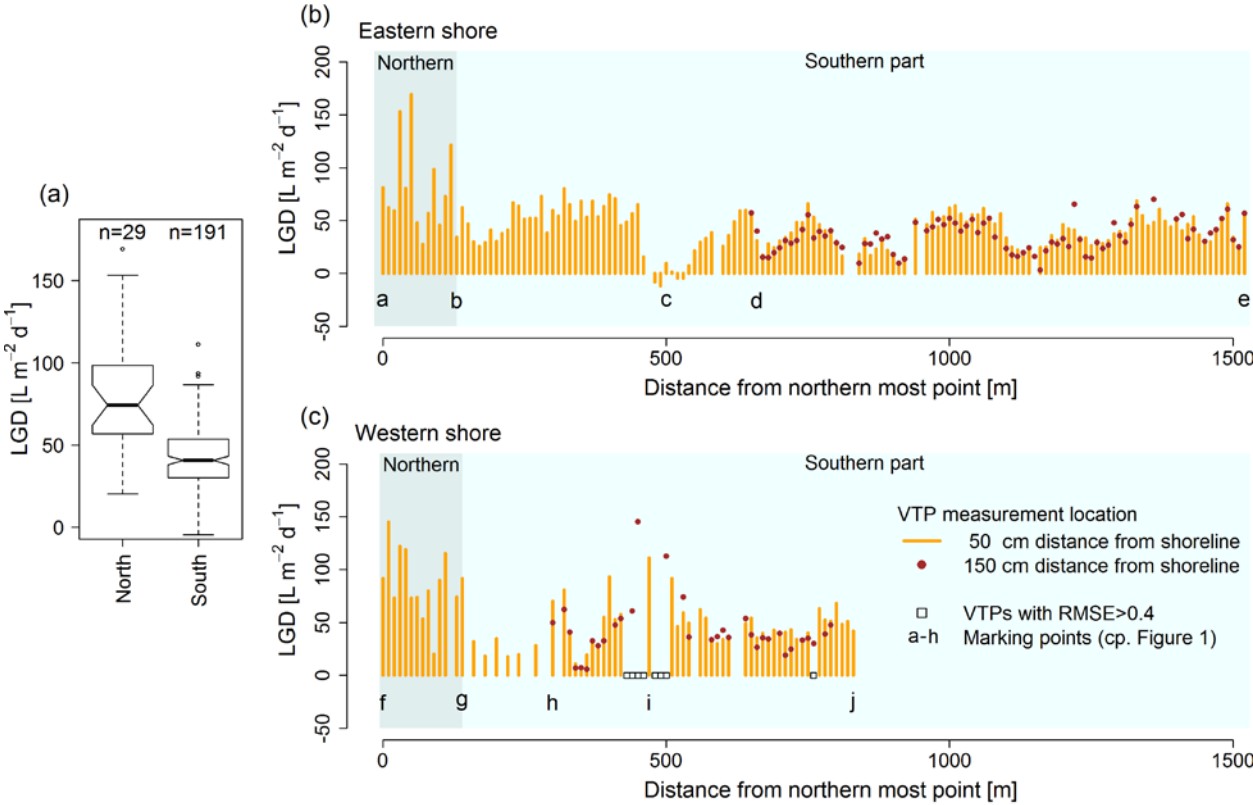

**Figure 2:** LGD estimated from VTPs measured at 50 cm and 150 cm distance from shoreline, (a) LGD distribution from VTPs in the northern and southern part measured at a distance of 50 cm LGD along eastern shore (b) and LGD along western shore (c). Locations where fits of the heat transport equation were poor (RMSE>0.4) are indicted with squares.

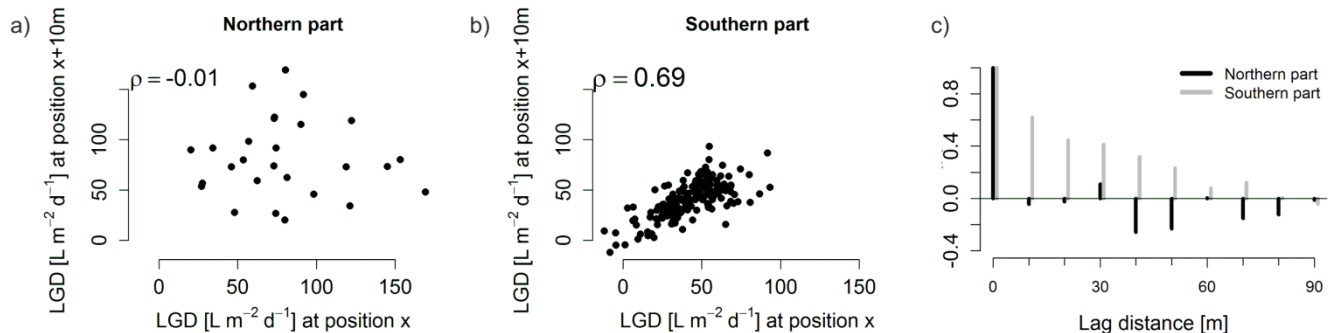

**Figure 3:** Correlation between neighbouring LGD measurement locations (distance 10m) in the northern part (a) and southern part (b), autocorrelogram for the LGD series of the northern and southern part of the lake (c).

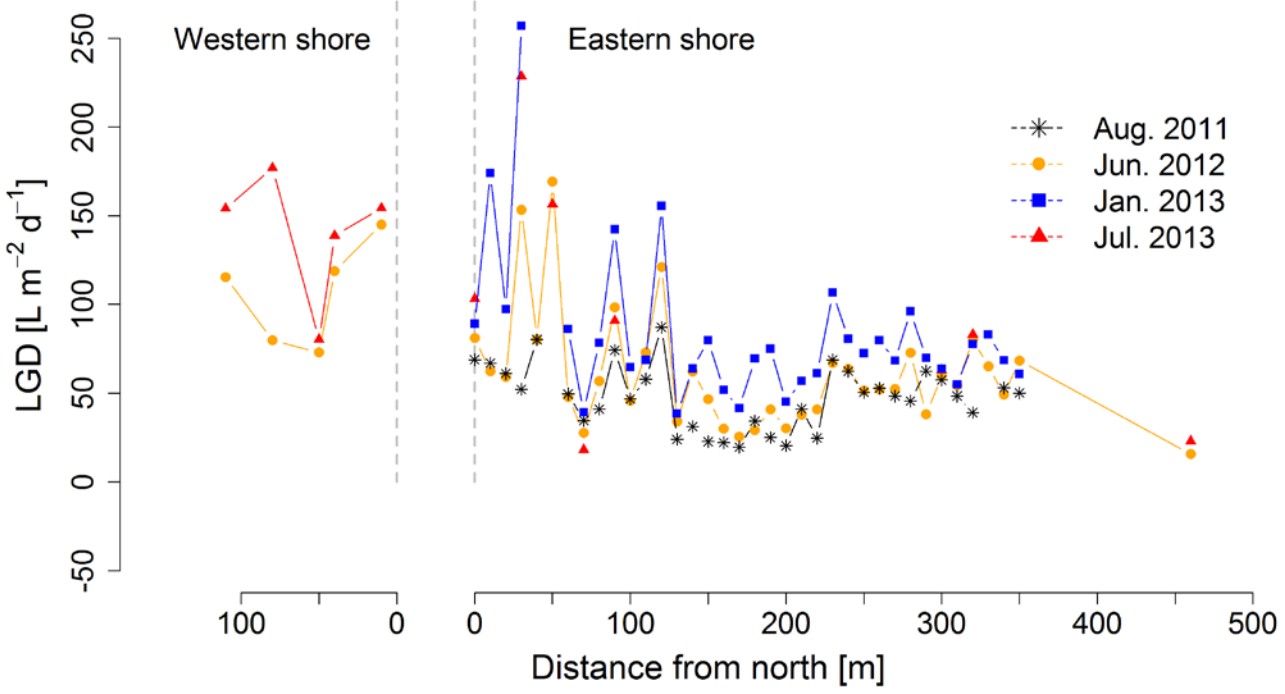

**Figure 4: Calculated LGD rates from repetitions of VTP measurements in August 2011, June 2012, January 2013 and July 2013 at the northern western and eastern shore.**

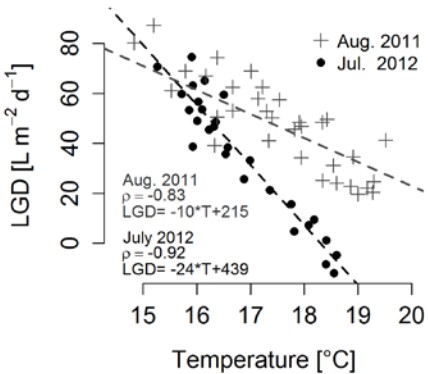

**Figure 5: Sediment temperature measured 30 cm below the sediment lake interface during two different VTP surveys vs LGD rates estimated from VTPs.**

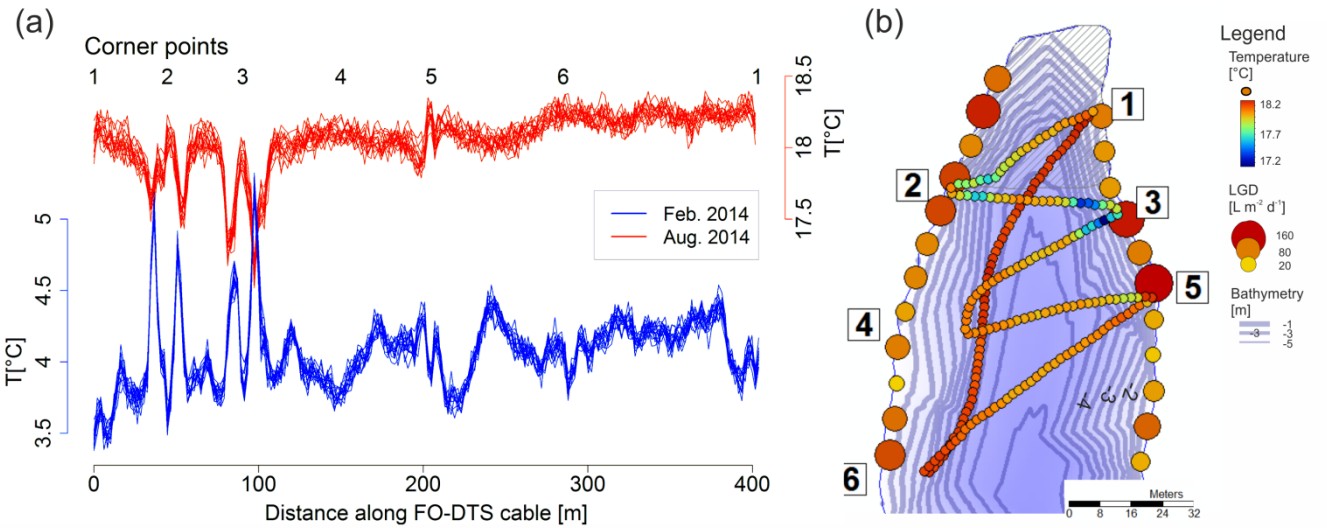

**Figure 6: Lake sediment temperatures measured with the FO-DTS system. (a) Temperatures measured in February and August along the FO-DTS cable. (b) Sediment temperatures measured with the FO-DTS in August 2014 (median) LGD rates along the shoreline were derived from VTPs measured in June 2012.**

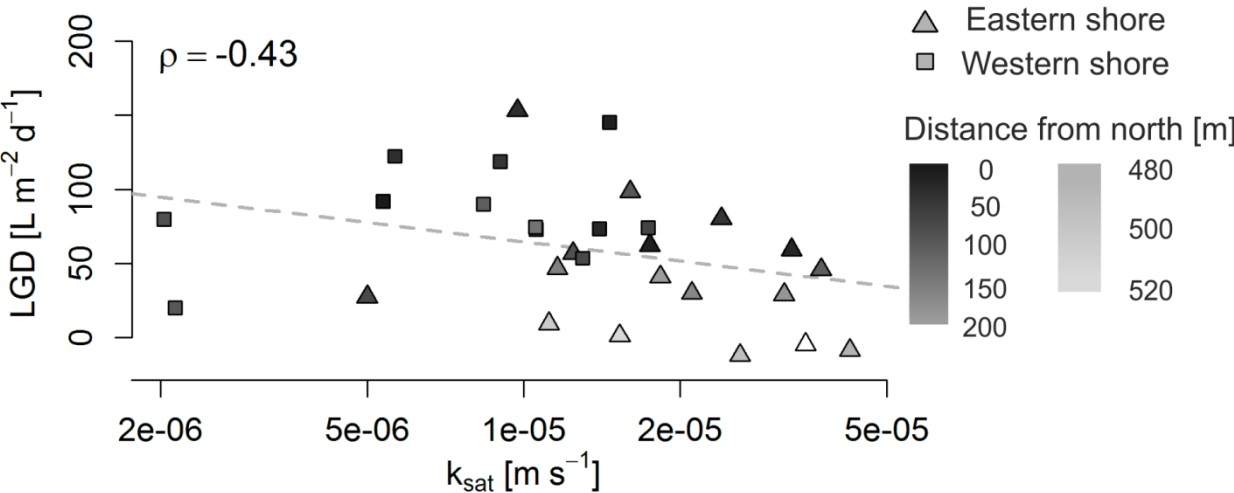

**Figure 5: LGD plotted against $k_{sat}$ values determined from slug test at the western and eastern shore, the grey shading indicates the distances of measurement locations from the northern tip of the lake.**

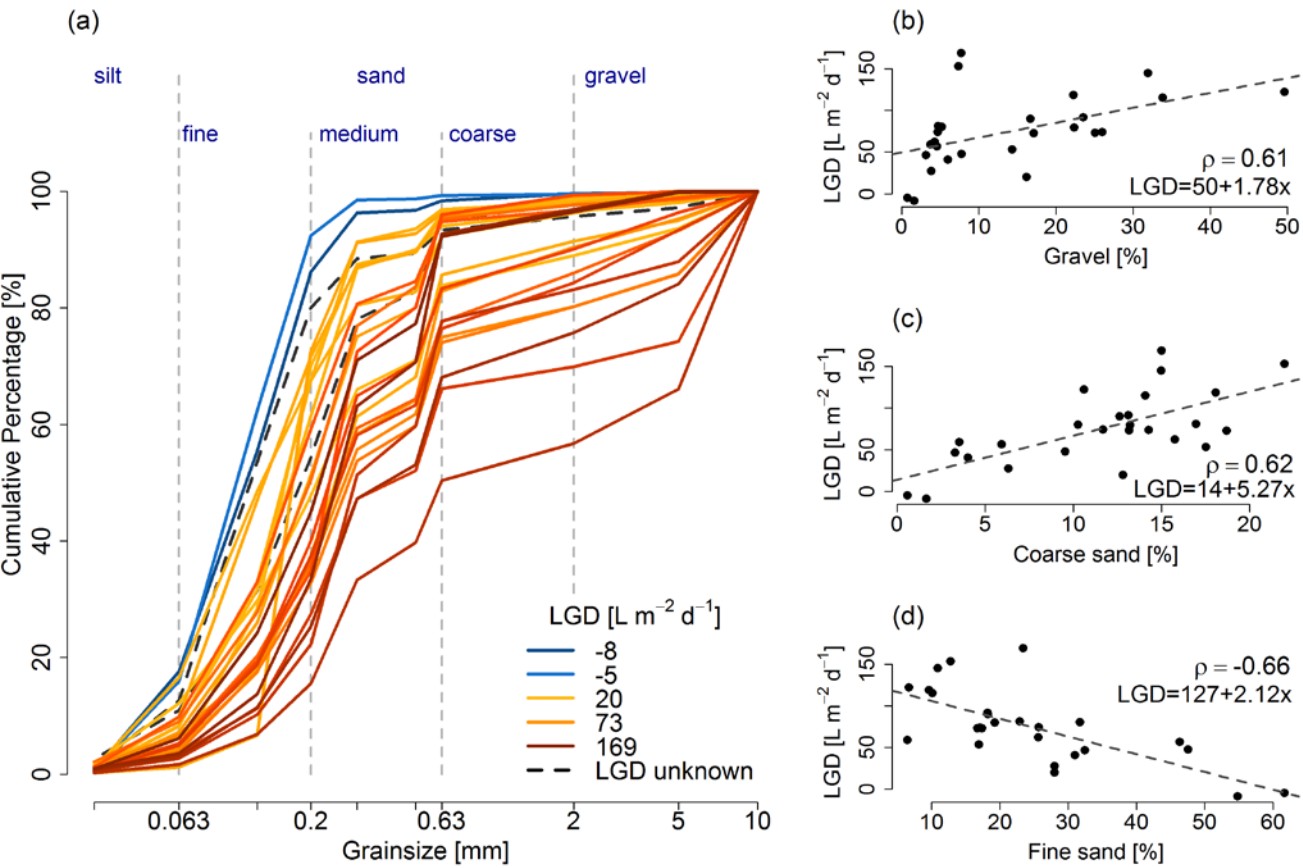

**Figure 6: (a) Grain size distributions averaged over the upper meter of the lake sediment from sediment cores coloured by the strength of LGD rate; (b–d) LGD rates are plotted against the grain sizes gravel (b), coarse sand (c) and fine sand (d).**

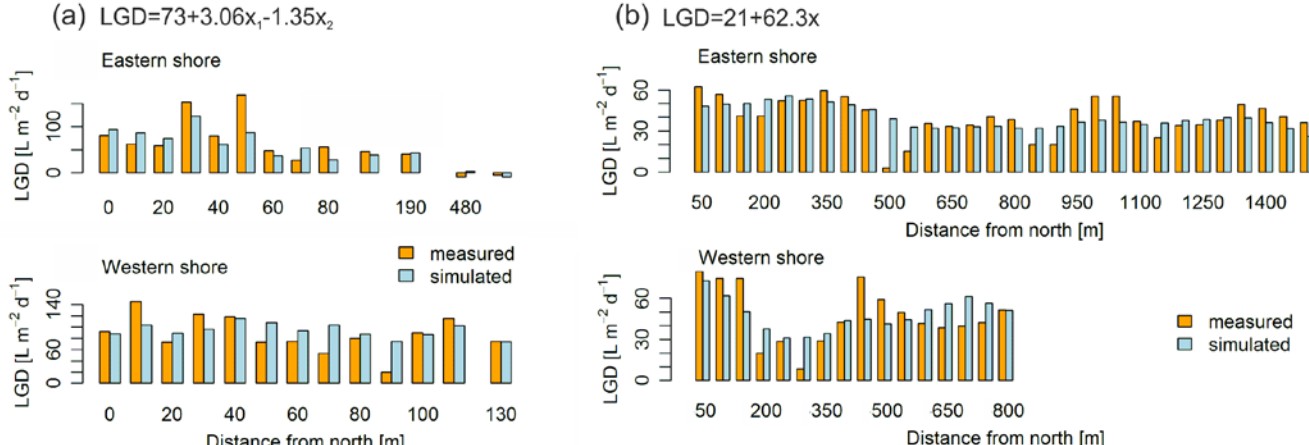

**Figure 7: Observed and calculated LGD distribution along the shore line. (a) Small scale patterns predicted using a multiple regression model with coarse sand and fine sand as predictor and (b) large-scale patterns predicted by the linear model based on groundwater gradients derived from regression kriging from zone $zi_{25m}$. Regression equations for both small and large-scale patterns are included in the upper right corner. In the small-scale variability equation $x_1$ stands for the fraction of coarse sand and $x_2$ for fine sand.**

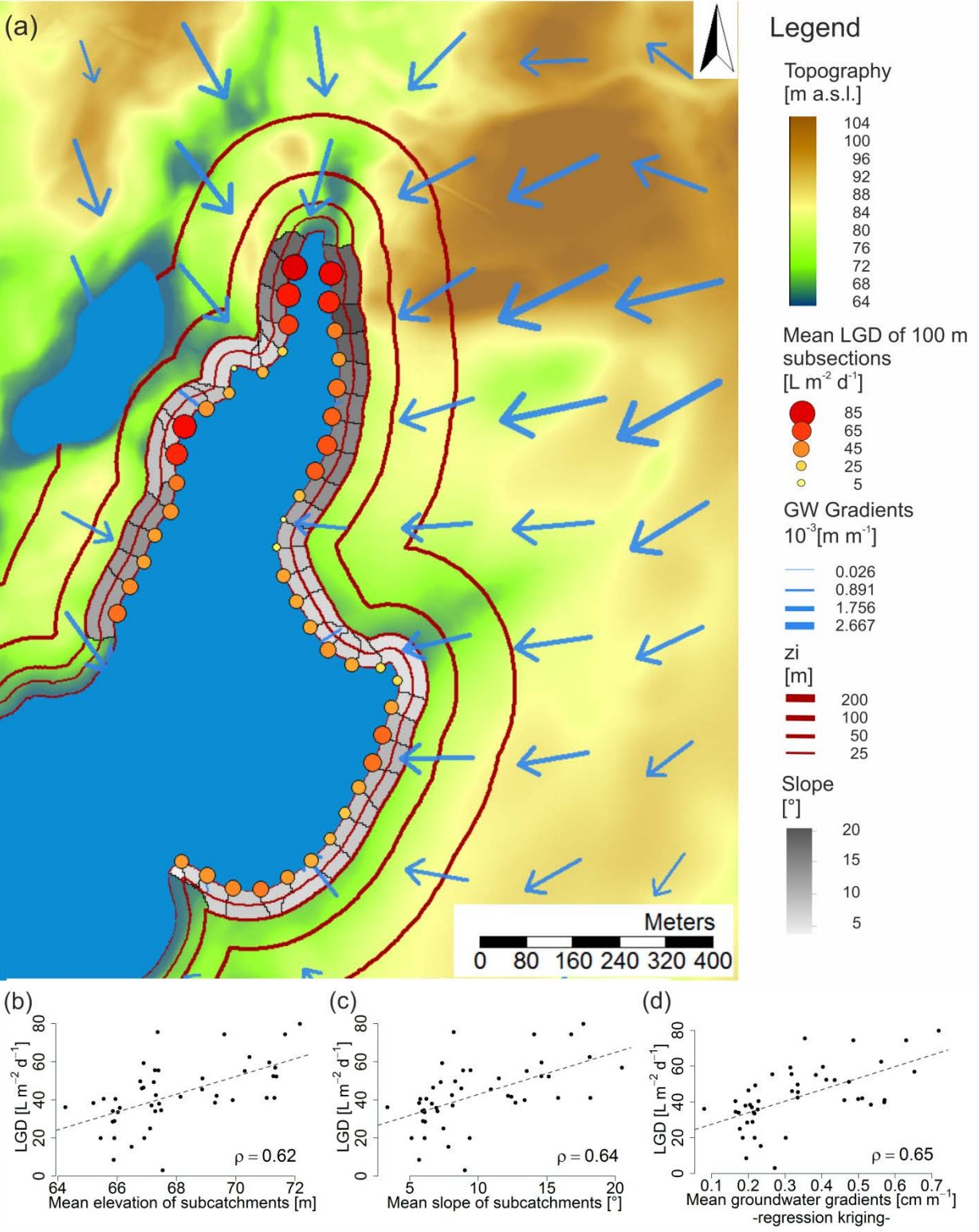

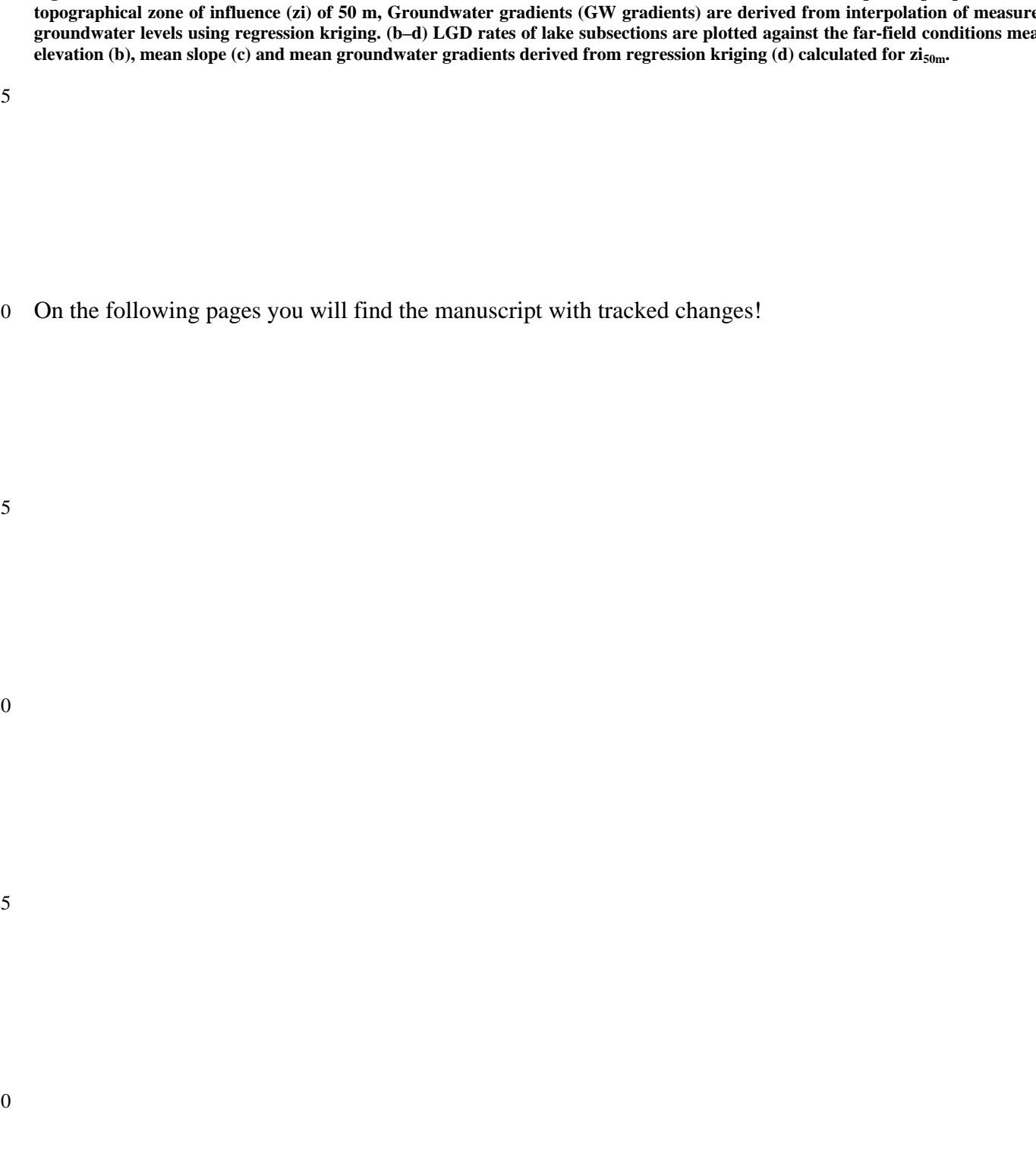

**Figure 8: Correlation between far-field conditions and LGD. a) LGD of lake subsections and mean slope of upslope areas for topographical zone of influence (zi) of 50 m, Groundwater gradients (GW gradients) are derived from interpolation of measured groundwater levels using regression kriging. (b–d) LGD rates of lake subsections are plotted against the far-field conditions mean elevation (b), mean slope (c) and mean groundwater gradients derived from regression kriging (d) calculated for $zi_{50m}$.**

10    On the following pages you will find the manuscript with tracked changes!

# Identifying, characterizing and predicting spatial patterns of lacustrine groundwater discharge

- Christina Tecklenburg[1], Theresa Blume[1]

[1]Helmholtz Centre Potsdam, GFZ German Research Centre for Geosciences, Section Hydrology, Potsdam, Germany

*Correspondence to*: Christina Tecklenburg (christina.tecklenburg@gfz-potsdam.de)

**Abstract.** Lacustrine groundwater discharge (LGD) can ~~play an important role for~~significantly affect lake water balances and lake water quality. However, quantifying LGD and ~~their~~ its spatial patterns is challenging ~~as~~ because of the large spatial extent of the aquifer-lake interface and pronounced spatial variability ~~is paired with a large spatial extent of the aquifer lake interface and factors controlling LGD patterns are not well understood.~~ This is the first experimental ~~based~~ study to specifically study these larger scale patterns with sufficient spatial resolution to systematically investigate how landscape and local characteristics affect the spatial variability in LGD. We measured vertical temperature profiles around a 0.49km² lake in north-eastern Germany with a needle-thermistor, which has the advantage of allowing for rapid (manual) measurements and thus, when used in a survey, high spatial coverage and resolution. Groundwater inflow rates were then estimated using the heat transport equation. ~~We used intensive field measurements including 520 vertical temperature profiles in the near-shore area,~~ These near-shore temperature profiles were complemented with sediment temperature measurements with a fibre-optic cable along 6 transects from shoreline to shoreline and radon measurements of lake water samples to qualitatively identify LGD patterns ~~at a lake in north-eastern Germany~~ in the off-shore part of the lake. As the hydrogeology of the catchment is sufficiently homogeneous (sandy sediments of a glacial outwash plain, no bedrock control) to avoid patterns being dominated by geological discontinuities, we were able to test the common assumptions that spatial patterns of LGD are mainly controlled by ~~s~~Sediment characteristic~~s~~ and the groundwater flow field. We also tested the assumption that~~s, topographic indices~~ and topographic gradients can be used as a proxy for gradients of the groundwater flow field. Thanks to the extensive data set these tests could be carried out in a nested design, considering both small and large-scale variability in LGD. ~~were considered as potential controls of small scale and large scale LGD patterns. The results revealed~~We found that LGD was concentrated in the near shore area, but ~~with~~ along-shore variability was high, with specific regions of ~~stronger~~ higher rates and higher spatial variability ~~in the northern part of the lake~~. Median inflow rates were 44 L m$^{-2}$ d$^{-1}$ with maximum rates in certain locations going up to 169 L m$^{-2}$ d$^{-1}$. ~~LGD generally decreased with distance to shore and o~~Offshore LGD was ~~insignificant~~ negligible except for ~~some~~ two local hotspots ~~of LGD~~ on steep steps ~~towards the lake bottom~~ in the lake bed topography. Large-~~-~~scale groundwater inflow patterns were correlated with topography and the groundwater flow field whereas small-~~-~~scale patterns correlated with grain size distributions of the lake sediment. These findings confirm results and assumptions of theoretical and modelling studies more systematically than was previously possible with coarser sampling designs. However, we also found that a significant fraction of the variance in LGD could not be explained by these controls alone and that additional processes need to be considered. ~~Regression~~ While regression models using ~~external~~ these controls as explanatory variables had limited power to predict LGD rates, ~~but~~ the results nevertheless encourage the use of topographic indices and sediment ~~heterogeneities~~ heterogeneity as an aid for targeted campaigns in~~experimental designs~~ future studies of groundwater discharge to lakes.

## 1 Introduction

By linking groundwater with the surface water body, lacustrine groundwater discharge (LGD) can strongly control lake water quality and lake water budgets. Hence, all processes affecting quantity and quality of groundwater could also affect lake water quantity and quality (Winter et al., 1998; Rosenberry et al., 2015). To understand the vulnerability of groundwater

dominated lakes it is not only important to know the total volume of groundwater lake exchange, but also the spatial patterns of LGD (Meinikmann et al., 2013; Lewandowski et al., 2015).

## 1.1 Spatial patterns of lacustrine groundwater discharge and their potential controls

In an isotropic homogenous aquifer, the exchange between groundwater and lake is expected to follow a distinct pattern
along a 2D transect: as sloping groundwater water tables meet the flat surface of the lake, groundwater inflow is strongest in
close proximity to the shoreline and decreases exponentially with distance to shore (McBride and Pfannkuch, 1975).
However, isotropic and homogenous conditions rarely exist and spatial distribution of groundwater inflow differs strongly
from lake to lake (Rosenberry et al., 2015). Experimental studies highlighted a large variety of observed exchange patterns
including decreasing seepage with distance from shoreline (McBride and Pfannkuch, 1975; Brock et al., 1982; Cherkauer &
Nader, 1989; Kishel & Gerla 2002), increasing seepage with distance from shoreline (Cherkauer and Nader, 1989; Schneider
et al., 2005; Vainu et al., 2015), local hotspots of off-shore seepage (Fleckenstein et al., 2009; Ono et al., 2013) and a high
small--scale variability in near shore zones (Kishel and Gerla, 2002; Blume et al., 2013; Neumann et al., 2013; Sebok et al.,
2013). Most often complex hydrogeological settings are the reason for deviations from the theoretical pattern of LGD
(Rosenberry et al., 2015). For example, it was found that off-shore LGD was forced bycaused by local connections with a
deeper aquifer (Fleckenstein et al., 2009; Ono et al., 2013) or forced by aresulted from local thinning of low permeable lake
sediment (Cherkauer & Nader 1989).

In general, the position of a lake in its regional groundwater flow system determines if a lake receives groundwater, loses
water towards the groundwater or both (Born et al., 1974). However,As the groundwater flow field is often not well known,.
L landscape topography can help to determine the groundwater flow field in humid regions and homogenous aquifers, where
groundwater tables are likelyare assumed to follow the topography (Toth, 1963). However, Haitjema and Mitchell-Bruker
(2005) found that the groundwater table is only topographically controlled if the ratio of groundwater recharge over
hydraulic conductivity is sufficiently large and that often groundwater tables are indeed not topography, but recharge
controlled (for a US wide classification of these groundwater table controls see also Gleeson et al. 2011).

Little is known about controls of small--scale variability of LGD. LGD is driven by the hydraulic gradients between lake and
aquifer and controlled by the hydraulic conductivity. So far, there is no clear picture about the role of lake sediment
characteristics in controlling LGD patterns and observations seem to be very site specific. For example, Kidmose et al.
(2013) found that low permeable lacustrine sediments can completely prevent groundwater upwelling, whereas Vainu et al.
(2015) observed LGD through low permeable lacustrine sediments. Kishel and Gerla (2002) associated small--scale
variabilities in LGD with small--scale heterogeneities in hydraulic conductivities (Kishel & Gerla, 2002), but in contrast,
Schneider et al., (2005) found no correlation between seepage rates and sediment characteristics. Methods used in these
studies include seepage meters (e.g. Kidmose et al. 2010, 2013, Kishel and Gerla, 2002, Schneider et al. 2005, Vainu et al.
2015), and piezometers (Kishel & Gerla, 2002).

**1.2 Pattern identification**

Although several methods exist to quantify groundwater-lake exchange (Rosenberry et al., 2015), measuring LGD patterns is challenging as we are faced with pronounced spatial variability across the large extent of the aquifer-lake interface. Heat as a natural tracer of groundwater-surface water interactions has received increasing attention in the last decade (Rau 2014). When groundwater and surface water temperatures differ significantly, sediment surface temperature can be used to localize groundwater inflows in lakes (Blume et al.; 2013; Sebok et al., 2013). Using a fibre optic distributed temperature sensing (FO-DTS) system for this purpose has the advantage of providing precise temperature measurements with a high spatial resolution along fibre optic cables up to a length of several kilometres (Selker et al., 2006). Although sediment surface temperatures do not allow a direct estimation of water exchange fluxes, sediment temperature anomalies can be taken as an indicator for groundwater inflows (Blume et al., 2013; Sebok et al., 2013). Another method which uses heat as a tracer and which has been used successfully to investigate exchange patterns between lakes and groundwater are vertical temperature profiles (VTP) (Blume et al., 2013; Meinikmann et al., 2013; Neumann et al., 2013; Sebok et al., 2013). In contrast to the FO-DTS method, VTP measurements allow the calculation of exchange rates by using the analytical solution of the heat transport equation (Schmidt et al., 2006). Temperature profiles can be measured manually or continuously using profile probes. The measurement of radon activity can also help to identify groundwater inflows (Kluge et al., 2012; Ono et al., 2013; Shaw et al., 2013). Elevated radon activities in surface water indicate groundwater inflow as radon is naturally enriched in groundwater but degasses quickly in surface water.

Existing lake studies have investigated LGD patterns with either a high spatial resolution (1–2 m²) but a local focus (10 m × 17 m – 25 m × 6 m) (Kishel & Gerla, 2002; Blume et al.; 2013; Sebok et al., 2013) or focused on the entire lake, but used a relatively low spatial resolution (measurements along the shoreline: every 200 m – 3000m) (Schneider et al., 2005; Meinikmann et al., 2013; Shaw et al., 2013). However, the experimental effort required rarely allows their extension to cover the lateral, along-shore dimension in sufficient extent and detail to identify the spatial variability and patterns of LGD along the shore-line.

**1.3 Objectives**

Identifying the processes and structures controlling LGD patterns is the key to ~~reliably~~ predicting them reliably (Grimm 2005). The aim of this study is the characterization of inflow patterns as well as the identification of their controls. The ability to identify patterns and their controls strongly depends on the spatial resolution and the extent of the applied experimental methods. ~~Existing lake studies have investigated LGD patterns with either a high spatial resolution (1–2 m²) but a local focus (10 m × 17 m – 25 m × 6 m) (Kishel & Gerla, 2002; Blume et al.; 2013; Sebok et al., 2013) or focused on the entire lake, but used a relatively low spatial resolution (measurements along the shoreline: every 200 m – 3 km) (Schneider et al., 2005; Meinikmann et al., 2013; Shaw et al., 2013).~~ By taking measurements with a high spatial resolution over large parts of the lake we are closing ~~this~~ the observational gap between high-resolution "plot"-scale studies (focusing

on a small shore line segment) and low resolution larger-scale studies (see section 1.2) and open the possibility to truly investigate not only shore-lake transects or plots, but along-shore spatial variability and patterns of LGD.

The study design aimed at answering the following research questions:

- How variable is LGD in space?
- Can we identify patterns?
- Can we link the patterns to external factors/controls?
- Can we use LGD patterns to test if groundwater tables are topography controlled?
- Can we predict LGD patterns?

To study these research questions we chose Lake Hinnensee, a typical post-glacial lake located in the intensively monitored TERENO observatory in the lowland landscape of northeast Germany. Strong water level declines observed in the last decades at this lake as well as at others in the region are currently under investigation. At this lake strong water level fluctuations were observed in the last decades, but processes and mechanisms causing these lake level fluctuations are yet not well understood. This lake system has the additional advantage that the upper unconfined aquifer in which the lake rests can be considered as largely homogeneous and isotropic (sandy sediments of a glacial outwash plain, no bedrock control). Therefore LGD patterns are unlikely to be dominated by geological discontinuities, and we were able to test the common assumptions that spatial patterns of LGD are controlled by sediment characteristics and topography as a proxy for gradients of the groundwater flow field.

To identify LGD patterns, we measured VTPs in the near shore area and used FO-DTS measurements and radon sampling in the off-shore area. VTPs were used to quantify LGD rates, whereas FO-DTS measurements and radon sampling were used as qualitative tracers to detect thefor presence andor absence of off-shore LGD. As potential controls of LGD patterns we considered a) heterogeneity of the lake sediment, such as variability in hydraulic conductivity and grain size distributions, b) topographic indices (elevation, slope) as a proxy for the flow field and c) the local groundwater flow field derived from groundwater levels measured in observation wells.

## 2. Methods

### 2.1 Study site

Lake Hinnensee is located in northeast Germany in the Müritz National Park (53°19'30.6"N, 13°11'16.2"E) and is one of the focus areas of the TERENO observatory Northeast Germany. The landscape of the Müritz National Park was shaped by the last glaciation and is dominated by lakes. Lake Hinnensee was formed as a glacio-fluvial tunnel valley and is located within the outwash plain. Borelogs of the 16 observation wells installed around the lake show largely homogeneous sandy sediments. The terminal moraine is situated north of the lake (Figure 1). Lake Hinnensee has a mean depth of 7 m with a maximum depth of 14 m and is connected to Lake Fürstenseer See in the south. The two lakes together cover an area of 2.68 km², the size of Lake Hinnensee is 0.49 km². The lake system has no surface water inflow or outflow, apart from a two minor

ditches connected with the Lake Fürstenseer See that only become active at very high lake level. Since 2011, when first LGD measurements were conducted at Lake Hinnensee, the ditches were only active for a period of four months (maximum observed inflow: 0.0083 m³ s⁻¹, 22 February 2012, maximum observed outflow: 0.0030 m³ s⁻¹, 11 May 2012). The connection to Lake Fürstenseer See is not assumed to influence LGD patterns of Lake Hinnensee, as the general flow direction of the groundwater flow field is from north to south with water leaving the lake system at the southern end of Lake Fürstenseer See. The relief of the lake catchment is hilly in the north, with steep slopes down to the lake, and more gentle slopes and lower elevations towards the south (Figure 1). Elevations range between 63 m a.s.l. and 115 m a.s.l.. The lake is surrounded by forest. The mean annual precipitation is 610 mm and the mean annual temperature is 8.1 °C (1901–2005 Neustrelitz, DWD-German Weather Service).

## 2.2 Estimating lacustrine groundwater discharge (LGD)

We applied three different methods to determine LGD patterns: VTPs in the near shore region and FO-DTS and radon in the off shore area. The VTPs allowed us to determine LGD rates, while the other methods were only used as indicators for the presence and absence of LGD. As the main body of the study focusses on the temperature-based methods, the radon methodology and results are described in the appendix.

### 2.2.1 Near-shore LGD derived from vertical temperature profiles (VTPs)

VTPs were used to estimate the spatial variability of LGD rates along the shoreline. Profiles were measured 50 cm away from the shoreline every 10 meters along 2.39 km of the shoreline. The dataset covers 62 % of the total shoreline (Figure 1). The VTPs were measured during five field campaigns in August 2011, June and July 2012, January and July 2013 (Table 1, Figure 1). In July 2013, sediment temperatures were additionally measured in 150 cm distance from the shoreline in order to analyse the trend of LGD with increasing distance to shore. Measurements from in August 2011 and January 2013 were conducted only on a 350 m long subsection of the shoreline in the north east of the lake in order to analyse the temporal stability of the observed patterns (Figure 1).

One VTP consisted of six temperature measurements: one at the sediment–water interface and five in the saturated sediment at 5, 10, 20, 30, and 40 cm depth. Temperatures were measured with a high-precision digital thermometer (Greisinger GMH 3750) and a corresponding thermocouple Pt100 thermistor with an accuracy of ± 0.03 °C. The needle had a length of 45 cm and a diameter of 3 mm.

LGD was calculated from the measured VTP using the analytical solution of the 1-D heat flow equation from Bredehoeft and Papaopulos (1965). Assuming a vertical water flux along the temperature profile and steady state temperatures at the sediment–water interface, sediment temperature at a specific depth is calculated as follows:

$$Tmod(z) = \frac{exp^{\frac{qz*pfcf*z}{kfs}} - 1}{exp^{\frac{qz*pfcf*L}{kfs}} - 1} * (T_L - T_0) + T_0, \tag{1}$$

where $qz$ is the vertical water flux [m s$^{-1}$] (positive for groundwater gaining), $pfcf$ is the volumetric heat capacity of the fluid [J m$^{-3}$ K$^{-1}$], $z$ is the depth below the upper boundary [m], $kfs$ is the thermal conductivity of the sediment [J s$^{-1}$ m$^{-1}$ K$^{-1}$], $L$ is the extent of the exchange zone and the depth of the lower boundary [m], and $T_L$ is the temperature of the lower and $T_0$ of the upper boundary (°C). The values of $pfcf$ of water and $kfs$ of lake sediment were taken from Stonestrom and Constantz (2003). $Pfcf$ of water was set to 4.19×10$^6$ J m$^{-3}$ K$^{-1}$ and $kfs$ to 2 J s$^{-1}$ m$^{-1}$ K$^{-1}$, a typical value for sandy sediment, which was the dominant grain size in the upper meter of lake sediment.

Usually the upper boundary is the sediment-water interface (Schmidt et al. 2006; Blume et al., 2013; Meinikmann et al., 2013). But at locations with shallow water depths in lakes, temperatures in the near-surface sediments can be strongly affected by daily temperature variations and thus violate the upper boundary condition of the steady state model. To avoid unreliable LGD calculations due to biased temperatures at the upper boundary, we instead used the temperatures measured at 10 cm sediment depth. A shift of the upper boundary from the sediment-water interface to a depth of 10 cm had a negligible effect on the estimation of the LGD rate assuming steady state conditions. This was validated with theoretical temperature profiles. A shift of the boundary condition to a depth of 10 cm caused a maximal deviation in the estimation of exchange rates of 1 L m$^{-2}$ d$^{-1}$ and the error decreased with increasing flowrates.

For the lower boundary, we used the shallow groundwater temperature measured in close vicinity of the lake (Figure 1c). For the length of the exchange zone, we tested different values. The quality of LGD estimation increased with increasing L, but was insensitive for values larger than 3 m. Thus, L was set to 3 m.

The exchange rate was optimized by minimizing the root mean square error (RMSE) between measured and calculated sediment temperatures as described in Schmidt et al. (2006):

$$RMSE = \sqrt{\frac{1}{n}\sum\left(T_{meas(z)} - T_{mod(z)}\right)^2}, \tag{2}$$

The quality of the fit between measured and modelled sediment temperature was also visually checked using plots of the measured and modelled VTPs. Fits with the RMSE greater than or equal to 0.4 °C were not used for further analyses as differences between modelled and measured data were considered too large.

Estimated LGD values were analysed for their lateral spatial variability using VTPs measured at a distance of 50 cm from shoreline, for their trend with increasing distance from shore using VTPs measured at 50 cm and 150 cm distance from shore and for their temporal stability using VTPs measured in the different years (Table 1).

Spatial variability ~~and correlation~~ of LGD ~~along the shoreline~~along different distances along the shoreline ~~was~~were analysed ~~with~~using autocorrelation plots and autocorrelation values (|ρ|) as described in Caruso et al. (2016). High autocorrelation ~~(|ρ|)~~ indicate~~s~~ that LGDs ~~for given~~along a given stretch of the shoreline ~~lag distances~~ are correlated (i.e. if LGD is high in a certain location it is also likely to be high at 10m distance), ~~which are in case of 10 m neighboured measurement locations,~~ whereas |ρ| < 0.2, indicate that LGDs are uncorrelated and strong spatial variability exists. ~~As a second measure, we calculated the median absolute deviation (MAD) of the LGD distribution for different subsections of the lake shore. In order to analyse the trend in LGD with increasing distance from shore line, we calculated differences between LGD rates measured at 50 cm and 150 cm distance from shore.~~ In order to analyse the temporal stability of spatial patterns we ~~analysed~~ used the differences ~~calculated the correlation~~ between the LGD rates measured ~~in different surveys~~at different points in time and calculated the correlation between the VTP surveys using the Spearman's rank correlation coefficient (ρ). Correlations were regarded as significant for p-values smaller than 0.05~~. Differences between LGD rates measured in different years were quantified with the RMSE.~~

To test if single sediment temperature measurements instead of profiles could be used as a quickly measureable qualitative indication for LGD spatial patterns, we determined the correlations between sediment temperatures at all individual depths with LGD rates determined from the full profiles.

### 2.2.2 Lake sediment temperature anomalies as indicators for offshore-LGD based on fibre optic distributed temperature sensing (FO-DTS)~~Identifying hot-spots of off-shore LGD with fibre optic distributed temperatures sensing (FO-DTS)~~

To identify offshore groundwater inflow patterns, we measured sediment surface temperatures with a 500 m long FO-DTS cable installed permanently along 6 transects through the northern part of the lake (Figure 1c). Two divers ensured good contact of the cable to the lake sediment and also tracked the location of the cable with a differential GPS system (Topcon GR-3) installed on a buoy.

The technology of the FO-DTS is based on the detection of the Raman scattering of a laser pulse through the optical fibre (Ukil et al. 2012). For our measurements, we used a Sensornet Halo device with a sampling resolution of 2 m and a measurement precision of 0.05 °C.

We carried out two measurement campaigns, in February and in August 2014 (Table 2). In February the DTS measurements were taken between 18:49 and 19:17 CEST with a temporal resolution of 2 minutes. We used a single ended set-up with a double ended mode (four channels, two in each direction) and two calibration baths, a warm bath (25.5°C) at one end and a cold bath (0°C) at the other end.

In the second campaign from the 27–28 August 2014 measurements extended over 24 hours, from 18:43 on the 27[th] until 18:45 CEST the next day with a temporal resolution of 2 min. The setup was the same as in February, but additionally both cable ends were run through the cold bath (warm bath: 20.9°C, cold bath 0.1°C).

The trend and offset in the DTS temperature data were corrected using external temperature loggers in the calibration baths (February: Greisinger GMH 3750 (accuracy: ± 0.03 °C); August: HOBO TidbiT v2 Water Temperature Data Logger (accuracy: ± 0.21 °C)).

All four channels showed the same pattern with only small differences in absolute temperature values and further analyses were based on one of the four traces. Sediment temperatures in August were strongly affected by solar radiation. Analysis of 24 hour amplitude or daily minimum temperature did not provide useful information of groundwater inflow patterns as the impact of solar radiation was too strong and spatially ~~irregular~~variable. Sebok et al. (2013) recommended using night time data to avoid the uncertainties caused by solar radiation. However, at night shallow near shore water cooled down and it was not easy to distinguish if temperature shifts resulted from groundwater inflow or resulted from a decrease of water temperature due to decreasing air temperature. We thus chose a time window in which the temperature at the near shore shallow region and in the deeper region of the lake were very similar and groundwater inflow induced temperature shifts were easy to identify. This time window was from 18:43 to 19:11 CEST on ~~the~~ August 27th.

Temperature depth profiles of the lake were available with 1 m spatial resolution (HOBO Water Temperature Pro v2 Data Logger, accuracy: ± 0.21 °C). In winter we had only one profile in the central part of the lake, but in August a second profile further north was available (Figure 1a, b). The groundwater temperature was measured in a piezometer (OTT Orpheus Mini, accuracy: ± 0.5 °C) close to the lake (Figure 1c) and air temperature data were available from a weather station 1.5 km away and in August an additional air temperature data logger (of the same type as used for the water profiles) was installed directly at the lake.

~~During the August campaign, we additionally measured VTPs along the cable, at five measurement locations at the eastern shore and four measurement locations at the western shore.~~

### 2.2.3 Radon as indication for offshore LGD

~~Radon ($^{222}$Rn) is produced within the natural decay chain of uranium and has a half-life of 3.82 days. Radon occurs naturally in the aquifer matrix and as it is soluble, groundwater is enriched in radon as it passes through the aquifer matrix. However, once radon-enriched groundwater is in contact with air, radon degasses quickly. Consequently, radon activities in surface water are low. Due to the pronounced differences between groundwater and surface water concentrations, elevated radon concentrations can be used as an indicator for groundwater inflows as shown for example by Kluge et al. (2012), Ono et al. (2013) and Shaw et al. (2013). In this study, we used radon as a qualitative indicator of presence or absence of off-shore LGD. We took water samples of 1.5 L or 1.75 L from the bottom of the lake (lower end of the water column) at 19 locations (Figure 1) during three campaigns (5 June 2013, 17 September 2013 and 21 August 2014). Water samples taken at the third date (Figure 1c) were taken by divers. All other samples were taken from a boat using a peristaltic pump. Contact between water samples and air was minimized and bottles were filled without air bubbles. To test the ability of radon concentrations to identify groundwater inflow at Lake Hinnensee, we took water samples from two near-shore VTP locations where LGD rates were known and significantly different, one with strong inflow (153 L m$^{-2}$ d$^{-1}$, located 40 m south from northern tip at~~

the eastern shore) and one with medium inflow (48 L m⁻²·d⁻¹, located 60 m south from northern tip at the eastern shore). In order to estimate the radon activity of the groundwater a groundwater sample was taken in a nearby piezometer (Figure 1e). The water samples were analysed using the alpha spectrometer RAD7 (Durridge Company Inc., Bedford, MA, USA) with the big bottle rad H₂O extension. Analysing time of the RAD7 was extended to 3.5 hours or longer to reduce measurement uncertainties. To remove background concentration of radon in the measurement device, the device was purged for 3.5 at least hours with fresh air between water samples. Relative humidity was kept below 8 % during measurements. Most of the samples were analysed in the first days directly after sampling. Maximum delay between sampling and analysis was one week. The data were decay corrected in dependence of the time between sampling and analysis.

### 2.3 Identifying controls of LGD patterns

In order to identify the controls of the observed LGD patterns, we characterized both the near-and the far-field conditions and correlated these characteristics with LGD rates using Spearman's rank correlation coefficient. At the local scale (near-field conditions) this includes sediment characteristics, while at the larger scale (far-field conditions) we considered topographic indices and the groundwater flow field as the most likely controls.

### 2.3.1 Sediment heterogeneity as a small--scale control on LGD patterns

Hydraulic conductivity from slug tests

At 37 VTP positions (Figure 1b), slug tests were performed to estimate hydraulic conductivity ($k_{sat}$) of the near-surface sediment. Slugtests were carried out in piezometers with an inner diameter of 36.4 mm. The ~~filter~~ screen placed on the lower end of the piezometer had a length of 10 cm and consisted of 4 mm diameter perforations in the PVC tube wrapped with fine mesh. The midpoint of the ~~filter~~ screen was 50 cm below the sediment surface. To minimize interference with the temperature profile measurements, the piezometers were installed at 50 cm distance. For the rising-head tests water was quickly removed out of the piezometer using a peristaltic pump. Recovery of the water table was measured with automatic pressure logger (HOBO 13-Foot Fresh Water Level Data Logger, accuracy: ± 0.3 cm) with a temporal resolution of 1 second. Recovery data were then analysed using the approach of Hvorslev (1951):

$$k_{sat} = \frac{\pi r^2}{T_0 c}, \tag{3}$$

where $r$ is the radius of the piezometer, $T_0$ the time needed to recover 37 % of initial water level and $c$ a shape factor. The shape factor depends on the ratio of screen length and radius. The piezometer had a screen length radius ratio of 5.5 and thus we used a shape factor introduced by Chapuis (1989), valid for wells with ratio smaller than 8:

$$c = 4\pi r \sqrt{\frac{L}{2r} + \frac{1}{4}}, \tag{4}$$

where $L$ is the length of the screen.

Grain size distributions from sediment cores

Sediment cores were taken from 30 selected slug test positions (Figure 1b). Sediment cores were taken with a transparent tube with an inner diameter of 32 mm. Length of cores varied between 42 cm and 145cm, with the majority of core lengths between 80 cm and 128 cm. Each core was split into samples according to ~~their sediment profile~~the visible sediment layers. The samples were oven-dried at a temperature of 105 °C and sieved with a vibratory sieve shaker (Retsch AS 200). The sieving setup included the following mesh sizes: 0.063, 0.125, 0.2, 0.3, 0.5, 0.63, 2, 5, 10 mm. Grain sizes smaller than 0.063 mm were classified as silt, grain sizes larger than 0.063 mm but smaller than 0.2 mm as fine sand, larger than 0.2 mm but smaller than 0.63 mm as medium sand, larger than 0.63 mm but smaller than 2 mm as coarse sand larger than 2 mm as gravel.

For the correlation analyses between sediment characteristics and LGD, we used only samples taken from the upper 100 cm of the lake sediment. The mean~~s~~ of each grain size fraction was calculated for each sampling location from all single samples of the upper 100 cm in which the core was split. ~~Additional~~In addition to the correlation analyses, simple and multiple linear regression models were calculated between LGD and each grain size~~s~~ fraction. For the calculation of the multiple linear regression models, correlations between explanatory variables were checked before and variables were regarded to be independent from each other if $\rho$ was below 0.7. Models were regarded as significant if p-values were below 0.05. The goodness-of-fit of the models ~~were~~was ~~determined~~estimated with the coefficient of determination ($R^2$).

### 2.3.2 Topographic indices as controls on large-scale LGD patterns

In order to analyse the effect of far-field conditions on LGD patterns the following topographic indices were calculated using SAGA GIS: average elevation, average slope and the percentage of area with low topographic gradient in direct vicinity to the lake shore. To determine the topographic indices we used a digital elevation model (DEM) of the area with a resolution of 1 m. The topographic indices were estimated for representative areas, i.e. upslope areas for shoreline sections of 100 m length. Therefore the shoreline was split into 46 subsections of 100 m length with an overlap of 50 m. As upslope areas can only be determined for points, not for lines, we calculated upslope areas every meter along the shoreline and aggregated them to one upslope area for each subsection. The upslope areas were determined using the multiple flow direction approach. To investigate the topographical zone of influence (zi), four different extents of the upslope areas were considered: ~~maximum extent of~~ 25, 50,100 and 200 m distance from shoreline. These zones of influence ~~were further~~will in the following be called: $zi_{25m}$, $zi_{50m}$, $zi_{100m}$, $zi_{200m}$. The indices slope and elevation were averaged over each upslope area (arithmetic mean). The percentage of area with low topographic gradient was here defined as the percentage of the upslope area not to be higher than 50 cm above lake level in direct vicinity to the lake shore. This threshold was chosen as this was the area flooded at maximum lake levels known within the last 25 years. Indices were correlated with median LGD rates for the 100 m long subsections derived from VTPs using the Spearman's rank correlation coefficient. Each subsection included 10 VTP measurement locations. Furthermore simple and multiple linear regression models were calculated between LGDs

and topographic indices derived for $zi_{25m}$ and $zi_{50m}$, as in these zones correlation between LGD and far-field conditions were strongest. Correlations between explanatory variables were checked and regarded as independent from each other if ρ was below 0.7. Predictors were regarded as significant if p-values were below 0.05. The goodness-of-fit of the models fits were was determined estimated based on the coefficient of determinationwith the R².

### 2.3.3 Groundwater flow field as control on large-scale LGD patterns

The groundwater flow field was the second far-field variable assumed to affect the LGD patterns. The groundwater flow field is generally assumed to be largely controlled by topography. We used two approaches to test this assumption: the water table ratio (Haitjema and Mitchell-Bruker 2005) and a comparative analysis of flow fields determined based on measured groundwater levels alone (ordinary kriging) or including topographic effects (regression kriging). The simple dimensionless water-table ratio: WTR= (RL²)/(mkHd) with R as annual recharge rate [m/d], L as mean distance between surface waters [m], m=8 or 16 [-] for either 1D or radial flow, k as average hydraulic conductivity [m/d], H as average aquifer thickness [m] and d being the maximum terrain rise [m] (Haitjema and Mitchell-Bruker, 2005, Gleeson et al. 2011) allows a first test of the potential influence of topography on the groundwater flow field, with WTR>1 indicating topography controlled water tables and WTR<1 indicating recharge controlled water tables. The average hydraulic conductivity was determined from 92 undisturbed cores taken during observation well installation and were measured in the lab using a permeameter.

In order to derive the groundwater flow field, measured groundwater heads from 17 16 observation wells located around Lake Hinnensee (Figure 1a) were spatially interpolated. 12 of the 17 16 bore wells were drilled in 2012, five three of them in 2014 and one existed already before installation of the TERENO monitoring network. Groundwater levels were measured regularly every seventh to ninthe weeks since 2012 using an electric contact meter (SEBA Hydrometrie, electric contact meter type KLL, accuracy: ± 1 cm). To determine the groundwater flow field we used groundwater levels measured in 2014, when all wells were completed. In 2014 groundwater levels were generally lower than during the VTP measurement campaigns (2011-2013), but the spatial patterns of groundwater heads of the 12 wells already installed in 2012 remained stable. Measured gGroundwater levels from March 2014 had the smallest differences (mean difference 5.55 cm) to available groundwater data around the time of the VTP measurement campaigns and were thus chosen to derive the groundwater flow field around Lake Hinnensee. For the interpolation of the groundwater measurements, we used both ordinary kriging and regression kriging. In regression kriging a linear regression between an external variable and a the depending variable is included. This allowed us to incorporate the potential effect of topography on the groundwater flow field. In order to minimize the effect of small-scale heterogeneities in the topography the DEM was smoothed by reducing the resolution from 1 m to 10 m and rescaled rescaling to a resolution of 1 m that results of the regression kriging still had a resolution of 1mto maintain a consistent resolution of the results. The groundwater gradients were then calculated from the interpolated groundwater surface.

To analyse the correlation of the groundwater flow field with the LGD patterns, we averaged groundwater gradients in each of the subcatchments for each zone of influence (arithmetic mean) and correlated these with the median LGD rates of the

subsections ~~using the Spearman's rank correlation coefficient~~ as described in *2.3.2*. ~~Additional~~ In addition to the correlation analyses, simple linear regression models were calculated between LGD and groundwater gradients. Furthermore groundwater gradients were also included in multiple linear regression models ~~calculated~~ with topographic indices.

All analyses were carried out in the statistic software R (R Development Core Team, 2011). For the geographical analyses we used the geographical information system SAGA GIS, ~~for some analyses applied within the R environment using~~ and the package "rsaga" (Brenning, 2008).

## 3. Results

### 3.1 Estimating lacustrine groundwater discharge (LGD)

#### 3.1.1 Near-shore LGD ~~estimations~~ derived from vertical temperature profiles (VTP)

At 216 locations along the shoreline of Lake Hinnensee (Figure 1) ~~In a~~ total of 520 VTPs were measured ~~along the shoreline at Lake Hinnensee~~ to analyse a) spatial patterns of near shore LGD, ~~analyse~~ b) the trend of LGD with increasing distance from shore and ~~to analyse~~ c) the temporal stability of LGD patterns. These 520 profiles thus include repeated measurements in time as well as measurements at two distances to shore. At the western lake section, 150 m to 290 m from the northern

tip, ~~at western shore spatial resolution of~~ VTP measurements ~~was reduced to~~ could only be taken every 20 m instead of every 10 m as the lake shore ~~was difficult to access~~ could either not be accessed ~~and~~ or the sediment was unsuitable for measuring due to a thick layer of muddy organic material. However, as lake sediments in this lake section were quite homogeneous, we assume that despite the wider spacing we still captured the spatial variability of LGD. The same reasons also precluded measurements ~~A~~ at 21 ~~additionally~~ 11 other locations around the lake. ~~temperature measurements were not possible as the~~

~~locations were either inaccessible (6 cases) or sediment was unsuitable for measuring, for example due to high stone content or a thick muddy organic material (5 cases).~~ These other 11 locations were irregularly distributed so that gaps were small and we do not expect a strong influence of these gaps on overall spatial patterns. 22 profiles (4 %) were excluded from the analyses as no satisfying fit to the heat transport equation could be achieved. The quality of all remaining LGD estimations was satisfying (~~mean(RMSE) = 0.09 °C,~~ median(RMSE) = 0.06 °C~~, interquartile range (IQR) (RMSE) = 0.12 °C~~, n = 498).

Spatial patterns along the shore line

LGD rates determined from VTPs every 10 m along 2.39 km of the shoreline (216 locations) ranged from -12 L m$^{-2}$ d$^{-1}$ (~~lake water~~ losses) to 169 L m$^{-2}$ d$^{-1}$ (~~groundwater gaining~~ gains) with a median of 44 L m$^{-2}$ d$^{-1}$. ~~The~~ and an interquartile range (IQR) ~~was~~ of 26 L m$^{-2}$ d$^{-1}$. Occurrence of very strong LGD rates of more than 94 L m$^{-2}$ d$^{-1}$ (positive outliners of LGD

distribution), was limited to the northern most 140 m on both the western and the eastern shore of the lake (between "a" and "b" and "f" and "g" in Figure 2) and to one single spot at the western shore 470 m to the south ("i" in Figure 1&2). The

northern most 140 m on both the west and the east shore (between "a" and "b" and "f" and "g" in Figure 1&2) are in the following called "the northern part" and the adjacent region in the south (between "b" and "e" and "g" and "j" in Figure 1&2) will be called "the southern part". Negative rates were only observed at the eastern shore, between 480 m and 530 m from the northern tip ("c" in Figure 1&2). In the northern part of the lake, LGD was stronger and spatially more variable (median = 74 L m$^{-2}$ d$^{-1}$, ~~|ρ|$_{10 m}$ = -0.04, MAD = 21 L m$^{-2}$ d$^{-1}$~~)) than in the southern part (median = 41 L m$^{-2}$ d$^{-1}$, ~~|ρ|$_{10 m}$ = 0.62, MAD = 12 L m$^{-2}$ d$^{-1}$~~) (Figure 2, Figure 3). In the northern part LGD was statistically uncorrelated for all lag distances ~~(Figure 3)~~, while in the southern part it was auto correlated up to a lag distance of 50 m with |ρ| ranging between 0.62 and 0.23 (Figure 3). Autocorrelation in the southern part was stronger on the eastern than on the western shore ~~(Figure 3)~~.

Spatial patterns perpendicular to the shore line

Between 660 m and 1520 m along the eastern shore and 300 m and 830 m south of the northern tip along the western shore, VTPs were measured at 50 cm and 150 cm distance from shoreline to analyse the trend of LGD with increasing distance from shore.

In more than two thirds of all cases (71 %), LGD measured at 150 cm from the shoreline was lower than the rate measured at 50 cm distance (Figure 2). The reduction of LGD rate was on average 20 % (median). However, in 29 % of all cases, LGD increased with distance to the shore (Figure 2) with an average increase of 15 % (median). The patterns of LGD along the shore line measured 50 cm and 150 cm apart from shore were very similar (ρ = 0.81 (p-value < 2×10$^{-16}$), Figure 2).

Temporal stability of spatial patterns

The annual repetitions of LGD rates measured at 43 VTP positions (Figure 1) highlight that the observed LGD patterns were correlated between the individual measurement campaigns (Figure 4). ~~Correlation~~ The correlation coefficient was 0.71 (p-value = 5×10$^{-6}$) between summer 2011 and 2012 (n = 34), 0.82 (p-value = 10$^{-3}$) between 2012 and summer 2013 (n = 13), 0.70 (p-value < 4×10$^{-6}$) between 2012 and winter 2013 (n = 34), and 0.66 (p-value < 3×10$^{-5}$) between 2011 ~~with~~ and winter 2013 (n = 33). The ~~RMSE~~ differences between LGD rates measured in different years ~~was~~ were lowest comparing rates from summer 2011 and summer 2012 (median = ~~23~~ 6 L m$^{-2}$ d$^{-1}$) and strongest comparing rates from summer 2011 and winter 2013 (median = ~~51~~ 27 L m$^{-2}$ d$^{-1}$).

Single sediment temperature measurements as a qualitative indicator for LGD spatial patterns

Sediment temperatures from the top of the sediment down to a depth of 10 cm were not well correlated with LGD rates, but strong correlations were found between LGD rates and sediment temperatures measured 20 cm below surface and deeper (correlation coefficients range between 0.46 and 0.96). While the correlation is generally high, the slope of the regression line varies in time (Figure 5).

### 3.21.1 2 Lake sediment temperature anomalies as indicators for LGD based on fibre optic distributed temperature sensing (FO-DTS)

We measured lake sediment temperature with a FO-DTS cable installed in 6 transects through the northern part of the lake (Figure 1c) during two measurement campaigns (20.02.2014, 27.08.2014) to identify offshore groundwater inflow. The complementary results of the radon measurement campaigns are described in the appendix.

The winter and summer measurements consisted of 15 measurements in 2-minute intervals. The repetitions resulted in very similar measurements (median range of temperature differences among repetitions in winter: 0.19 °C; maximum range in winter: 0.44 °C, median range in summer: 0.22 °C; maximum range in summer: 0.38 °C) (Figure 5a6a). In winter sediment temperature ranged between 3.4 °C and 5.3 °C and in summer between 17.0 °C and 18.4 °C (Figure 56). In summer, groundwater was 7 °C cooler than lake water and in winter 3 °C warmer than the lake. The air temperature, groundwater temperatures and temperature depth profiles of the lake measured during the DTS measurements are presented in Table 2.

The spatial patterns of sediment temperature anomalies, i.e. the shifts of sediment temperatures towards groundwater temperatures, were similar in both campaigns. Strongest deviations from sediment temperatures towards the groundwater temperatures (positive in winter and negative in summer) were located near the shoreline at corners two and three (Figure 56). A slight shift towards groundwater temperatures was also observed near corner 5, but only along the DTS cable north of the corner. These hot and cold spots in winter and summer respectively were not located nearest to the shoreline, but between 2 m and 14 m offshore, where lake depth steeply increased (Figure 5b6b).

LGD rates estimated from VTPs measured along the fibre optic cable ranged between 109 L m$^{-2}$ d$^{-1}$ and 238 L m$^{-2}$ d$^{-1}$ at corner two and between 60 L m$^{-2}$ d$^{-1}$ and 169 L m$^{-2}$ d$^{-1}$ at corner three, which correlated with the intensities of the sediment temperature anomalies at these locations (Figure 6).

### 3.2.2 Identification of offshore LGD based on radon concentrations

To test the capability of radon as a tracer of groundwater inflow at Lake Hinnensee, we took two samples at VTP measurement locations where LGD rates were known to be significantly different. The radon activity at the sampling location with stronger LGD was significantly higher (787 Bq m$^{-3}$) than the radon activity in the other sample (90 Bq m$^{-3}$). The groundwater had a radon activity of 12151 Bq m$^{-3}$. Radon activities in lake water samples ranged between 0 and 103 Bq m$^{-3}$, with a median of 41 Bq m$^{-3}$ and an IQR of 30 Bq m$^{-3}$. Radon activities in the uppermost quartile, (> 54 Bq m$^{-3}$) were measured in the northern part of the lake and at one sampling point in the central part of the lake 100 m east of the southern temperature logger chain. The three southernmost points had lowest radon activities (0 – 22 Bq m$^{-3}$).

### 3.~~3~~ 2 Identifying controls of LGD patterns~~Controls of LGD patterns~~

#### 3.~~3~~2.1 Sediment heterogeneity as a small~~-~~-scale control

Hydraulic conductivity from slug tests

~~Hydraulic conductivity~~

The $k_{sat}$ values estimated from the slugtests ranged between $2.03\times10^{-6}$ m s$^{-1}$ and $4.25\times10^{-5}$ m s$^{-1}$ with an IQR of $1.41\times10^{-5}$ m s$^{-1}$ (Figure 7). Points with $k_{sat}$ values lower than the 25 % quartile ($9.20\times10^{-6}$ m s$^{-1}$) were mainly located at the western shoreline, while points with values higher than the 75 % quartile ($2.33\times10^{-5}$ m s$^{-1}$) were located on the eastern shore. Instead of a positive correlation between $k_{sat}$ values and LGD rates, there was a slight negative, but statis~~pas~~tically insignificant (p-value = 0.05), correlation of -0.36 (Figure 7).

Grain size distributions from sediment cores

The sediment samples taken from 30 VTP measurement locations were dominated by sand with a small fraction of gravel and silt. The median percentages of sand, gravel and silt were 92.3 %, 6.8 % and 0.6 %, respectively. Within the sand fraction, medium sand dominated (median = 59.2 %), followed by fine sand (median = 23.3 %) and coarse sand (median = 13.7 %). No consistent layering or trends of grain sizes with depth could be identified. Only the fraction of medium sand decreased slightly with increasing sampling depth ~~with~~ by 2 % every 10 cm, but also here the correlation was weak ($\rho = 0.3$, p-value = $6\times10^{-4}$). ~~Along-shore textural patterns were more pronounced with the highest percentage of gravel at the western shore of the northern part. Here the percentage of gravel ranged between 20 % and 80 %, at all remaining locations percentage of gravel was below 10 %. The percentage of coarse sand was highest, between 15 % and 28 %, in samples from the northern part of the lake, but was below 10 % at the sampling locations from the southern part. Fine sand, in contrast, dominated the grainsize distribution in the south.~~

Relating grain size distributions~~,~~ averaged over the upper meter of the lake sediment~~,~~ to the strength of LGD showed that low LGD rates occurred at locations dominated by fine sand and stronger LGD rates occurred at locations with higher fraction of larger grain sizes (Figure 8a). LGD was positively correlated with the percentage of gravel and coarse sand~~, but~~ and negatively correlated with the percentage of fine sand (Table 3, Figure 8 b-c), but LGD rates were uncorrelated with the grain size fractions medium sand and silt (Table 3). LGD varied by a factor of three across these grain size fractions. ~~The linear regression models describing these correlations can be found in Table 3~~

~~One~~ For the multiple regression model considering all grain sizes ~~was calculated.~~ ~~O~~only coarse sand and fine sand were significant variables (p-values < 0.05). The model had a~~n~~ R² of 0.54 (Table 3). The absolute residuals were on average 21 L m$^{-2}$ d$^{-1}$, with largest positive residuals (observed < calculated) at a distance of 50 m and 10 m from the northern tip at the eastern and western shore, respectively and largest positive residuals (observed > calculated) at a distance of 70 m and 90 m from the northern tip at the eastern shore (Figure 9a).

### 3.~~3~~2.2 ~~Large scale controls on LGD patterns~~Topographic indices as controls on large-scale LGD patterns

~~Topographic indices~~

Subcatchments derived from the 46 shoreline sections differed significantly in size. While subcatchments in the flatter areas of the south were larger, elevations were higher and slopes generally steeper in the north. There was no clear correlation between LGD rates and the size of the subcatchment in each topographical zone~~s~~ of influence (Table 4). Percentages of area with low topographic gradient in direct vicinity to the lake shore area were also not correlated with LGD rates, except for $zi_{25m}$, where a weak negative correlation was found (Table 4). The correlation between LGD rates and the indices elevation and slope were both positive (Table 4, Figure 10 right) and the strength of correlation was strongest ~~considering a~~for $zi_{50m}$ and decreased with increasing zone of influence (Table 4).

### 3.2.3 Groundwater flow field as control on large-scale LGD patterns

The water table ratio (Haitjema and Mitchell-Bruker, 2005) as an indicator for either recharge or topography controlled groundwater tables was determined based on conservative estimates of the input variables: annual recharge rate R=0.00351 m/d (based on values from Müller et al. (2009) determined in the same region), mean distance to the next surface waters L being approximately 2000 m, m=8 [-] for 1D flow (however, changing this to 16 for radial flow does not change the outcome), the average hydraulic conductivity k=7.776 m/d (based on laboratory analyses of undisturbed cores obtained during observation well installation), average aquifer thickness H=15 m (from bore-logs) and the maximum terrain rise d=52 m. The water table ratio in this case amounts to 0.029 which is << 1 and thus indicates that water tables in the study area are not topography- but instead recharge controlled.

The interpolated groundwater table based on the water tables in the observation wells showed groundwater flow towards Lake Hinnensee from all directions (Figure 10). In general, groundwater gradients were stronger in the north than in the south. ~~Differences between the ordinary and regression kriging were strongest where the topography is most pronounced. The linear predictor function used for the regression kriging was $gw = 62.5 + 0.02e$ where $gw$ is the groundwater level in meter and $e$ the smoothed surface elevation in meter. The topography was a significant model predictor with a p-value of $10^{-5}$.~~

The maximum deviation between interpolated and measured groundwater levels was below 1 cm for both interpolations, but in comparison to the ordinary kriging, the regression kriging (which included topographical information) resulted in significantly stronger gradients: median groundwater gradient in the $zi_{200m}$, for example, was 0.24 cm m$^{-1}$ using regression kriging and 0.09 cm m$^{-1}$ using ordinary kriging.

While the correlation between LGD and groundwater gradients derived from ordinary kriging was weak with correlation coefficients ranging between 0.28 and 0.37 (Table 2), stronger correlation was found for LGD and gradients derived from regression kriging (between 0.55 and 0.64) (Table 2). The linear predictor function used for the regression kriging was

$gw = 62.5 + 0.02e$ where $gw$ is the groundwater level in meter and $e$ the smoothed surface elevation in meter. The topography was a significant model predictor with a p-value of $10^{-5}$.

### 3.2.4 Linear regression models between LGD and far-field predictors

The linear regression models describing the correlation between LGD and far-field conditions were estimated for all predictors of $zi_{25m}$ and $zi_{50m}$. Significant predictors of LGD large-scale patterns were elevation, slope, percentage of area with low topographic gradient, but (only in $zi_{25m}$) and groundwater gradients (Table 5). No significant linear regression model was found with the potential predictors size of subcatchment and percentage of area with low topographic gradient in $zi_{50m}$. The R² of all models were not larger than 0.37 (Table 5), but were below 0.12 for the predictors "percentage of area with low topographic gradient" and "groundwater gradients derived from ordinary kriging" below 0.12 (Table 5) and. Therefore these predictors were therefore not regarded excluded for further analyses.

The calculation of multiple regression models revealed no significant relations between LDG and topographic indices in both zones of influence. Reducing stepwise the most insignificant predictor until all predictors became significant resulted in the single linear regression models as presented above and in Table 5.

Calculated large-scale LGD patterns along the shoreline using the best linear regression model we found (considers based on groundwater gradients derived from regression kriging in $zi_{25m}$, Table 5), are shown in Figure 9b. The general spatial pattern was captured by the model, but absolute deviations were on average 10.4 L m$^{-2}$ d$^{-1}$. Strongest overestimations occurred at the distances 500 m at the eastern and 300 m at the western shore and strongest underestimation by the model were found at the distances 450 m and 150 m at the western shore.

## 4. Discussion

### 4.1 LGD patterns along the shoreline and potential controls

The here employed experimental design based on extensive field campaigns provided exceptionally detailed information on both small-scale variabilities and large-scale patterns of LGD rates along a lake shore line. This data set thus bridges the gap between the detailed local and low resolution regional larger scale investigations of previous LGD studies.

While the employed method of measuring VTPs with a needle-thermistor is sufficiently rapid to make this large high resolution data set possible it nevertheless takes considerable effort. We therefore investigated if measuring temperature at just a single depth instead of measuring an entire depth profile (thus reducing measurement time even further) would already supply at least qualitative information on LGD patterns. We found that temperatures measured at depths larger than 20cm generally correlated well with LGD rates, thus reproducing the general pattern of groundwater inflow. However, to convert these temperatures to LGD rates it would be necessary to measure at least some complete profiles for a large enough range of LGD rates to obtain a decent calibration. As sediment temperatures and their gradients change over time this calibration needs to be repeated at each survey date.

### 4.1.1 Large-scale patterns

As expected, ~~T~~the observed large-scale patterns of LGD at Lake Hinnensee correlated with mean groundwater gradients. Interestingly, even though the system classifies as recharge controlled and not topography controlled according to the water table ratio (Haitjema and Mitchell-Bruker, 2005, Gleeson et al. 2011), correlation was stronger between LGD and groundwater gradients derived from regression kriging (which includes topographic information) than between LGD and groundwater gradients derived from ordinary kriging. This suggests that the groundwater surface was more realistically interpolated using regression kriging and indicates at least some influence of topography on groundwater movement, which is consistent with the theory of topography-controlled groundwater flow described by Toth (1963), The predictors based on surface slopes and surface elevation, so purely topographical information, also correlated with LGD rates (with a correlation coefficient of 0.6), another indication of at least some topographic control on the groundwater flow field. However, even the best regression model (based on the groundwater table gradients from regression kriging) did not capture all of the observed variability in LGD, thus suggesting the existence of additional controls. ~~could only partially be explained by predictors derived from topography or the groundwater flow field. However, correlation between LGD and surface elevation, slope or groundwater gradients derived from regression kriging indicate a linkage between topography, groundwater movement and patterns of LGD. A similar link had previously been observed in groundwater river interactions (Caruso 2016). As correlation was stronger between LGD and groundwater gradients derived from regression kriging than between LGD and groundwater gradients derived from ordinary kriging, we assume that the groundwater surface was more realistically interpolated using regression kriging. This indicates again a link between topography and groundwater movement, which is in consistent with the theory of topography-controlled groundwater flow introduced by Toth (1963).~~

In contrast to observations of streamflow generation in mountain catchments (Jencso et al., 2009, 2010), no positive correlation was found between the size of the subcatchment and LGD rates (Table 4) indicating that surface catchments derived for lake subsections were not a good estimator for the amount of subsurface water flowing into the corresponding lake sections. We assume our result indicates that subsurface catchments differ from surface catchments which is not unusual in low land areas. The weak negative correlation between LGD and the topographic index "percentage of area with low topographic gradient in direct vicinity to the lake shore" in $zi_{25m}$ indicated that low topographic gradients at the shoreline could buffer groundwater flow towards the lake.

Even though the interpolated groundwater surface showed groundwater flow towards Lake Hinnensee from all directions (Figure 10), we measured negative LGD rates at one small subsection of the lake (Figure 2). Reasons for this flow reversal are unclear. However, the neighbouring stretches of shoreline~~locations next to the locations of interest~~ were characterized by very low LGD rates (Figure 2), ~~while~~even though $k_{sat}$ values at this section were comparab~~le~~y high (Figure 7) and thus we assume that very low hydraulic gradients are the cause for the low LGD rates. While transpiration is likely to cause diurnal fluctuations in groundwater levels all around the lake, it can result in a temporary local inversion of the groundwater – lake

gradients at locations where these gradients are very low (Winter et al., 1998). This could be a potential explanation for the negative LGD rates measured at this location.

### 4.1.2 Small-scale patterns

Our measurements revealed strong small-scale spatial variability in LGD along the shoreline (Figure 2). Absolute amounts and spatial variability of LGD were within the range of previous studies (Rosenberry et al., 2015; Blume et al.; 2013; Neumann et al., 2013). Measuring VTPs with a high measurement resolution along large parts of the shoreline and analysing LGD values with autocorrelation plots highlighted furthermore that the strength of small-scale variability also varies along the shoreline. This type of information is likely to be overlooked in studies focusing either on entire lake systems but using a low spatial resolution (Schneider et al., 2005; Meinikmann et al., 2013; Shaw et al., 2013) or using a high spatial resolution but only on a very local scale (Kishel & Gerla, 2002; Blume et al.; 2013; Sebok et al., 2013). The repetitions of VTP measurements revealed that the observed patterns were stable in time and are thus likely controlled by static characteristics. Differences in LGD rates measured in different years ~~might result from climate driven processes.~~are likely the result of annual differences in groundwater recharge and thus gradients of the flow field.

Surprisingly no positive correlation was found between LGD rates and $k_{sat}$ values derived from slugtest at Lake Hinnensee. The relationship between $k_{sat}$ and LGD could be confounded as a result of strong differences in hydraulic gradients. However, even when only adjacent measurement locations with similar gradients were taken into account no clear positive correlations appeared (Figure 7). Slug test were found to be the most accurate method to determine $k_{sat}$ values of sandy stream beds (Landon et al., 2001), but estimation of hydraulic conductivity is always ~~characterised by~~subject to high uncertainties (Landon et al., 2001; Kalbus et al., 2006). Even though the slug tests were carried out carefully, we cannot exclude that pore structure~~,~~ was altered during the piezometer installation and thus the $k_{sat}$ values of lake sediments were changed. In contrast to $k_{sat}$ values, LGD rates clearly correlated with both the finest and the coarsest grain size fractions. Grain sizes give no direct information ~~of the~~on hydraulic conductivity, but coarse sand and gravel is associated with higher hydraulic conductivity values than well sorted fine sand (Bear, 1972). As LGD rates correlated positively with percentages of gravel and coarse sand and negatively with the percentage of fine sand, this corroborates the assumption that sediment heterogeneities do at least partially control small scale variability in LGD. We furthermore see that a variation of LGD of up to a factor of three can be due to grain size variability alone.

### 4.2 LGD patterns with increasing distance to shore and ~~its~~their potential controls

To identify LGD patterns with increasing distance to shore, we analysed VTP profiles measured in 50 cm and 150 cm distance from the shoreline in the southern part. Results showed a prevailing decrease of LGD with distance to shore (median decrease of 20%). This observation corresponds to the theoretical pattern of LGD found by McBride and Pfannkuch (1975) and other experimental studies (Brock et al., 1982; Cherkauer & Nader 1989, Kishel & Gerla 2002; Blume et al.; 2013). The study from Blume et al. (2013), conducted at small shore line section of 20 m length and 4 m width in the

northern part of Lake Hinnensee, indicated that the strongest decrease of LGD occurred in the first 1.5 m distance from shore. However, ~~in 29 % of the locations,~~ LGD increased with increasing distance from shore in 29 % of the locations. The locations with anomalously increasing LGDs did not show any obvious anomalies with respect to local bathymetry, density of vegetation or organic top-layers, which could have been used to explain the observed patterns. The hypothesis that

differences in sediment characteristics cause these anomalies could unfortunately not be tested as no information on sediment characteristics is available for distances from shore larger than 50cm.

### 4.3 Offshore-LGD patterns and potential controls

We investigated the presence and absence of offshore LGD with two ~~methods~~natural tracers: radon and heat, ~~(sediment temperature measurements with FO-DTS) as natural tracers~~and the two methods lead to similar results (for the radon results and discussion see appendix).

~~Measured radon activities of lake samples indicate groundwater inflow, but activities were low. Sampling locations were mainly located in the epilimnion, where lake levels were too shallow to allow for a thermocline. In the epilimnion, the lake water~~radon activity ~~is assumed to be well mixed (Kluge et al., 2012). Low radon concentration sampled close to the lake bottom could be the result from radon outgassing to the atmosphere and advective and diffusive radon fluxes from lake bottom towards the lake atmosphere exchange zone. But comparison of radon activities~~

~~at locations with known groundwater inflow in the near shore part showed a significant effect of local groundwater inflow on radon activities without considering wind-driven radon losses to the atmosphere. As well, Kluge et al. 2007 who measured several vertical profiles in a lake related increasing radon activities at the bottom of the lake to local groundwater inflow. Thus we assume that local groundwater inflow in deeper parts would as well enhance locally radon concentrations at the sediment lake interface. Therefore it is possible that the homogeneous and very low~~As radon activities could be assumed to~~

be well mixed in the epilimnion, the observed slight enhanced~~ radon activity in the deeper parts of the ~~northern part of the lake in comparison to the southern part might be the result from stronger groundwater inflow from the near-shore area instead from the deep sampling locations itself. As also~~ The FO-DTS measurements showed no shifts in sediment temperatures towards the groundwater temperature in the flatter and deeper parts of the lake (Figure ~~5b~~6b), and we assume that groundwater inflow is insignificant here. Low radon activities measured offshore at the lake bottom across the entire

lake also support this assumption (see appendix). Insignificant groundwater inflow in the flatter and deeper part of the lake is in correspondence with the theory of exponentially decreasing groundwater inflow with distance from the shore introduced by McBride & Pfannkuch (1975). Furthermore we know from observations by divers that the lake bottom was covered with fine-grained organic sediments, which typically accumulates in the deep, flat parts, where wave action cannot re-suspend the fine sediments (Rosenberry et al., 2015). A layer of fine-grained sediment could significantly decrease hydraulic

conductivity of the lake bed sediment and can even totally prevent groundwater lake exchange (Kidmose et al., 2013).

~~The radon activity of the sample taken few meters apart from the shoreline close to corner 3 (Figure 5b) had the strongest radon activity of the lake water samples, and also FO-DTS measurement showed a clear groundwater signal at this location. FO-DTS showed local hotspots of groundwater inflow at three locations, all on steep steps between the near shore part and~~

the flatter central basin. We only had detailed information of sediment temperatures at the steep steps close to corner 1, 2, 3 and 5. At the other corners fallen trees made it impossible to lay out the FO-DTS cable closer to the shoreline. However, FO-DTS showed local hotspots of groundwater inflow at three locations, all on steep steps between the near shore part and the flatter central basin. These locations with sediment temperature anomalies coincided with high near shore LGD rates

(estimated with VTPs), while corners where no temperature anomalies were found, coincided with low near shore LGD rates (Figure 5b6b). As hotspots in near shore LGD occurred locally and mainly in the northern part of the lake, we assume that the occurrence of hotspots of LGD at steep steps is also a local phenomenon limited to the northern part. As the occurrence of the local sediment temperature anomalies is temporally stable (Figure 5a6a), we assume that static characteristics are responsible – at this locations, where near shore LGD was already strong, the morphology of steep steps might force a local

offshore increase of LGD again: at steep slopes fine sediment is prone to be re-disturbed by turbidity current activities (Håkanson, 1977), locally increasing hydraulic conductivity and thus also LGD.

## 4.4 Prediction of LGD patterns

Using linear regression models based on topographic characteristics or the groundwater flow field to predict large-scale patterns of LGD at Lake Hinnensee roughly reproduced the observed patterns, but locally strongly over- or underestimated

the observed LGD rates (Figure 9b). However, regression models considering sediment heterogeneities were able to explain more than 50 % of the observed small-scale variability in LGD ($R^2$ = 0.55). We calculated linear regression models separately for topographic indices and sediment heterogeneities, because sediment cores were only taken from a fraction of the lake covering not more than one lake subsection used for far-field analysis. But as LGD is driven by both, the hydraulic gradients between lake and aquifer and by the hydraulic conductivity, combining the information would likely explain more

of the observed variability. Another possible explanatory variable is heterogeneityinfluence are currently unknown local heterogeneous features within the adjoining aquifer (Winter 1999; Cherkauer & Nader, 1989; Fleckenstein et al., 2009). Based on the analysis of cores from the bore holes of the observation wells and also on geophysical measurements around Lake Hinnensee, the presence of till lenses in the aquifer seems likely (H. Wilke, pers. comm. 2016). SSuch small-scale structures, such as these may influence the groundwater flow paths and cause variability in LGD.

## 5. Summary and Conclusion

As LGD couldan significantly contribute to a lake water budgets and could furthermore significantly influence lake water quality by transporting large loads of nutrients or contaminants. Thus, quantifying LGD rates and determining LGD patterns iscan be essential for a sustainable lake management (Meinikmann et al., 2013; Lewandowski et al., 2015). While LGD is

known to be strongly spatialspatially variable, across the entire aquifer lake interface it is also not easily measured, especially not with an extent and spatial resolution that allows for the characterisation of LGD patterns. Furthermore,

~~structures controlling~~ causes and controls of these patterns are not well understood. Our aim was the characterization of LGD patterns at Lake Hinnensee based on ~~intensive experimental investigations~~a unique high resolution data set that extends along most of the shoreline~~.~~, ~~As experimental investigations of this calibre are rarely feasible, we furthermore wanted~~ and to use this data set to test common assumptions. Identifying ~~~~external (and easily measureable) controls as a means for pattern prediction would greatly reduce experimental effort. ~~identify the controls behind these patterns, in the hope of pattern predictability via external (and easily measureable) controls.~~ By using VTPs in the near shore area and FO-DTS measurements and radon sampling in the off-shore area, we identified the following pattern in LGD for Lake Hinnensee: LGD was concentrated in the near shore area and generally decreased with distance to shore; some local hotspots of LGD were identified in locations of steep steps towards the lake bottom; overall, offshore-LGD was insignificant ~~in the central part of the lake~~. LGD was generally stronger and more variable in the northern part than in the southern part of the lake. Repetitions of LGD measurements indicated that the observed patterns in LGD remained stable in time. As the hydrogeology of the catchment is sufficiently homogeneous to avoid patterns being dominated by geological discontinuities, we were able to test the common assumptions that spatial patterns of LGD are controlled by sediment characteristics and the groundwater flow field (here interpolated from observation wells) and the potential of topographic indices as proxies for gradients of the groundwater flow field. We identified the following links between LGD patterns and external factors at Lake Hinnensee: Even though classified not as topography-, but as recharge controlled based on the water table ratio, ~~L~~large-–scale LGD patterns ~~at Lake Hinnensee~~ were linked to the local topographic gradient and also to groundwater gradients derived from regression kriging, which also included topographic information. The explanatory power of these indices was strongest when derived locally up to a distance of 50 m from shoreline, ~~but~~and decreased with increasing distance from the lake. Small-scale LGD patterns in the north were linked to sediment heterogeneities: LGD patterns correlated positively with percentages of gravel and coarse sand and negatively with the percentage of fine sand. However, LGD patterns did not correlate with $k_{sat}$ values derived from slug tests. We assume that our findings are transferrable to similar lowland landscapes with quasi-homogeneous aquifers. As the water table ratio at this site indicated recharge control, we assume topography to have an even greater influence on LGD patterns in areas where groundwater tables classify as topography controlled. However, more complex hydrogeological settings which include discontinuities can override and mask the topographic signal.

Our results furthermore showed that predictions of LGD rates using regression models derived from correlation ~~of~~with external controls ~~and LGD~~ were associated with high uncertainties, but nevertheless allowed a rough estimation of LGD patterns. Topographic indices, such as elevation or slope, are often readily available. Analysing grain size distributions of lake sediment is labour intensive, but sediment cores taken with transparent ~~cores~~sampling tubes can be easily analysed at least qualitatively. This information combined with information of topographic gradients can then be used to develop an effective and efficient measurement design for more a detailed characterisation of spatial patterns of LGD that goes beyond the rough estimate that the linear regression models can provide~~.~~, ~~whereby information of topographic gradients could help to get information on large scale patterns and sediment cores could be used as indicator for small scale patterns.~~.

~~However, our experience m~~Measuring VTPs with a 45 cm long~~, but a small~~ needle-thermometer ~~of 3 mm, present~~is a fast and inexpensive method to ~~measure~~determine LGD rates without disturbing the sediment. Correlation between LGD rates and temperature values measured in ~~2~~30 cm depth also show~~,~~ that even single depth temperature measurements ~~could already help to identify~~can provide at least some rough qualitative information on LGD patterns. While installing a fiber optic cable along the shore line would have the advantage of providing a large high resolution spatial data set as well as continuous measurements, the cable would need to be installed at a fixed depth > 20cm to provide reliable data, uninfluenced by solar radiation and boundary effects. Such an installation of a kilometre-long cable at a fixed depth is challenging and would significantly disturb the sediment, potentially causing preferential flow paths and changing the very fluxes we want to measure. We therefore favour the manual measurements using the thermistor-needle, as this method has furthermore the added advantage of providing temperature profiles and thus enabling us to obtain reliable quantitative estimates of LGD rates at a large number of locations. While seepage meters provide flux data directly they are less suitable for high numbers of measurements. Piezometers provide vertical hydraulic gradients, however, for the determination of LGD rates from this data, the necessary information on saturated hydraulic conductivities is often subject to high uncertainties.

From the experience gained in this study we would suggest the following protocol for future studies of LGD patterns and controls:

1) Determine topographic indices for the 50m region around the lake (broken down in partially overlapping subsections of a length representative of the variability in topography – in our study 100m with 50 m overlap) and combine this with groundwater flow field information from observation wells. Determine if system classifies as groundwater or topography controlled by using the water table ratio, if only limited well information exists, to get a better idea of the ground water table characteristics. 2) Predict LGD patterns based on this information. 3) Test predictions at 8 locations (more if feasible) by measuring VTPs and estimating LGD rates based on the heat transport equation. If possible perform additional single depth temperature measurements and compare the obtained qualitative LGD patterns with the ones predicted from topographic indices to further evaluate the reliability of the prediction. 4) Characterise small-scale variability at 2 of these locations (covering high and low inflow regions) by additional measurements at higher spatial resolution. 5) Use clear plastic tubes to sample sediment cores for quick visual inspection and rough classification according to grain size/permeability and relate this to the corresponding LGD at each of these locations. 6) If interested in the temporal stability of the LGD patterns, repeat step 3 at different points in time.

**Acknowledgments**

This study is a contribution to the Virtual Institute of Integrated Climate and Landscape Evolution Analysis –ICLEA– of the Helmholtz Association. We would like to thank Lisei Köhn, Christian Budach, Sigfried Tusche, Philip Müller, Markus Morgner, Henriette Wilke and Knut Günther for their help in the field, the divers Silke Oldorff and Frank Kroll for their help laying out the FO-DTS cable and the Mueritz National Park authorities for their cooperation. Furthermore we thank Matthias

Munz and Jana Carus for helpful discussions in the early stage of the manuscript. Bathymetry data were provided by the Ministerium für Landwirtschaft, Umwelt und Verbraucherschutz Mecklenburg-Vorpommern. We also thank two anonymous referees, whose suggestions improved the manuscript considerably. The study took place in the TERENO observatory of north-east Germany funded by the Helmholtz Association.

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

**Table 6: Dates, ~~information~~ boundary conditions and use of vertical temperature profile surveys.**

| Date | Groundwater temperature [°C] | Lake water temperature [°C] | Data analysis |
|---|---|---|---|
| 24–25 August 2011 | 10.7 | 22.7 | Temporal stability of LGD patterns |
| 12–14 June 2012 | 8.9 | 18.2 | Spatial patterns of LGD along the shoreline |
| 16–17 July 2012 | 10.1 | 20.1 | Spatial patterns of LGD along the shoreline |
| 21–23 January 2013 | 7.0 | 0.0 | Spatial patterns of LGD along the shoreline |
| 17–25 July 2013 | 10.1-10.3 | 23.2 | Spatial patterns of LGD along the shoreline and LGD patterns with increasing distance from shore |

**Table 7: Groundwater temperatures and lake water temperatures depth profile measured during FO-DTS campaigns.** [1] **measured at the nearby weather station,** [2] **measured close to the lake**

| Date | Air temperature [°C] | Groundwater temperature [°C] | Temperature depth profile [°C] |
|---|---|---|---|
| 20 February 2014 | 6.8[1] | 6.3 | 0m: 1.8<br>-1m: 3.4<br>-2m: 3.4<br>-3m: 3.4<br>-4m: 3.4<br>-5m: 3.4 |
| 27 August 2014 | 19.4[1]<br>16.1[2] | 11.6 | 0m: 18.9<br>-1m: 18.7<br>-2m: 18.7<br>-3m: 18.5<br>-4m: 18.4<br>-5m: 18.4 |

**Table 8: Correlation coefficients (ρ), linear models describing the correlation between LGD and predictors and the coefficient of determination (R²). ρ coloured in black indicate significant correlation (p-value < 0.05), light red indicate insignificant correlations (p-value > 0.05)**

| Predictor (x) | ρ | Models (LGD = ...) | R² |
|---|---|---|---|
| gravel | 0.61 | 49.84 + 1.78x | 0.25 |
| coarse sand | 0.62 | 14.03 + 5.27x | 0.46 |
| medium sand | 0.01 | - | - |
| fine sand | -0.7 | 127 - 2.12x | 0.48 |
| Silt | 0.01 | - | - |
| Coarse + fine sand | | $72.72 + 3.06x_1 - 1.35x_2$ | 0.54 |

**Table 9: Correlation coefficients (ρ) between LGD and far-field predictors calculated for upslope areas in certain topographical zones of influence (zi). ρ coloured in light red indicate insignificant coefficients (p-value > 0.05). ρ coloured in grey to black in dependence of the strength of correlation indicate significant coefficients (p-value < 0.05). gg is the abbreviation for groundwater gradients and ltg the abbreviation for low topographic gradient in direct vicinity to the lake shore.**

| $z_i$/Predictors | Size | Mean elevation | Mean slope | Percentage of ltg | Mean gg -ordinary kriging- | Mean gg -regression kriging- |
|---|---|---|---|---|---|---|
| 25 m | 0.15 | 0.61 | 0.58 | -0.44 | 0.33 | 0.64 |
| 50 m | 0.03 | 0.62 | 0.64 | -0.30 | 0.36 | 0.65 |
| 100 m | -0.19 | 0.45 | 0.58 | 0.00 | 0.32 | 0.59 |
| 200 m | -0.31 | 0.23 | 0.54 | -0.02 | 0.33 | 0.55 |

**Table 10: Linear regression models describing the correlation between LGD and far-field predictors and the coefficient of determination ($R^2$). gg is the abbreviation for groundwater gradients and ltg the abbreviation for low topographic gradient in direct vicinity to the lake shore.**

| Predictor (x) | $zi_{25m}$ Models (LGD = ...) | $R^2$ | $zi_{50m}$ Models (LGD = ...) | $R^2$ |
|---|---|---|---|---|
| Elevation | -591.62 + 9.65x | 0.35 | -273.14 + 4.65 | 0.34 |
| Slope | 21.66 + 2.04x | 0.33 | 20.74 + 2.2x | 0.32 |
| Percentage of ltg | 52.61 - 1.29x | 0.11 | - | - |
| gg -ordinary kriging- | 30.12 + 141.11x | 0.09 | 27.18 + 167.90x | 0.12 |
| gg -regression kriging- | 21.16 + 62.30x | 0.37 | 21.08 + 64.36x | 0.36 |

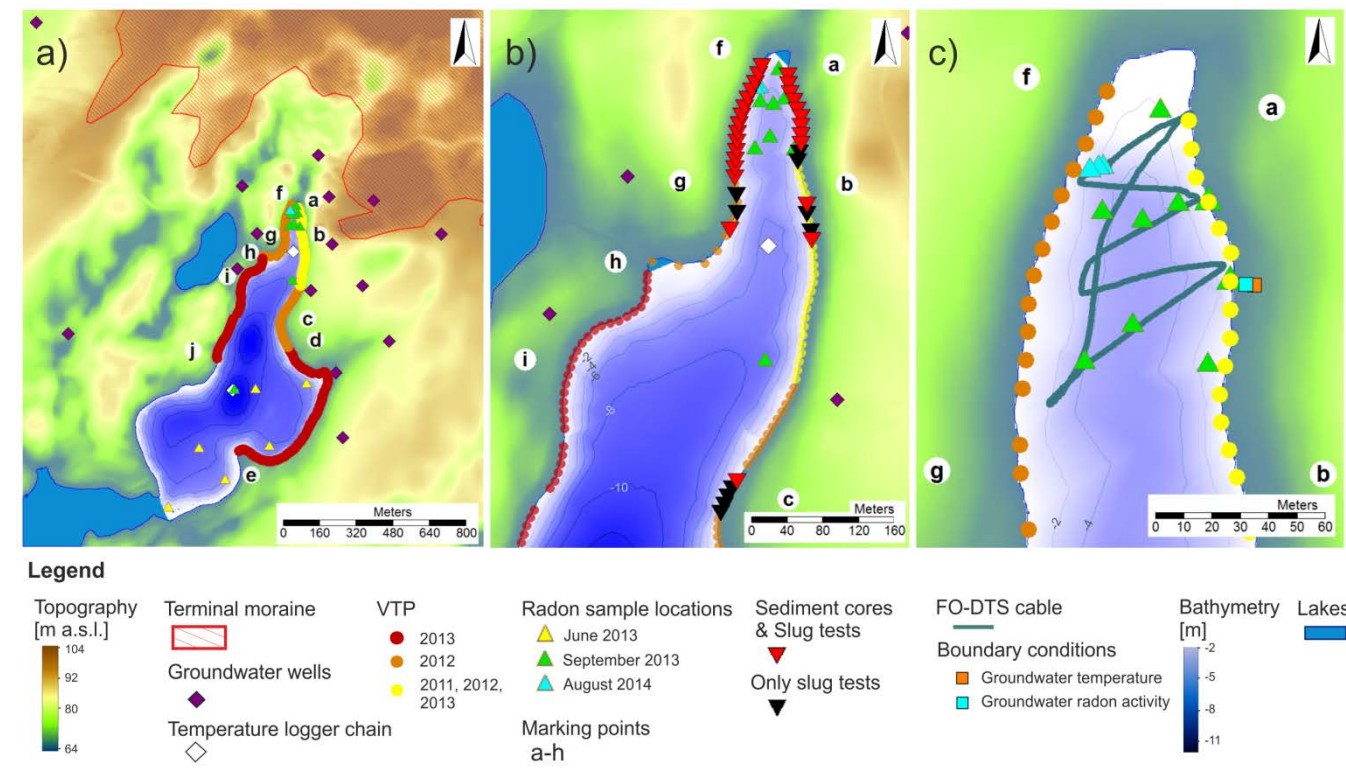

**Legend**

Topography [m a.s.l.]
104
92
80

Terminal moraine

Groundwater wells

Temperature logger chain

VTP
2013
2012
2011, 2012, 2013

Radon sample locations
June 2013
September 2013
August 2014

Marking points
a–h

Sediment cores & Slug tests

Only slug tests

FO-DTS cable

Boundary conditions
Groundwater temperature
Groundwater radon activity

Bathymetry [m]
-2
-5
-8
-11

Lakes

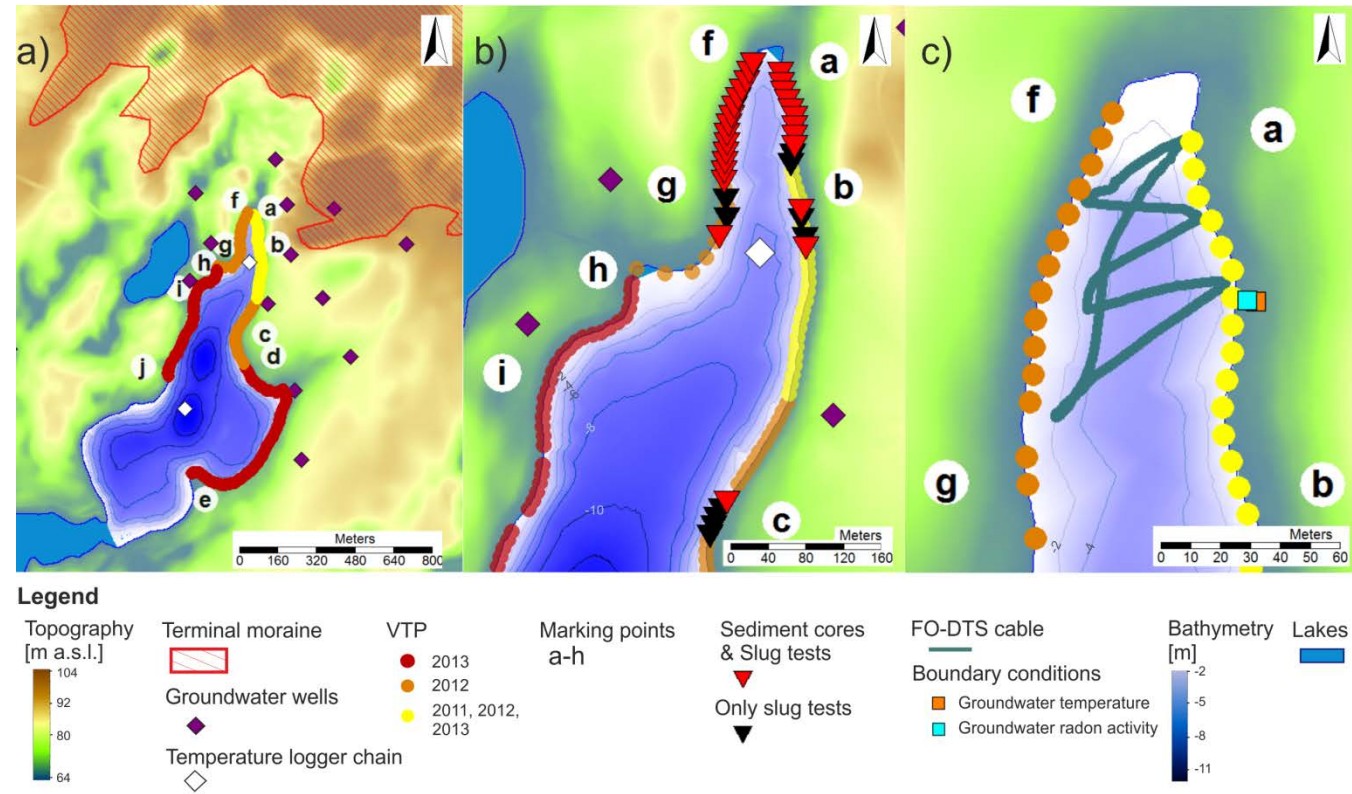

**Figure 9: Study site and experimental infrastructure.** (̶a̶) Overview of the study site with VTP measurement ~~and radon sampling locations~~ and locations of groundwater wells and temperature logger chains, (̶b̶) Slug test and sediment core sampling locations, (̶c̶) FO-DTS cable installation ~~and radon sample location~~ in the northern part of the lake.

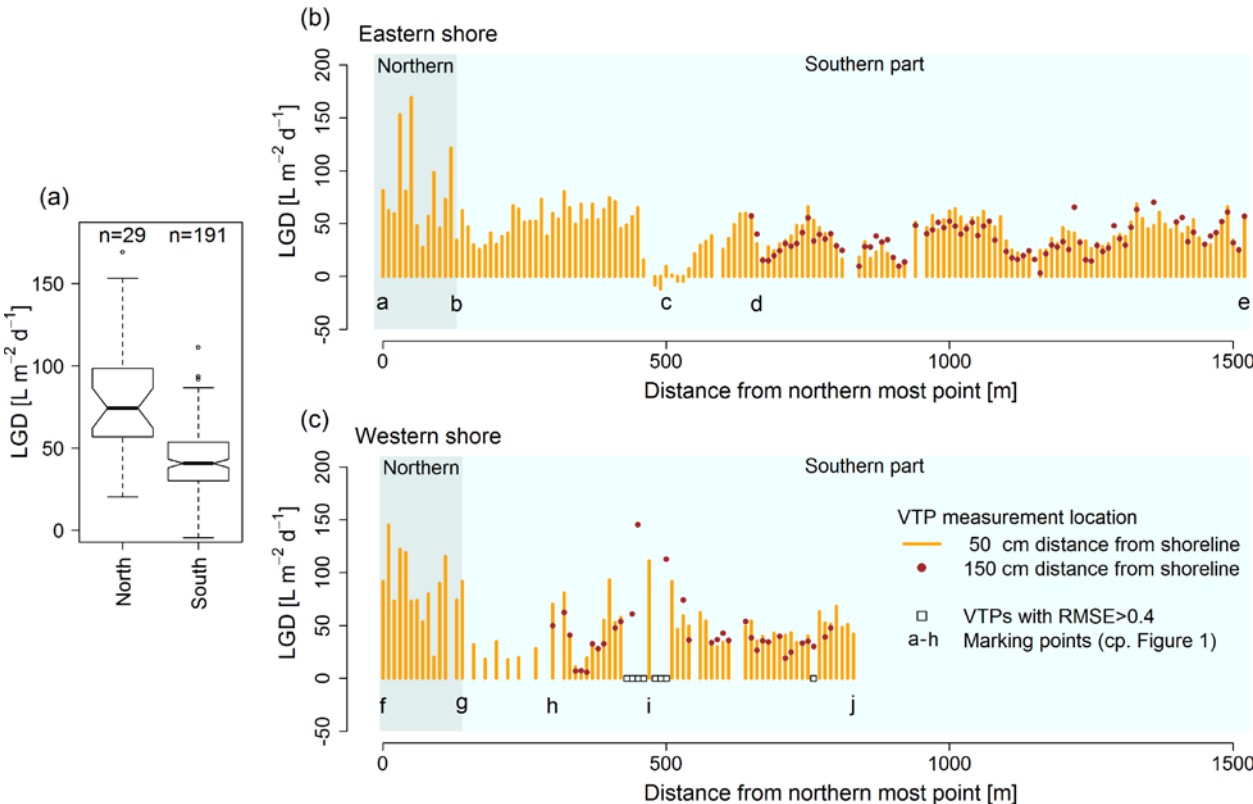

**Figure 10: LGD estimated from VTPs measured at 50 cm and 150 cm distance from shoreline, (a) LGD distribution from VTPs in the northern and southern part measured at a distance of 50 cm LGD along eastern shore (b) and LGD along western shore (c). Locations where fits of the heat transport equation were poor (RMSE>0.4) are indicted with squares.**

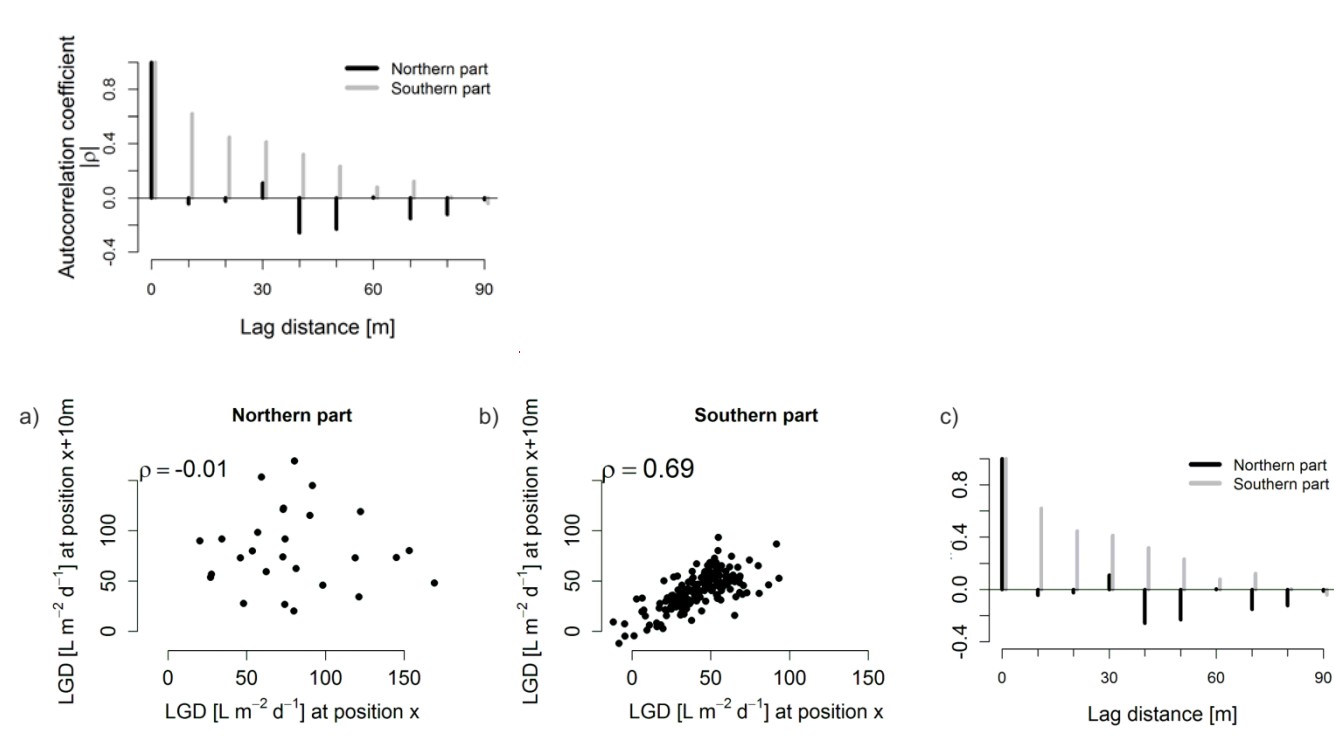

**Figure 11:** Correlation between neighbouring LGD measurement locations (distance 10m) in the northern part (a) and southern part (b), ~~Autocorrelogram~~ autocorrelogram for the LGD series of the northern and southern part of the lake (c).

We added the two scatter plots to clarify the use of the autocorrelation plot as well as the difference between the north and the south

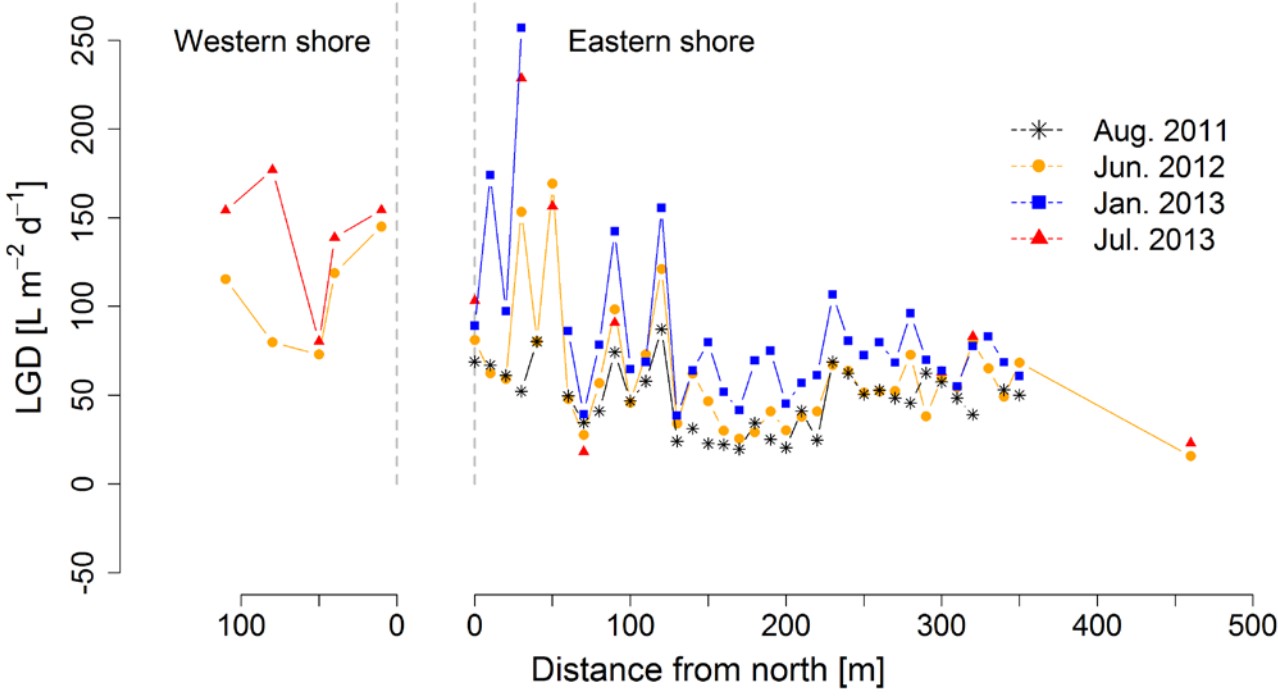

**Figure 12: Calculated LGD rates from repetitions of VTP measurements in August 2011, June 2012, January 2013 and July 2013 at the northern western and eastern shore.**

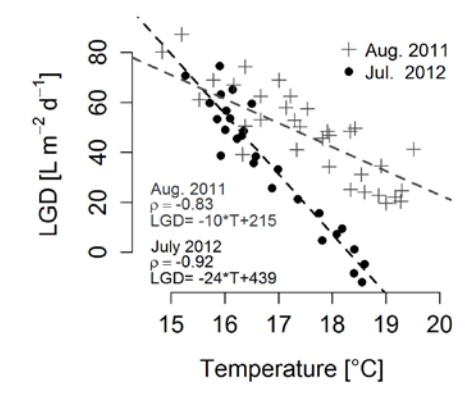

**Figure 5: Sediment temperature measured 30 cm below the sediment lake interface during two different VTP surveys vs LGD rates estimated from VTPs.**

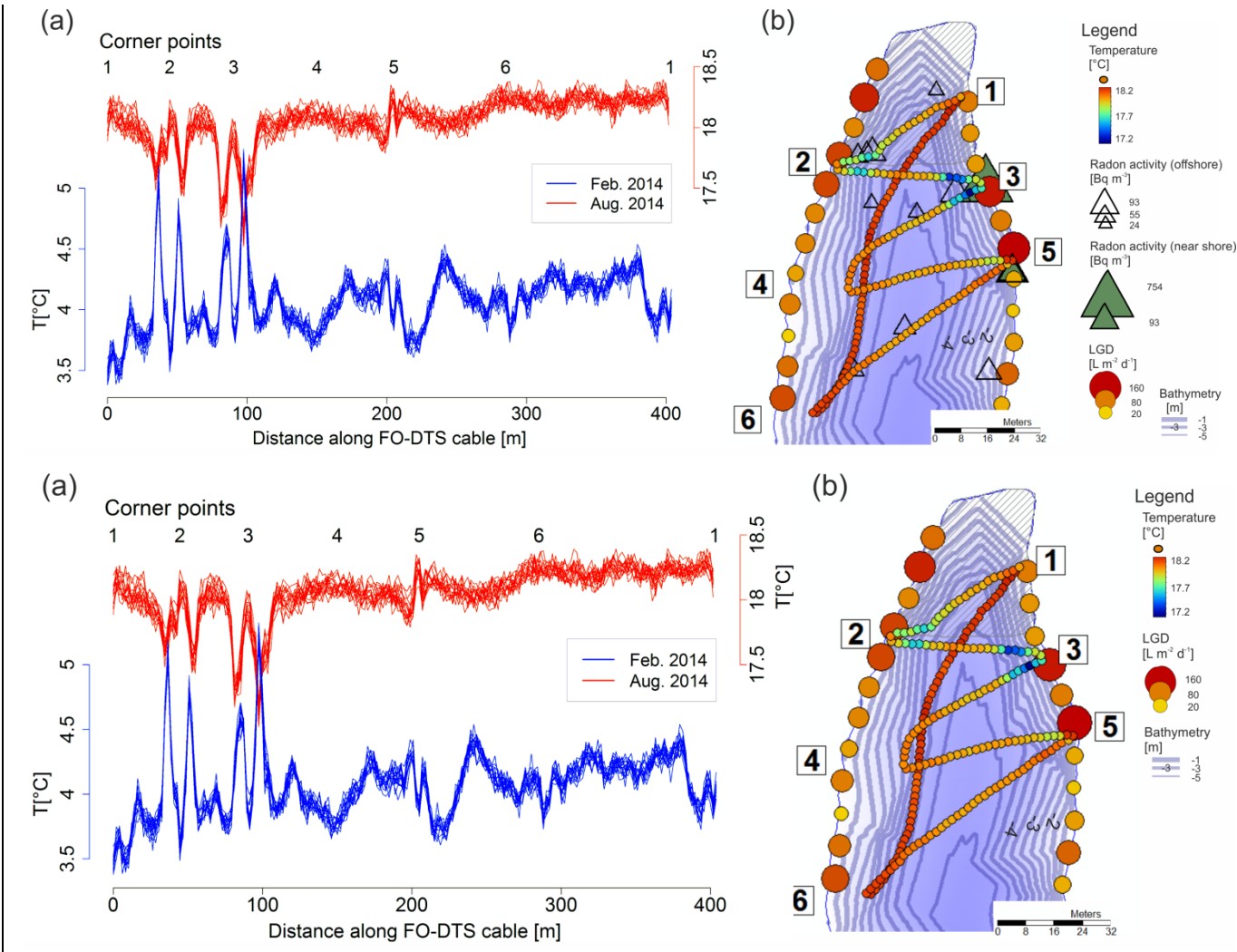

**Figure 56: Lake sediment temperatures measured with the FO-DTS system. (a) Temperatures measured in February and August along the FO-DTS cable. (b) ~~Localisation of S~~sediment temperatures measured with the FO-DTS in August 2014 (median) ~~and radon activities measured in September 2013 and  August 2014.~~ LGD rates along the shoreline were derived from VTPs measured in June 2012.**

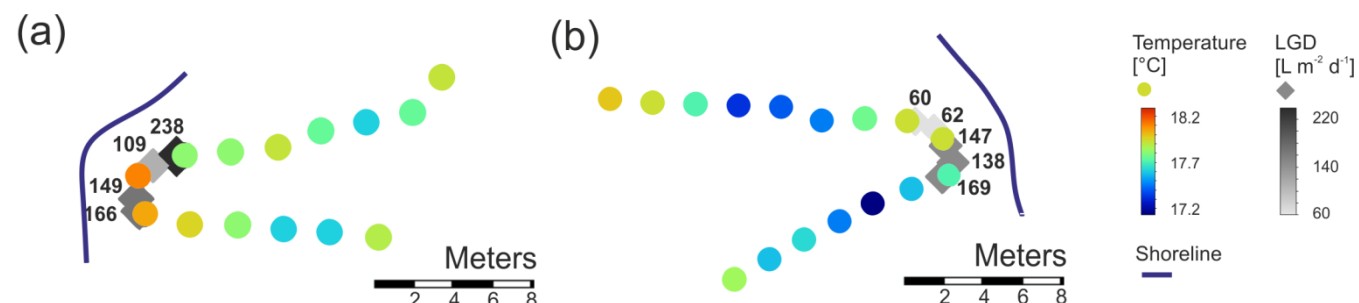

**Figure 6: Sediment temperatures measured with the FO-DTS in August 2014 (median) and LGD rates derived from VTPs measured at corner two (a) and three (b). Labels of rhombuses show the measured LGD rates.**

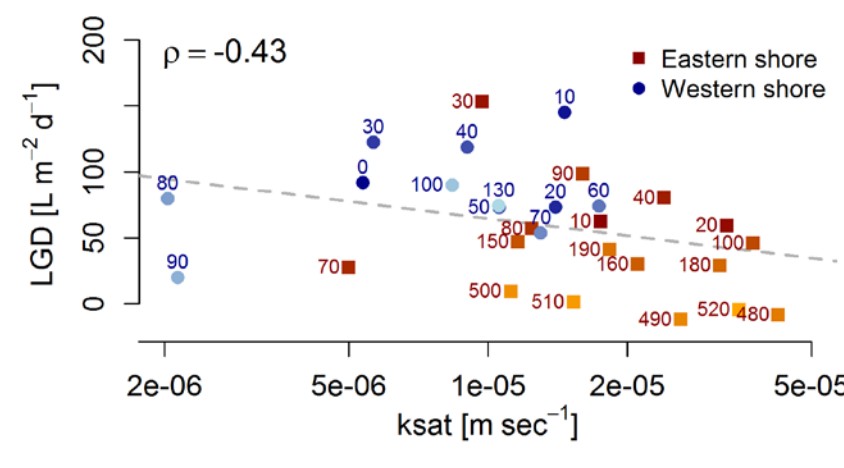

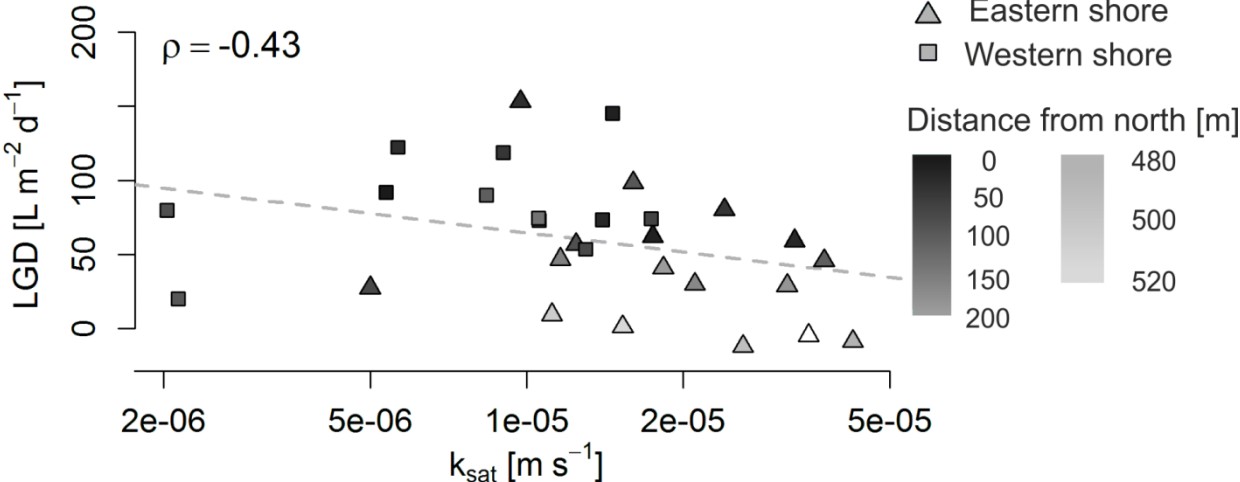

**Figure 13: LGD plotted against ~~kf~~ k_sat values determined from slug test at the western and eastern shore, ~~the labels and the strength of~~ the grey ~~shadingcolour~~ indicates the distances of measurement locations from the northern tip of the lake.**

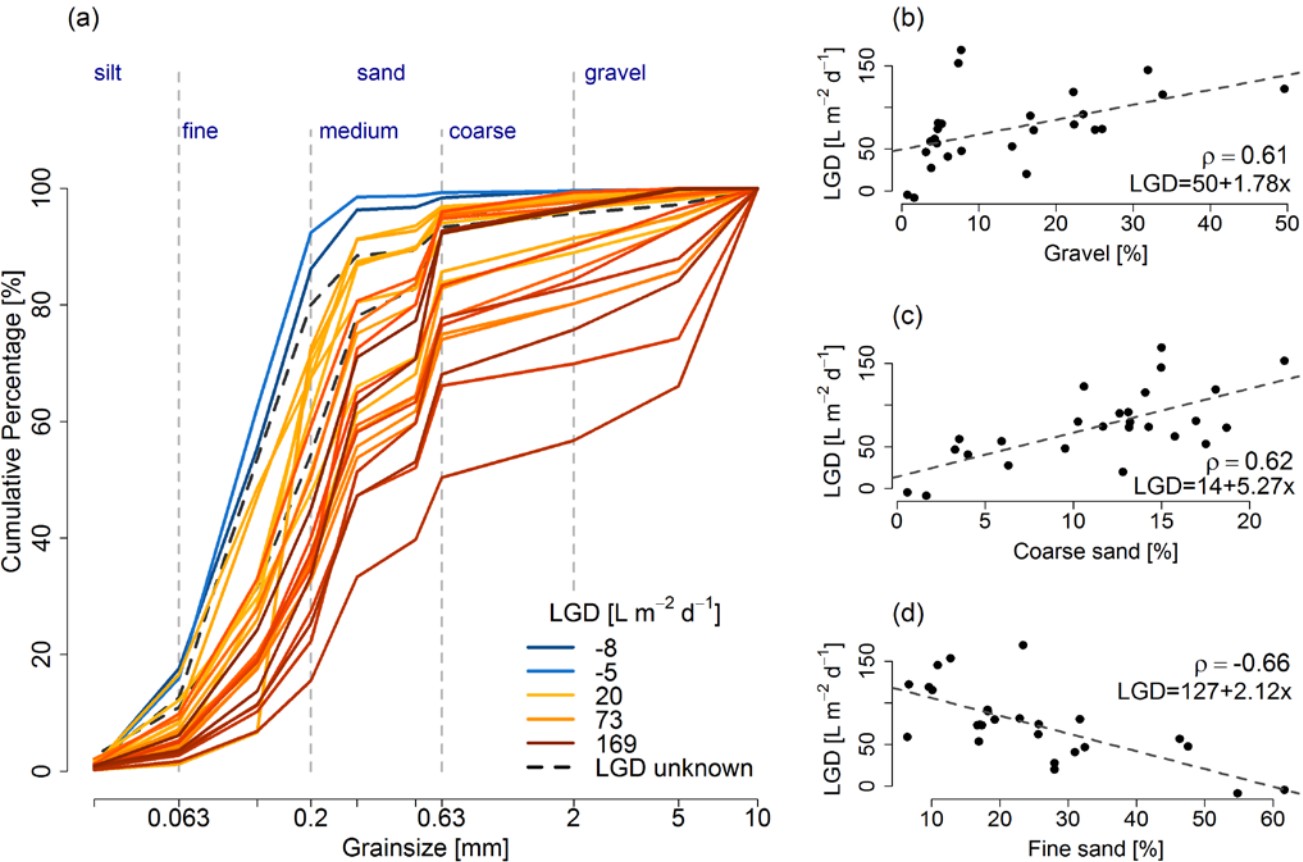

**Figure 14: (a)** Grain size distributions _averaged over the upper meter of the lake sediment_ from sediment cores coloured by the strength of LGD rate; **(b–d)** LGD rates are plotted against the grain sizes gravel (b), coarse sand (c) and fine sand (d).

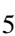

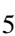

**Figure 15: Observed and calculated LGD distribution along the shore line. (a) Small scale patterns predicted using a multiple regression model with coarse sand and fine sand as predictor and (b) large--scale patterns ~~using~~ predicted by the linear model ~~considering~~ based on groundwater gradients derived from regression kriging from zone $zi_{25m}$. Regression equations for both small and large-scale patterns are included in the upper right corner. In the small-scale variability equation $x_1$ stands for the fraction of coarse sand and $x_2$ for fine sand.**

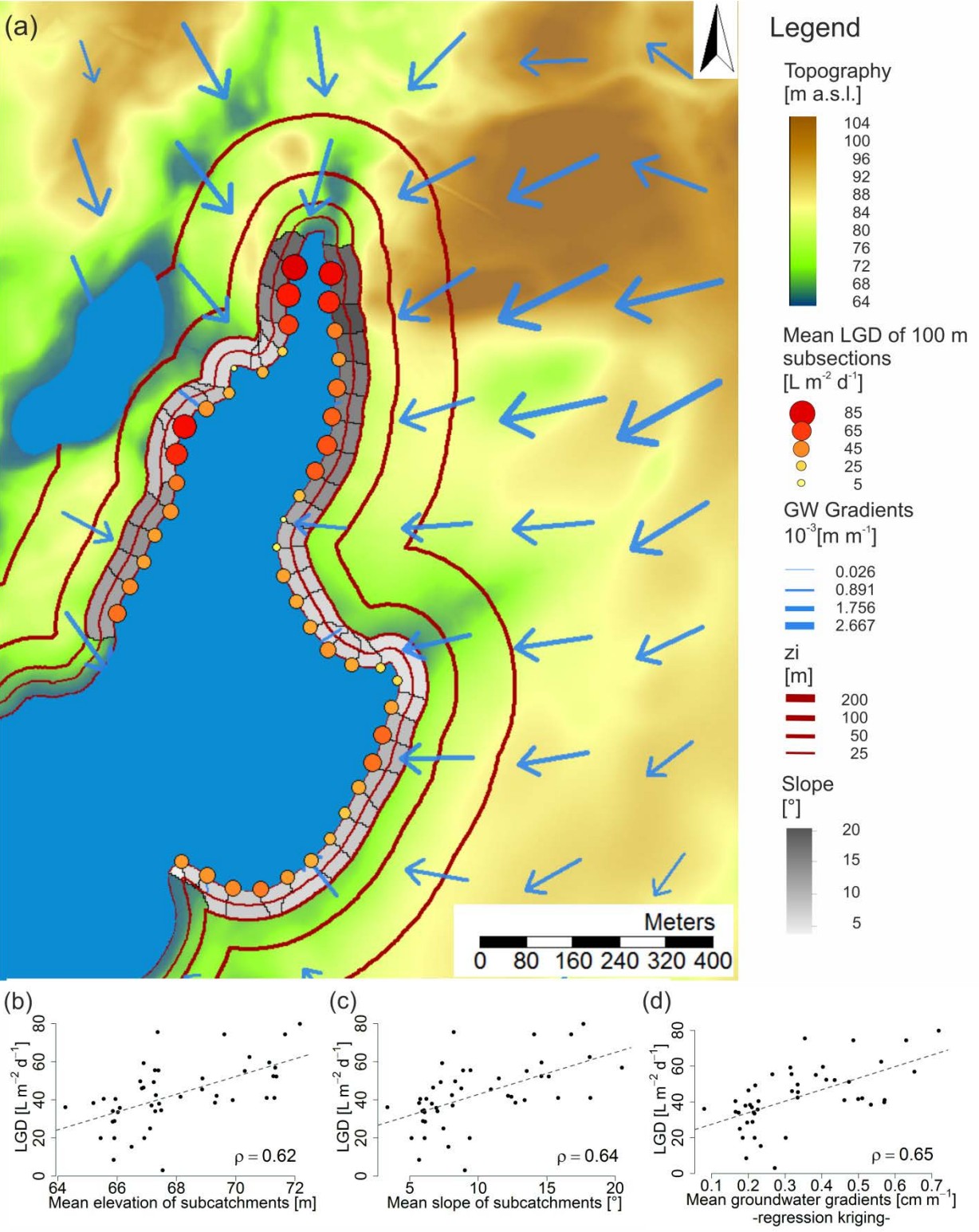

**Figure 16: Correlation between far-field conditions and LGD.** (a) LGD of lake subsections and mean slope of upslope areas for topographical zone of influence (zi) of 50 m, Groundwater gradients (GW gradients) are derived from interpolation of measured groundwater levels using regression kriging. (b–de) LGD rates of lake subsections are plotted against the far-field conditions mean elevation (b), mean slope (c) and mean groundwater gradients derived from regression kriging (d) and ordinary kriging calculated for $zi_{50m}$.