# Peer review of "Identifying, characterizing and predicting spatial patterns of lacustrine groundwater discharge"

_Hydrology and Earth System Sciences, 2016_

## Referee Comment (RC1) · Anonymous Referee #1 · 22 Jan 2017

The paper describes a case study of a very detailed investigation of groundwater discharge patterns to a lake. The authors used a range of methods and took a large number of measurements with the goal to derive relationships between parameters that are easy to obtain (e.g. topographic indices) and groundwater discharge patterns. They found that groundwater discharge was higher closer to the lake shore. They also found correlations between topography and large-scale groundwater inflow patterns, and between small-scale groundwater inflow patterns and sediment grain size distributions.

The paper does not really present anything new. The methods are well established and the outcomes confirm the general assumptions made about groundwater discharge

to lakes, including that it is usually very heterogeneous. The only surprise was that hydraulic conductivity did not seem to correlate with groundwater discharge.

However, the investigation was well designed, the methods are nicely described and the paper is overall very well written. I think it therefore merits publication and it will be a very useful reference for other researchers working on similar subjects.

I only have a few minor comments that should be addressed before publication:

There is no information on the motivation for this study. Was it a purely science-driven study and the lake was selected for convenience reasons, or was there a problem that drove the initiation of the project, such as lake water quality issues which may result from groundwater discharge? How does the selection of the lake impact on the transferability of the findings?

P9L26: Groundwater levels were measured "regularly": What does that mean? "Regular" can be once every year… please specify.

P15L20: You give transpiration as a possible reason for a near-shore depression in groundwater levels. Is there a type or density of vegetation at this location that potentially transpires more than at other locations at the lake shore, i.e. is there a reason to believe transpiration could be the cause?

Conclusion: My main question is: so what? Your main recommendation seems to be to take topographic indices combined with a few sediment cores as a first step, and then do more investigations at areas of interest. But which ones are most useful and give you most value for money? Are the findings transferable to other lake settings?

I am also wondering how such an investigation would help remediation planning. How important is it to know the spatial distribution or local hot spots of groundwater discharge to a lake? What could you do about those hot spots, or would you use them to trace back a contamination source?

References: P10L11-13: the two references are missing in the reference list

[Figure]

Language/typos:

Readability could be improved by using more hyphens, e.g.:

Large-scale patterns

Small-scale variability

High-precision thermometer

Near-surface sediments

Far-field conditions

Climate-driven processes

and similar constructions throughout the manuscript.

"Grain size" should be two words

P1L27: groundwater-lake exchange

P7L22/23: "purged for 3.5 at least hours" should be "purged for at least 3.5 hours"

P8L4 and 6: I think internationally the term "screen" is preferred to "filter"

P8L27: mean. . .was or means. . .were

P9L1: goodness-of-fit

P10L4: "resolution of 1 m that" should be "so that"

P11L1: "norther" should be "northern"

P11L31: "measurements in2 minute intervals" should be "measurements in 2-minute intervals"

P1L28: "a slight negative, but spastically" should be "a slightly negative, but statistically"
P15L31/32: ksat should be ksat (subscript)

P16L5: remove comma after structure

Figures:

Figure 2: The use of (a) (b) and (c) in the caption is a bit confusing

Figure 7: kf should be ksat in the caption

Figure 10: (e) is missing in the caption

---

## Referee Comment (RC2) · Anonymous Referee #2 · 21 Feb 2017

General comments This manuscript describes intensive field studies of LGD processes. An impressive amount of field data is presented, and an attempt is made to better comprehend spatial patterns of LGD on different spatial scales based on this large set of data. The major novelty of this contribution is the amount of data collected (as eluded to by the authors), but in my view overall few new insights into LGD processes are presented in the ms in its present form. In general, the ms is well-written, although it would benefit from a more concise and clearer structuring considering the large amount of data presented. I have a number of general comments that I believe should be addressed: 1) In parts, the ms reads like a data report, describing one experiment after another, with insufficient linkage between sections. It might help to

re-structure and use the same subsection titles in methods and results. Similarly, the figures, in particular the presented maps, are very busy. Consider breaking up the figures into smaller sub-figures rather rhan overlying too many things on one graph. I encourage the authors to carefully consider which level of information and especially which data are really required to support the main findings and eliminate those that aren't. For example, data or analyses that have not yielded conclusive results could be removed, but mentioned in one sentence that this was tried, moved to an appendix etc. 2) I suggest to remove the radon part from the study. Data is very sparse, and does not contribute significantly to the final interpretation. It is argued that low Radon concentrations in the lake's center are a result of low LGD, but they could equally well be a result of (a) not taking into account greater depth (ie inventories were not calculated) (b) wind-driven radon loss to the atmosphere. In short, there is not sufficient information to adequately interpret the radon data. 3) It appears that the fibre-optics part of this study has previously been published (Blume et al 2013), or that the data presented here does not add new insights to those found previously. Please explain in detail what is new and / or consider reducing the part on FO-DTS in this ms. 4) It is in times difficult to follow the heavy reliance on statistical treatment. For example, what is the point of the autocorrelation analysis of LGD values (e.g. page 5)? Is there a physical process associated with LGD that requires an understanding of autocorrelation of LGD along a shoreline (there is no use made of this analysis in the discussion unless I overlooked it?). 5) A linear mixed effects model should be applied instead of independent linear relationships with explanatory far field predictors. 6) Data distribution is very irregular, in particular the largest coverage is in the northern section of the lake where you already know that the highest LGD is found from a previous paper. In addition, the lake is connected to a larger lake to the south (of which you make no mention). Could this affect LGD patterns, in particular with respect to far field predictors? 7) LGD flux was estimated / calculated / modelled using different methods, and it is somewhat difficult to reconcile the results obtained by different methods in quantitative terms. How do the flux estimates compare, and are they consistent with each

other? 8) It was found that LGD correlated with grain size but not with slug test results (hydraulic conductivity). This is surprising (in particular in a large data set), and would require more detailed discussion. Were the slug tests done correctly, correct equations applied (there are a few for different experimental configurations), is there a possibility that there is a problem with the slug test results?

In the end I am a little lost what key message to take away from the ms. I believe this is likely due to 'too much data, too much analyses'. In this light, I suggest to include a detailed account of the value of each of the applied methods (incl a comparison and appraisal of the different tools to estimate flux on different scales), and to provide a 'recipe' how to design a 'good' study on LGD variability in the future. In summary, whilst some aspects of the study provide new general insights into LGD variability, overall this ms appears to be primarily of regional interest and as such I wonder if it would not be better placed in a journal with a more regional focus e.g. Journal of Hydrology – Regional Studies.

Specific comments Abstract – include relevant quantitative data on results (order of LGD fluxes etc). Also, include size of lake in abstract and study site description. Page 2 line 15-20 and relevant sections after: previous study results are not contradictory – they just highlight the fact that the effect of sediment structure on discharge is highly site-specific. Are your findings of a more general nature and as such applicable to other sites ? Page 3 methods: include site coordinates and a larger overview map. Page 5 line 23: show example data with and without SGD in a graph. Page 6: line 2: why compare 2 datasets with RMS / statistics? Why not just subtract one from another and analyse the difference if you have data from the same locations? Use statistical methods only where they are required. Page 5 line 14: provide temperature values of calibration points. Page 6 line 29: include month (August). Page 10 line 21: VTP could not be taken in areas with high stone content – however, these are areas with potentially high bulk hydraulic conductivity and could present a preferential flow path. Please acknowledge that you might have missed important entry points or

explain why they are not considered important. Page 10 line 27 and onwards: the interquartile range is irrelevant as far as I can see. Further on in the section, you statistically analyse the LGD variation along lag distances. What is the point of this analysis, are the results interpreted in the discussion, and what is the physical meaning behind them? Again, use simpler metrics and less statistics where possible. Pag2 12 line 11 – am I mistaken or does this indicate higher LGD rates further offshore than measured with other methods? If so, this would not fit with your overall appraisal. Along those lines, please provide a table that summarises the LDG rate measurements with different tools including error bars and discuss advantages and disadvantages of different methods.

Page 13 line 5 – provide a grain size map, remove most of the text in this paragraph describing the grain size distribution and include only one or two sentences with the main information on grain size distribution that is required to understand the context. Page 13 line 15: include the data in table 3 in figure 4. The grain size model is nice, although I wonder a little about applicability given the large residual errors? In my view the most interesting message of this paper is shown in Figure 8: a variation of LGD of up to a factor 3 can be due to grain size alone. Page 13 line 32 and onwards – remove ordinary kriging from the ms (and remove this paragraph which is then obsolete) and use only the method that works best. If you can, justify the use of regression kriging. Similarly, remove the discussion section page 15 lines 5-9 – it is a circular argument to say that regression kriging is the more appropriate method because results fit better. Page 14 line 11 and onwards: applying individual linear regressions between far field predictors and LGD rates assumes the simplistic view that parameters can be isolated. Instead, use a linear mixed model to comprehensively analyse the combined effect of topographic far field predictors on LGD. Page 15 line 16-20 : can flow reversals be a result of measurement errors? Page 16 line 11: give % variations. Page 16 line 21: it is unfortunate that this important data is missing. This would have considerable strengthened the ms. Conclusions – only include FO-DTS results that provide new insights compared to those already published by your group. Fig 9 – would be better

presented in a plot modelled vs measured. Figures – include bathymetry contour labels in maps.

---

## Author Comment (AC1) · 20 Mar 2017

Response to comments by reviewer#1

We would like to thank the reviewer for the very constructive and helpful review which provided great support for the improvement of the paper. We have addressed all issues indicated in the review report. Please find detailed responses to the reviewers' suggestions and comments below.

The paper describes a case study of a very detailed investigation of groundwater discharge patterns to a lake. The authors used a range of methods and took a large number of measurements with the goal to derive relationships between parameters that are easy to obtain (e.g. topographic indices) and groundwater discharge patterns. They found that groundwater discharge was higher closer to the lake shore. They also found correlations between topography and large-scale groundwater inflow patterns, and between small-scale groundwater inflow patterns and sediment grain size distributions. The paper does not really present anything new. The methods are well established and the outcomes confirm the general assumptions made about groundwater discharge to lakes, including that it is usually very heterogeneous. The only surprise was that hydraulic conductivity did not seem to correlate with groundwater discharge. However, the investigation was well designed, the methods are nicely described and the paper is overall very well written. I think it therefore merits publication and it will be a very useful reference for other researchers working on similar subjects.

I only have a few minor comments that should be addressed before publication:

There is no information on the motivation for this study. Was it a purely science-driven study and the lake was selected for convenience reasons, or was there a problem that drove the initiation of the project, such as lake water quality issues which may result from groundwater discharge?

At Lake Hinnensee significant lake level fluctuation were observed in the last 30 years, but processes and mechanisms driving these fluctuations and the role of lacustrine groundwater discharge (LGD) within the water balance of the lake were only poorly understood.
We will include the background of the study within the introduction (in subsection 1.3 Objectives) on P3L21 as follows:
"To study these research questions we chose Lake Hinnensee, a typical post-glacial lake located in the intensively monitored TERENO observatory in the lowland landscape of northeast Germany. At this lake strong water level fluctuations were observed in the last decades, but processes and mechanisms causing these lake level fluctuations are yet not well understood. "

How does the selection of the lake impact on the transferability of the findings?

We assume that our findings are transferable to other lakes in similar landscapes, which means humid lowland landscapes. We assume that the main important restriction of the transferability of our findings is the occurrence of complex hydrogeological settings overriding the topographical signal, for example discontinuities in the aquifers. We will address this question in the conclusion of the revised manuscript.

P9L26: Groundwater levels were measured "regularly": What does that mean? "Regular" can be once every year…please specify.

Thank you for this advice. Groundwater levels were measured approximately bimonthly. We will change the sentence as follows:
"Groundwater levels were measured  roughly bimonthly (intervals of 7-9 weeks) from 2012 to 2014 using an electric contact meter (SEBA Hydrometrie, electric contact meter type KLL, accuracy: ± 1 cm)."

P15L20: You give transpiration as a possible reason for a near-shore depression in groundwater levels. Is there a type or density of vegetation at this location that potentially transpires more than at other locations at the lake shore, i.e. is there a reason to believe transpiration could be the cause?

The type and density of vegetation does not significantly differ from other locations, but this lake section is characterized by low groundwater gradients. Transpiration could reduce groundwater gradients (Winter et al., 1998) and if groundwater gradients are already very low, transpiration could even cause a reversal of groundwater gradients. Thus, we assume the reversal results from the combination of transpiration and the occurrence of low groundwater gradients.

We will make this clearer by changing the paragraph as follows:

"Even though the interpolated groundwater surface showed groundwater flow towards Lake Hinnensee from all directions (Figure 10), we measured negative LGD rates at one small subsection of the lake (Figure 2). Reasons for this flow reversal are unclear. However, this lake section is characterized by low groundwater gradients (Figure 10). While transpiration is likely to cause diurnal fluctuations in groundwater levels all around the lake, it can result in a local inversion of the groundwater – lake gradients at locations where these gradients are very low (Winter et al., 1998)."

Conclusion: My main question is: so what? Your main recommendation seems to be to take topographic indices combined with a few sediment cores as a first step, and then do more investigations at areas of interest. But which ones are most useful and give you most value for money?

The value of using topographic indices or sediment cores to predict LGD patterns depends on the focus of the study. From our results, we assume that topographic indices can help to predict large scale LGD patterns and sampling sediment cores could help to get a picture of small scale variability in LGD patterns. This will be added in the conclusion as follows:

P18 L23-25 ", as topographic gradients could help to get information on large scale patterns and sediment cores could be used as indicator for small scale patterns."

We will clarify the outcome of our study by including a short discussion on the employed methods and by providing a recipe for designing a study on LGD variability. In this part we would emphasize the advantages of using a needle for measuring sediment temperatures, because this instrument allows precise sediment temperature measurements without disturbing the sediment and the flow. Furthermore, we will evaluate the use of temperature measurements from one single sediment depth (instead of the entire profile) to get a fast impression of LGD patterns and will therefore present correlations between LGD rates estimated from VTPs and sediment temperatures from different depths of the of the profile. Sediment temperatures from the top of the sediment down to a depth of 10 cm were not well correlated with LGD rates, but strong correlations were found between LGD rates and sediment temperatures measured 20 cm below surface and deeper (absolute correlation coefficients ranged between 0.46 and 0.96, see Figure below). Measuring sediment temperatures only in one sediment depth instead of complete VTPS would save a lot of time in the field. A discussion of these points will be included in the revised manuscript.

[Figure]

Figure 1: Sediment temperature measured 30 cm below the sediment lake interface measured during different VTP surveys are plotted against LGD rates estimated from VTPs.

Are the findings transferable to other lake settings?

As mentioned above, we assume that our findings are transferable to other lakes in similar landscapes, which means humid lowland landscapes in which the groundwater tables likely follow the topography. We assume that the main important restriction of the transferability of our findings is the occurrence of complex hydrogeological settings overriding the topographical signal, for example discontinuities in the aquifers. We will address this point in the recipe for an experimental designs focusing on LGD variability.

I am also wondering how such an investigation would help remediation planning. How important is it to know the spatial distribution or local hot spots of groundwater discharge to a lake? What could you do about those hot spots, or would you use them to trace back a contamination source?

In general, knowing LGD variability, especially identifying LGD hotspots, is important from two perspectives: for water quantity issues and water quality issues. Missing hotspots of LGD could lead to a wrong estimation of LGD and hotspots could be of great importance, as these hotspots could carry larger loads of contaminations, especially if LGD hotspots meet contamination hotspots. You could thus indeed use this to trace back contamination sources. This will briefly be mentioned in the beginning of the conclusion as follows:
P15 L2: "Quantifying LGD rates and determining LGD patterns can be essential for a sustainable lake management if LGD significantly contribute to a lake water budgets or significantly influence lake water quality by transporting large loads of nutrients or contaminants (Meinikmann et al., 2013; Lewandowski et al., 2015)."

References: P10L11-13: the two references are missing in the reference list
Thank you for this advice. We will include these references in the reference list.

Language/typos:

Readability could be improved by using more hyphens, e.g.: Large-scale patterns, Small-scale variability, High-precision thermometer, Near-surface sediments, Far-field conditions, Climate-driven processes and similar constructions throughout the manuscript. "Grain size" should be two words

P1L27: groundwater-lake exchange

P7L22/23: "purged for 3.5 at least hours" should be "purged for at least 3.5 hours"

P8L4 and 6: I think internationally the term "screen" is preferred to "filter"

P8L27: mean was or means were

P9L1: goodness-of-fit

P10L4: "resolution of 1 m that" should be "so that"

P11L1: "norther" should be "northern"

P11L31: "measurements in2 minute intervals" should be "measurements in 2-minute intervals"

P1L28: "a slight negative, but spastically" should be "a slightly negative, but statistically"

P15L31/32: ksat should be ksat (subscript)

P16L5: remove comma after structure

Figures:

Figure 2: The use of (a) (b) and (c) in the caption is a bit confusing

Figure 7: kf should be ksat in the caption

Figure 10: (e) is missing in the caption

Thank you for these specific and valuable advices. We will include these suggestions in the revised manuscript.

---

## Author Comment (AC2) · 20 Mar 2017

Response to comments by reviewer#2

The authors would like to thank the reviewer #2 for the constructive and helpful review. It provided great support for the improvement of the manuscript. However, some of the comments seem to be the result of misunderstandings. We hope that our answers and changes in the manuscript clarify the misunderstandings. Please find detailed responses to the specific reviewers' suggestions and comments below.

General comments
This manuscript describes intensive field studies of LGD processes. An impressive amount of field data is presented, and an attempt is made to better comprehend spatial patterns of LGD on different spatial scales based on this large set of data. The major novelty of this contribution is the amount of data collected (as eluded to by the authors), but in my view overall few new insights into LGD processes are presented in the ms in its present form. In general, the ms is well-written, although it would benefit from a more concise and clearer structuring considering the large amount of data presented.

I have a number of general comments that I believe should be addressed:

**1**) In parts, the ms reads like a data report, describing one experiment after another, with insufficient linkage between sections. It might help to re-structure and use the same subsection titles in methods and results.

Thank you for the constructive advice. The structure was adapted and now fits the short outline given at the end of the introduction (p3 L21-24) and, as suggested as well, we changed subsection titles in methods and result that they are now the same.

| 2. Methods | 3. Results |
|---|---|
| 2.1 Study site | |
| 2.2 Estimating lacustrine groundwater discharge (LGD) | 3.1 Estimating lacustrine groundwater discharge (LGD) |
| 2.2.1 Near-shore LGD estimations from vertical temperature profiles (VTPs) | 3.1.1 Near-shore LGD estimations from vertical temperature profiles (VTPs)
Spatial patterns along the shore line
Spatial patterns perpendicular to the shore line
Temporal stability of spatial pattern |
| |  |
| 2.2.2 Lake sediment temperature anomalies as indicators for LGD based on fibre optic distributed temperature sensing (FO-DTS) | 3.1.2 Lake sediment temperature anomalies as indicators for LGD based on fibre optic distributed temperature sensing (FO-DTS) |
| 2.2.3 Identification of offshore LGD based on radon concentrations  | 3.1.3 Identification of offshore LGD based on radon concentrations |
| 2.3 Identifying controls of LGD patterns | 3.2  Identifying controls of LGD patterns |
| 2.3.1 Sediment heterogeneity as a small scale control
Hydraulic conductivity from slug tests
Grain size distributions from sediment cores | 3.2.1 Sediment heterogeneity as a small scale control
Hydraulic conductivity from slug tests
Grain size distributions from sediment cores   |
| |  |

| | |
|---|---|
| 2.3.2 Topographic indices as controls on large scale LGD patterns | 3.2.2 Topographic indices as controls on large scale LGD patterns |
| 2.3.3 Groundwater flow field as control on large scale LGD patterns | 3.2.3 Groundwater flow field as control on large scale LGD patterns |
| | 3.2.4 Linear regression models between LGD and far-field predictors |

Similarly, the figures, in particular the presented maps, are very busy. Consider breaking up the figures into smaller sub-figures rather rhan overlying too many things on one graph. I encourage the authors to carefully consider which level of information and especially which data are really required to support the main findings and eliminate those that aren't. For example, data or analyses that have not yielded conclusive results could be removed, but mentioned in one sentence that this was tried, moved to an appendix etc.

Thank you for this general suggestion. Concerning the suggestion of reducing data and data analysis, detailed responses are given on specific suggestions and comments below. Concerning the maps and figures we followed your suggestion and checked each figure and map concerning the level of information. We agree with the reviewer that some maps are busy, but we feel that leaving out data from maps or breaking up figures into subfigures would lead to a decrease of information on either spatial distribution of LGD or spatial correlation between different LGD measurement methods and would therefore keep the level of information in the figures, but to increase the clarity in the figures, we would modify the figures as follows:

- Figure 1b: bathymetry contour labels will be included
- Figure 2 b-c: different background colours will be used for the southern and northern part and x labels will be inserted in figure 2b.
- Figure 3: y label will be changed from "|ρ|" to "autocorrelation coefficient (|ρ|)"
- Figure 5b: we will use columns instead of triangles as symbols for radon activity and different colours will be used for near-shore and off-shore radon samples.
- Figure 6: subfigures will be placed side by side instead of among each other, Shoreline, map scale and bathymetry contours with labels will be included
- Figure 7: labels will be excluded and presentation of the distances of measurement locations will be enhanced by grouping the measurement locations into two groups: first group: 0 m - 190 m south from the northern tip of the lake, second group: 480 m – 520 m south from the northern tip of the lake. For each group a different colour ramp will be used. A corresponding legend will be added to the figure.
- Figure 10: legend of subfigure (a) will be placed right of the figure, subfigures (b-d) will be placed below figure (a). Subfigure (e) will be removed

**2)** I suggest to remove the radon part from the study. Data is very sparse, and does not contribute significantly to the final interpretation. It is argued that low Radon concentrations in the lake's center are a result of low LGD, but they could equally well be a result of (a) not taking into account greater depth (ie inventories were not calculated) (b) wind-driven radon loss to the atmosphere. In short, there is not sufficient information to adequately interpret the radon data.

We used the radon measurements in order to localise groundwater inflow in the offshore part of the lake. Our approach was orientated on the approach of Shawn et al., 2013 or Ono et al., 2012 measuring radon activities at the lake bottom at different locations and relating significant differences in radon activities to spatial patterns in groundwater inflow. Shawn et al., 2013 took samples when lake was frozen and could therefore neglect the effect of wind losses. However, by taking samples

from the lake bottom, we assume that the effects of wind-driven losses are low. Kluge et al. measured several vertical profiles in a lake and related increasing radon activities at the bottom of the lake to local groundwater inflow. Furthermore, our test of the ability of radon as an indicator for groundwater inflow at Lake Hinnensee showed significant stronger radon activities at locations with groundwater inflow (P12 L16-17), even though wind losses were not considered and lake depths were shallow (< 1 m) at these locations. Thus we assume, that measurements of radon activities from the lake bottom at Lake Hinnensee could be used to identify locations with significant groundwater inflow. To consider the comment, we would change the discussion as follows:

P16 L23 – 30 "We investigated the presence and absence of offshore LGD with two methods: radon and heat (sediment temperature measurements with FO-DTS) as natural tracers. Measured radon activities of lake samples indicate groundwater inflow, but activities were low.  Sampling locations were mainly located in the epilimnion, where lake levels were too shallow to allow for a thermocline. In the epilimnion, the lake water radon activity is assumed to be well mixed (Kluge et al., 2012). Low radon concentration sampled close to the lake bottom could be the result from radon degassing to the atmosphere and advective and diffusive radon fluxes from the lake bottom towards the lake atmosphere interface. But comparison of radon activities at locations with known groundwater inflow in the near shore part showed a significant effect of local groundwater inflow on radon activities which overrides possible wind-driven radon losses to the atmosphere. Kluge et al. 2007 who measured several vertical profiles in a lake related increasing radon activities at the bottom of the lake to local groundwater inflow.  As radon activities could be assumed to be well mixed in the epilimnion, the observed slightly enhanced radon activity in the deeper parts of the northern part of the lake in comparison to the southern part might be the result from stronger groundwater inflow from the near shore area instead from the deep sampling locations itself. As also FO-DTS showed no shifts in sediment temperatures towards the groundwater temperature in the flatter and deeper parts of the lake (Figure 5b), we assume that groundwater inflow is insignificant here."

We are not sure what is meant by "not taking into account greater depth". Indeed, we did not calculate inventories, as we were only interested on identifying presence or absence of groundwater inflow in the central part of the lake, not in calculating groundwater inflow rates. For the purpose of calculating groundwater inflow rates, indeed, sufficient information is not available, but this was as well beyond the scope of these measurements, which were only used as simple qualitative indicators. We furthermore agree that sampling locations were sparse as the number of samples was limited, but the sampling design covered the entire lake, including the deep parts in the south. As the radon data provide us with information from this southern area, where no other data are available, we would prefer to leave the data in the manuscript, but leave it to the editor to make this decision. Alternatively, we could move the radon part (section 2.2.3 and section 3.2.2) to an appendix.

**3)** It appears that the fibre-optics part of this study has previously been published (Blume et al 2013), or that the data presented here does not add new insights to those found previously. Please explain in detail what is new and / or consider reducing the part on FO-DTS in this ms.

The FO-DTS data were not presented before. Blume et al. (2013) conducted a FO-DTS study at Lake Hinnensee three years before our FO-DTS campaign, but the study focused only on a small shore section of 20 m length and 4 m width and the objectives were different, in case of  Blume et al. (2013) methodical-driven ("The objective of the present study is to test whether FO-DTS-based upscaling of point measurements of lacustrine groundwater discharge rates is an adequate and feasible approach to represent the spatial heterogeneity of LGD rates." (Blume et al., 2013)). The experimental designs differ strongly from each other. While Blume et al. (2013) arranged the DTS cable in the near shore part on a small section of 20 m length and 4 m width, we installed the cable along 6 transects across the lake. Beside the methodical insight from the study of Blume et al. (2013), they mentioned spatial variability in LGD and a decrease of LGD with increasing distance from shore within the focus area. As we found as well spatial variability in LGD and a decrease of LGD with increasing distance from shore, we compared our results with the previous study within the discussion part. To clarify the different approaches and foci of the two studies, we change the following sentence:

P16 L16 The study from Blume et al. (2013), conducted at a small shore line section of 20 m length and 4 m width in the northern part of Lake Hinnensee, indicated that the strongest decrease of LGD occurred in the first 1.5 m distance from shore.

**4)** It is in times difficult to follow the heavy reliance on statistical treatment. For example, what is the point of the autocorrelation analysis of LGD values (e.g. page 5)? Is there a physical process associated with LGD that requires an understanding of autocorrelation of LGD along a shoreline (there is no use made of this analysis in the discussion unless I overlooked it?).

Thank you for the advice we will reduce the statistical methods to increase ease of reading. The following changes will be made:
- removing MAD as an additional measure of spatial variability (P11 L4-6),
- replacing RMSE by median differences (P11 L24-25) and
- removing the description of grain size distribution (P 13 L 6 - 10).

Concerning the comment on autocorrelation: this method has the advantage that it clearly addresses the question, how similar neighboured measurement locations are and up to which distance a correlation between measurement locations exist. This is exactly what we wanted to show. To reduce statistical methods, we could alternatively present correlation plots and correlation coefficients of neighboured LGD measurement locations (Lag distance 10 m, spatial resolution of VTP measurements), as presented in the figures below. We would prefer to leave the autocorrelation analysis in the manuscript, but leave the decision to the editor. However, we will include the result of correlation between LGD measurement locations within the discussion and would clarify the introduction into the method of autocorrelation in the method part as follows:

P5 L27-29: Spatial variability and correlation of LGD along different distances along the shoreline were analysed using autocorrelation plots and autocorrelation values ($|\rho|$) as described in Caruso et al. (2016). High autocorrelation ($|\rho|$) indicate that LGDs along a given stretch of the shoreline are correlated, whereas $|\rho| < 0.2$, indicate that LGDs are uncorrelated and strong spatial variability exists.

[Figure]

**Figure 1: Correlation between neighbouring LGD measurement locations in the northern part (left) and southern part (right).**

**5)** A linear mixed effects model should be applied instead of independent linear relationships with explanatory far field predictors.

We applied simple regression models to examine the relation between single far field predictors and LGD and applied multiple regression models to examine the combined relation of all far field predictors with LGD. But the multiple linear regression models revealed no significant relationship (or improvements over the individual regressions) between LGD and the combined use of far field

predictors (P 14 L19-21). As we see no advantage by using linear mixed models instead of multiple linear regression models for our dataset and want to keep the statistical treatment as simple as possible, we would therefore prefer to use the multiple regression models instead of linear mixed models.

**6)** Data distribution is very irregular, in particular the largest coverage is in the northern section of the lake where you already know that the highest LGD is found from a previous paper. In addition, the lake is connected to a larger lake to the south (of which you make no mention). Could this affect LGD patterns, in particular with respect to far field predictors?

We do not expect an effect on LGD patterns at Lake Hinnensee as a result of the open connection to Lake Fürstenseer See in the south. From topography and the regional flow field, we expect in general smaller groundwater inflow rates at Lake Fürstenseer See and negative LGD rates in the south of Lake Fürstenseer See. One process might influence temporally LGD patterns at Lake Hinnensee: if wind direction is from south and wind is strong, wind- driven water movements might increase water table towards the north of Lake Hinnensee, which might reduce groundwater inflow rates in the northern part.
We will add a sentence explaining that the connection to Lake Fürstenseer See is unlikely to affect LGD patterns in Lake Hinnensee.

**7)** LGD flux was estimated / calculated / modelled using different methods, and it is somewhat difficult to reconcile the results obtained by different methods in quantitative terms. How do the flux estimates compare, and are they consistent with each other?

LGD rates were only estimated from VTPs. Subsequently regression models were used to model these LGD rates based on various predictors. FO-DTS and radon measurements were only used as qualitative indicators for offshore LGD. A comparison between measured and modelled LGD rates is given on P13 L17 -20 and P14 L23-26. A subsection with a comparison of the applied methods (VTPs, FO-DTS, radon) could be given in the discussion part. In order to make clearer that FO-DTS and radon measurements were only used as qualitative tracers, we insert the following sentence in the objectives:

P 3 L21-22 "To identify LGD patterns, we measured VTPs in the near shore area and used FO-DTS measurements and radon sampling in the off-shore area. VTPs were used to quantify LGD rates, whereas FO-DTS measurements and radon sampling were used as qualitative tracers for presence and absence of off-shore LGD. As potential controls of LGD patterns…"

**8)** It was found that LGD correlated with grain size but not with slug test results (hydraulic conductivity). This is surprising (in particular in a large data set), and would require more detailed discussion. Were the slug tests done correctly, correct equations applied (there are a few for different experimental configurations), is there a possibility that there is a problem with the slug test results?

We checked the equations again, but found no problems, we also checked the correlation applying the equation of Hvorslev (1951) without the shape correction factor introduced by Chapuis (1989) and a second adaption of the Hvorslev equation even though both equation fit  less well to our experimental setup. However, with both equations absolute values of estimated ksat changed slightly, but calculated correlation coefficients between LGD and $k_{sat}$ values were the same ($\rho$=-0.36). Thus conclusion from measurements would be the same even by applying different equations. Slug tests were carried out carefully in the field. However, as we discussed on P 15 L31 – P16 L5, we assume that the uncertainty and errors of the slug test method are larger than the differences in hydraulic conductivity in the sandy sediment at Lake Hinnensee.

In the end I am a little lost what key message to take away from the ms. I believe this is likely due to 'too much data, too much analyses'. In this light, I suggest to include a detailed account of the value of each of the applied methods (incl a comparison and appraisal of the different tools to estimate flux on

different scales), and to provide a 'recipe' how to design a 'good' study on LGD variability in the future.

Thank you for these constructive suggestions, we will include a subsection on evaluating the applied methods (VTPs, FO-DTS, radon) in the discussion part and discuss the problems of using piezometers to identify LGD rates and patterns due to uncertainties in $k_{sat}$ estimations. Furthermore we will give a recipe how to design a good study. In this part we would also emphasize the advantages of using a needle for measuring sediment temperatures, because this instrument allows precise sediment temperature measurements without disturbing the sediment and the flow. Furthermore, we will evaluate the use of temperature measurements from one single sediment depth (instead of the entire profile) to get a fast impression of LGD patterns and will therefore present correlations between LGD rates estimated from VTPs and sediment temperatures from different depths of the of the profile. Sediment temperatures from the top of the sediment down to a depth of 10 cm were not well correlated with LGD rates, but strong correlations were found between LGD rates and sediment temperatures measured 20 cm below surface and deeper (correlation coefficients range between 0.46 and 0.96 and see figure below). Measuring sediment temperatures only in one sediment depth instead of complete VTPS would save a lot of time in the field. A discussion of these points will be included in the revised manuscript.

[Figure]

Figure 2: Sediment temperature measured 30 cm below the sediment lake interface measured during different VTP surveys are plotted against LGD rates estimated from VTPs.

In summary, whilst some aspects of the study provide new general insights into LGD variability, overall this ms appears to be primarily of regional interest and as such I wonder if it would not be better placed in a journal with a more regional focus e.g. Journal of Hydrology – Regional Studies.

As our study bridges the observational gap between detailed local and low resolution regional investigations of previous LGD studies and highlighted correlation between LGD patterns and external controls, which are transferable to other landscapes with similar settings and provides methodological recommendations, we feel that our study does provide new insights which are beyond the regional

interest. The recommendations and transferability of our results will be more strongly highlighted in the revised manuscript.

Specific comments
Abstract – include relevant quantitative data on results (order of LGD fluxes etc). Also, include size of lake in abstract and study site description.

Following your suggestion, we will include lake size, mean and maxima of measured LGD rates and correlation coefficients.

Page 2 line 15-20 and relevant sections after: previous study results are not contradictory – they just highlight the fact that the effect of sediment structure on discharge is highly site-specific. Are your findings of a more general nature and as such applicable to other sites?

Thank you for your suggestion, we agree with the reviewer, the effect of sediment characteristics on LGD patterns is site-specific. Thus we will rephrase the section as follows:

"P2 L16-21There is no clear picture of the role of sediment characteristics in controlling LGD patterns. For example, Kidmose et al. (2013) found that low permeable lacustrine sediments can completely prevent groundwater upwelling, whereas Vainu et al. (2015) observed LGD through low permeable lacustrine sediments. Kishel and Gerla (2002) associated small scale variabilities in LGD with small scale heterogeneities in hydraulic conductivities (Kishel & Gerla, 2002), but in contrast, and Schneider et al., (2005) found no correlation between seepage rates and sediment characteristics.

Page 3 methods: include site coordinates and a larger overview map.

Site coordinates will be included, a general description of the location of the study site is given in the subsection 2.1 "Study site".

Page 5 line 23: show example data with and without SGD in a graph.

We are sorry, but we are not sure what is meant with this suggestion. The sentence on Page 5 in line 23 does not seem to fit to this suggestion and what is SGD?

Page 6: line2: why compare 2 datasets with RMS / statistics? Why not just subtract one from another and analyse the difference if you have data from the same locations? Use statistical methods only where they are required.

Thank you for the comment, RMSE will be replaced by the median of the differences.

P5 L32- P6 L2: "In order to analyse the temporal stability of spatial patterns we analysed the differences between the LGD rates measured in different surveys and calculated the correlation between the surveys using the Spearman's rank correlation coefficient (ρ). Correlations were regarded as significant for p-values smaller than 0.05. Differences between LGD rates measured in different years were quantified with the RMSE. "

P11L23-25: "The differences between LGD rates measured in different years were lowest comparing rates from summer 2011 and summer 2012 (median difference = -6 L m-2 d-1) and strongest comparing rates from summer 2011 and winter 2013 (median difference = 27 L m-2 d-1)."

Page 5 line 14: provide temperature values of calibration points.

Will be included

Page 6 line 29: include month (August).

Will be included

Page 10 line 21: VTP could not be taken in areas with high stone content – however, these are areas with potentially high bulk hydraulic conductivity and could present a preferential flow path. Please acknowledge that you might have missed important entry points or explain why they are not considered important.

Thank you for this helpful suggestion. We agree that areas with high stone contents could potentially be areas of high LGD. Thus we checked the measurement protocols again and noted that all locations, where VTPs were not measured because sediment was unsuitable, were characterized by thick muddy organic materials. Stones in the sediment only caused single measurement gaps within the VTPs, but estimation of LGD rates from VTPs at these locations were still possible. Thus, measurement locations characterized by high stone contents are included in the LGD distribution. Most of the gaps occurred in the lake section 150 m to 290 m south from the northern tip at western shore (Figure 2). At this section, the lake was difficult to access and VTP were difficult to measure as sediment was very muddy. Therefore, we reduced the spatial resolution within this section to 20 m. However, we assume that we still covered the spatial variability of LGD, because lake sediment in this lake section was very similar between the measurement locations. All other 11 locations, where VTP measurements were not possible, were irregularly distributed so that we do not expect strong influence of these measurement gaps on observed spatial patterns. We will include our assessment of the meaning of the measurement gaps on the observed spatial patterns within the discussion part and will correct the sentences on P10 L21 in the revised manuscript as follows:

P10 L18-21: "In total 520 VTPs were measured along the shoreline at Lake Hinnensee to analyse spatial patterns of near shore LGD, analyse the trend of LGD with increasing distance from shore and to analyse the temporal stability of LGD patterns. At the lake section 150 m to 290 m south from the northern tip of the western shore, spatial resolution of VTP measurements were reduced to 20 m as lake shore was difficult to access and sediment was unsuitable for measuring due to thick muddy organic material. At 21 11 other locations temperature measurements were not possible, as the locations were either inaccessible or sediment was unsuitable for measuring, for example due to high stone content or a thick muddy organic material."

Page 10 line 27 and onwards: the interquartile range is irrelevant as far as I can see. Further on in the section, you statistically analyse the LGD variation along lag distances. What is the point of this analysis, are the results interpreted in the discussion, and what is the physical meaning behind them? Again, use simpler metrics and less statistics where possible.

We used the IQR to describe the variability of measured LGD. As variability is a focus of our study, describing of the data dispersion is important for us and IQR is a common robust statistical measure of variability. As mention above, it was a conscious decision to use autocorrelation analyses as this method exactly fits the aim of describing the spatial variability along the shoreline. But to reduce the statistical methods we will leave out the median absolute differences calculated for different subsections of the lake (P 11, L3-6). And as mentioned above we will interpret the results of the autocorrelation analysis in the discussion and would clarify the introduction into the method of autocorrelation in the method part.

Page 12 line 11 – am I mistaken or does this indicate higher LGD rates further offshore than measured with other methods? If so, this would not fit with your overall appraisal.

We only used one method to quantify LGD rates: VTP measurements. We conducted VTP measurements along the DTS cable in the shallow near-shore area to verify temperature anomalies measured with the FO-DTS system and results showed a correlation of temperature anomalies and LGD estimations at each "corner" (P12 L11 – 13; Figure 6). Indeed, VTP measurements showed strong inflow at the most northern VTP measurement location taken along the cable at corner 2 (Figure 6). The hotspot in LGD is well in correspondence with the temperature anomaly observed at this location with the FO-DTS system. This consistent observation fits very well in our overall

appraisal as this location is still in the near shore area at the beginning of a steep step toward the lake centre. An interpretation of temperature anomalies observed at corner 2 and 3 is given on P17 L4-16. The comparison between LGD measurements along the DTS cable and temperature anomalies measured with the FO-DTS system will be discussed in a new subsection about a comparison of the applied methods within the discussion part. Bathymetry contour lines and distances from shoreline will be included in figure 6 in order to illustrate better the position of VTP measurement locations along the FO-DTS cable.

Along those lines, please provide a table that summarises the LDG rate measurements with different tools including error bars and discuss advantages and disadvantages of different methods.

As stated above, this is a misunderstanding as we only used one method to estimate LGD rates: VTPs. FO-DTS and radon results were only interpreted qualitatively. A subsection with a comparison of the applied methods (VTPs, FO-DTS, radon) will be included in the discussion part.

Page 13 line 5 – provide a grain size map, remove most of the text in this paragraph describing the grain size distribution and include only one or two sentences with the main information on grain size distribution that is required to understand the context.

We feel that the important point -the relationship of grainsize and LGD- is shown in figure 8. More detailed information of spatial patterns of grainsize would only distract from this main point. To further reduce the distraction from this main point we will remove P13 L 6 - 10.

Page 13 line 15: include the data in table 3 in figure 4.

If we interpret this comment in the right way, equations of regression models should be included in the corresponding figures (would be figure 8 and figure 9, not figure 4). This will be implemented.

The grain size model is nice, although I wonder a little about applicability given the large residual errors? In my view the most interesting message of this paper is shown in Figure 8: a variation of LGD of up to a factor 3 can be due to grain size alone.

As residual errors can indeed not be neglected, we don't recommend using the equation for LGD prediction, but, as the results showing a strong impact of grain size on LGD patterns, to use grain size observations to develop an effective and efficient experimental design. We will strengthen the outcome of the correlation in the text by including the information that LGD could vary up to a factor of three only by changes in grain size.

Page 13 line 32 and onwards – remove ordinary kriging from the ms (and remove this paragraph which is then obsolete) and use only the method that works best. If you can, justify the use of regression kriging. Similarly, remove the discussion section  page 15 lines 5-9 – it is a circular argument to say that regression kriging is the more appropriate method because results fit better.

As suggested, ordinary kriging will be removed from the manuscript.

Page 14 line 11 and onwards: applying individual linear regressions between far field predictors and LGD rates assumes the simplistic view that parameters can be isolated. Instead, use a linear mixed model to comprehensively analyse the combined effect of topographic far field predictors on LGD.

We entirely agree that applying individual linear regression models assume the simplistic view that parameters can be isolated. As mentioned above, we wanted both: to examine the relation between single far field predictors and LGD and therefore applied simple regression models and examined the relation of all far field predictors and LGD and also applied multiple regression models. But the multiple linear regression models revealed no significant relations (or improvements over the individual regressions) between LGD and the combined use of far field predictors (P 14 L19-21). As

we see no advantage by using linear mixed models instead of multiple linear regression models for our dataset and want to keep the statistical treatment as simple as possible and would therefore stick to the use of multiple regression models instead of linear mixed models.

Page 15 line 16-20 : can flow reversals be a result of measurement errors?

Following your suggestion, we checked the data of the VTP indicating a flow reversal and compared the measurements with a theoretical profile if LGD would be zero, At the VTPs measurement locations with -4 L m$^{-2}$ d$^{-1}$ it is possible that measurement errors caused negative rates, but at the locations with -8 and -11 L m$^{-2}$ d$^{-1}$ we trust our LGD estimations as at these locations differences between measured sediment temperatures and the modelled temperature profile assuming -8 and -11 L m$^{-2}$ d$^{-1}$ were on average 0.02°C, but significant larger between measured sediment temperatures and a theoretical profile if flow would be zero (0.23°C). As the quality of LGD estimation is much better assuming negative rates than zero exchange and temperature differences between a theoretical profile assuming zero exchange were larger than the measurement accuracy (0.03°C), we conclude that the reversal is not the result of a measurement error. To clarify the subsection about LGD reversals, we changed the subsection as follows:

P15 L17-20 "Even though the interpolated groundwater surface showed groundwater flow towards Lake Hinnensee from all directions (Figure 10), we measured negative LGD rates at one small subsection of the lake (Figure 2). Reasons for this flow reversal are unclear. However, this lake section is characterized by low groundwater gradients (Figure 10). While transpiration is likely to cause diurnal fluctuations in groundwater levels all around the lake, it can result in a local inversion of the groundwater – lake gradients at locations where these gradients are very low (Winter et al., 1998)."

Page 16 line 11: give % variations.

Will be included

Page 16 line 21: it is unfortunate that this important data is missing. This would have considerable strengthened the ms.

We agree, it is indeed unfortunate, but the number of sediment cores was already large with 30 samples, analysing more cores was simply beyond our capacity. .

Conclusions – only include FO-DTS results that provide new insights compared to those already published by your group.

As explained before, the FO-DTS results are all new.

Fig 9 – would be better presented in a plot modelled vs measured. #

We would prefer to leave figure 9 as it is. A plot with modelled vs measured data would not present the ability of the regression models to roughly represent the observed patterns. As our focus is on spatial patterns, we would like to keep this information in the plots.

Figures – include bathymetry contour labels in maps.
Will be included

---

## Author Response (AR1)

Response to comments by reviewer#1

We would like to thank the reviewer for the very constructive and helpful review which provided great support for the improvement of the paper. We have addressed all issues indicated in the review report. Please find detailed responses to the reviewers' suggestions and comments below.

Following the suggestions of reviewer 2 we have revised the manuscript substantially, concerning both structure and story line. We added a section on the use of the water table ratio as indicator if groundwater tables are expected to be topography- or recharge controlled. As the system at hand classifies as recharge controlled, the identified correlations between topographic indices and LGD patterns are somewhat surprising and indicate that even in recharge controlled lowland systems groundwater gradients are influenced by topography to some degree. We furthermore added a more in-depth evaluation of the applied methods, including a further simplification of the manual temperature measurements for a qualitative impression of LGD patterns. We end the revised manuscript with a suggested protocol and experimental design for future studies. To improve conciseness of the manuscript we removed the radon discussion from the main body of the text and summarized this information in the appendix.

The paper describes a case study of a very detailed investigation of groundwater discharge patterns to a lake. The authors used a range of methods and took a large number of measurements with the goal to derive relationships between parameters that are easy to obtain (e.g. topographic indices) and groundwater discharge patterns. They found that groundwater discharge was higher closer to the lake shore. They also found correlations between topography and large-scale groundwater inflow patterns, and between small-scale groundwater inflow patterns and sediment grain size distributions. The paper does not really present anything new. The methods are well established and the outcomes confirm the general assumptions made about groundwater discharge to lakes, including that it is usually very heterogeneous. The only surprise was that hydraulic conductivity did not seem to correlate with groundwater discharge. However, the investigation was well designed, the methods are nicely described and the paper is overall very well written. I think it therefore merits publication and it will be a very useful reference for other researchers working on similar subjects.

I only have a few minor comments that should be addressed before publication:

There is no information on the motivation for this study. Was it a purely science-driven study and the lake was selected for convenience reasons, or was there a problem that drove the initiation of the project, such as lake water quality issues which may result from groundwater discharge?

At Lake Hinnensee significant lake level fluctuation were observed in the last 30 years, but processes and mechanisms driving these fluctuations and the role of lacustrine groundwater discharge (LGD) within the water balance of the lake were only poorly understood.
We will include the background of the study within the introduction (in subsection 1.3 Objectives) as follows:
" To study these research questions we chose Lake Hinnensee, a typical post-glacial lake located in the intensively monitored TERENO observatory in the lowland landscape of northeast Germany. Strong water level declines observed in the last decades at this lake as well as at others in the region are currently under investigation. This lake system has the additional advantage that the upper unconfined aquifer in which the lake rests can be considered as largely homogeneous and isotropic (sandy sediments of a glacial outwash plain, no bedrock control). Therefore LGD patterns are unlikely to be dominated by geological discontinuities, and we were able to test the common assumptions that spatial patterns of LGD are controlled by sediment characteristics and topography as a proxy for gradients of the groundwater flow field."

How does the selection of the lake impact on the transferability of the findings?

We assume that our findings are transferable to similar lowland landscapes with quasi-homogeneous aquifers. As the water table ratio at this site indicated recharge control, we assume topography to be an even greater influence on LGD patterns in areas where groundwater tables are also topography controlled. However, more complex hydrogeological settings which include discontinuities can override and mask the topographic signal. This discussion is now included in the revised document

P9L26: Groundwater levels were measured "regularly": What does that mean? "Regular" can be once every year…please specify.

Thank you for this advice. Groundwater levels were measured approximately bimonthly. We will change the sentence as follows:
"Groundwater levels were measured every seven to nine weeks since 2012using an electric contact meter (SEBA Hydrometrie, electric contact meter type KLL, accuracy: ± 1 cm)."

P15L20: You give transpiration as a possible reason for a near-shore depression in groundwater levels. Is there a type or density of vegetation at this location that potentially transpires more than at other locations at the lake shore, i.e. is there a reason to believe transpiration could be the cause?

The type and density of vegetation does not significantly differ from other locations, but this lake section is characterized by low groundwater gradients. Transpiration could reduce groundwater gradients (Winter et al., 1998) and if groundwater gradients are already very low, transpiration could even cause a reversal of groundwater gradients. Thus, we assume the reversal results from the combination of transpiration and the occurrence of low groundwater gradients.
We will make this clearer by changing the paragraph as follows:
"Even though the interpolated groundwater surface showed groundwater flow towards Lake Hinnensee from all directions (Figure 10), we measured negative LGD rates at one small subsection of the lake (Figure 2). Reasons for this flow reversal are unclear. However, the neighbouring stretches of shoreline were characterized by very low LGD rates (Figure 2), even though $k_{sat}$ values at this section were comparably high (Figure 7) and thus we assume that very low hydraulic gradients are the cause for the low LGD rates. While transpiration is likely to cause diurnal fluctuations in groundwater levels all around the lake, it can result in a temporary local inversion of the groundwater – lake gradients at locations where these gradients are very low (Winter et al., 1998). This could be a potential explanation for the negative LGD rates measured at this location."

Conclusion: My main question is: so what? Your main recommendation seems to be to take topographic indices combined with a few sediment cores as a first step, and then do more investigations at areas of interest. But which ones are most useful and give you most value for money?

As mentioned above, we have now revised the manuscript substantially to clarify outcome and novelty as well as resulting recommendations. The value of using topographic indices or sediment cores to predict LGD patterns depends on the focus of the study. From our results, we assume that topographic indices can help to predict large scale LGD patterns and sampling sediment cores could help to get a picture of small scale variability in LGD patterns. We have revised the conclusions to clarify the main outcomes and also included a suggestion for an experimental protocol for future studies. Furthermore, we have evaluated the use of temperature measurements from one single sediment depth (instead of the entire profile) to get a fast impression of LGD patterns. Sediment temperatures from the top of the sediment down to a depth of 10 cm were not well correlated with LGD rates, but strong correlations were found between LGD rates and sediment temperatures measured 20 cm below surface and deeper (absolute correlation coefficients ranged between 0.46 and 0.96, see Figure below). Measuring sediment temperatures only in one sediment depth instead of complete VTPS would save a lot of time in the field. A discussion of these points is now included in the revised manuscript.

[Figure]

**Figure 1: Sediment temperature measured 30 cm below the sediment lake interface measured during different VTP surveys are plotted against LGD rates estimated from VTPs.**

Are the findings transferable to other lake settings?

As mentioned above, we assume that our findings are transferable to similar lowland landscapes. As the water table ratio at this site indicated recharge control, we assume topography to be an even greater influence on LGD patterns in areas where groundwater tables are also topography controlled. However, more complex hydrogeological settings which include discontinuities can override and mask the topographic signal. This discussion is now included in the revised document

I am also wondering how such an investigation would help remediation planning. How important is it to know the spatial distribution or local hot spots of groundwater discharge to a lake? What could you do about those hot spots, or would you use them to trace back a contamination source?

In general, knowing LGD variability, especially identifying LGD hotspots, is important from two perspectives: for water quantity issues and water quality issues. Missing hotspots of LGD could lead to a wrong estimation of LGD and hotspots could be of great importance, as these hotspots could carry larger loads of contaminations, especially if LGD hotspots meet contamination hotspots. You could thus indeed use this to trace back contamination sources. This will briefly be mentioned in the beginning of the conclusion as follows:
P21: "As LGD can significantly contribute to a lake water budgets and could furthermore significantly influence lake water quality by transporting large loads of nutrients or contaminants, quantifying LGD rates and determining LGD patterns can be essential for a sustainable lake management (Meinikmann et al., 2013; Lewandowski et al., 2015).

References: P10L11-13: the two references are missing in the reference list
Thank you for this advice. We will include these references in the reference list.

Language/typos:

Readability could be improved by using more hyphens, e.g.: Large-scale patterns, Small-scale variability, High-precision thermometer, Near-surface sediments, Far-field conditions, Climate-driven processes and similar constructions throughout the manuscript. "Grain size" should be two words

P1L27: groundwater-lake exchange

P7L22/23: "purged for 3.5 at least hours" should be "purged for at least 3.5 hours"

P8L4 and 6: I think internationally the term "screen" is preferred to "filter"

P8L27: mean was or means were

P9L1: goodness-of-fit

P10L4: "resolution of 1 m that" should be "so that"

P11L1: "norther" should be "northern"

P11L31: "measurements in2 minute intervals" should be "measurements in 2-minute intervals"

P1L28: "a slight negative, but spastically" should be "a slightly negative, but statistically"

P15L31/32: ksat should be ksat (subscript)

P16L5: remove comma after structure

Figures:

Figure 2: The use of (a) (b) and (c) in the caption is a bit confusing

Figure 7: kf should be ksat in the caption

Figure 10: (e) is missing in the caption

Thank you for catching these typos. We have made all the corrections.

We removed Fig. 10e) (LGD rates of lake subsections are plotted against mean groundwater gradients derived from ordinary kriging) as reviewer 2 suggest to reduce the amount of information in the figures.

Response to comments by reviewer #2

The authors would like to thank the reviewer #2 for the constructive and helpful review. It provided great support for the improvement of the manuscript. However, some of the comments seem to be the result of misunderstandings. We hope that our answers and changes in the manuscript clarify the misunderstandings. Please find detailed responses to the specific reviewers' suggestions and comments below. (The given page numbers refer to the document with tracked changes.)

General comments
This manuscript describes intensive field studies of LGD processes. An impressive amount of field data is presented, and an attempt is made to better comprehend spatial patterns of LGD on different spatial scales based on this large set of data. The major novelty of this contribution is the amount of data collected (as eluded to by the authors), but in my view overall few new insights into LGD processes are presented in the ms in its present form. In general, the ms is well-written, although it would benefit from a more concise and clearer structuring considering the large amount of data presented.

We have revised the manuscript substantially, concerning both structure and story line. We added a section on the use of the water table ratio as indicator if groundwater tables are expected to be topography- or recharge controlled. As the system at hand classifies as recharge controlled, the identified correlations between topographic indices and LGD patterns are somewhat surprising and indicate that even in recharge controlled lowland systems groundwater gradients are influenced by topography to some degree. We furthermore added a more in-depth evaluation of the applied methods, including a further simplification of the manual temperature measurements for a qualitative impression of LGD patterns. We end the revised manuscript with a suggested protocol and experimental design for future studies. To improve conciseness of the manuscript we removed the radon discussion from the main body of the text and summarized this information in the appendix.

I have a number of general comments that I believe should be addressed:

**1)** In parts, the ms reads like a data report, describing one experiment after another, with insufficient linkage between sections. It might help to re-structure and use the same subsection titles in methods and results.

Thank you for the constructive advice. The structure was adapted and now fits the short outline given at the end of the introduction and, as suggested, we parallelized subsection titles in methods and results.

| 2. Methods | 3. Results |
| --- | --- |
| 2.1 Study site | |
| 2.2 Estimating lacustrine groundwater discharge (LGD) | 3.1 Estimating lacustrine groundwater discharge (LGD) |
| 2.2.1 Near-shore LGD derived from vertical temperature profiles (VTPs) | 3.1.1 Near-shore LGD derived from vertical temperature profiles (VTPs)
    Spatial patterns along the shore line
    Spatial patterns perpendicular to the shore line
    Temporal stability of spatial pattern |
| |  |
| 2.2.2 Lake sediment temperature anomalies as indicators for offshore-LGD based on fibre optic distributed temperature sensing (FO-DTS) | 3.1.2 Lake sediment temperature anomalies as indicators for offshore-LGD based on fibre optic distributed temperature sensing (FO-DTS) |
|  | |

| 2.3 Identifying controls of LGD patterns | 3. 2  Identifying controls of LGD patterns |
|---|---|
| 2.3.1 Sediment heterogeneity as a small scale control
    Hydraulic conductivity from slug tests
    Grain size distributions from sediment cores | 3.2.1 Sediment heterogeneity as a small scale control
    Hydraulic conductivity from slug tests
    Grain size distributions from sediment cores
 |
|  |  |
| 2.3.2 Topographic indices as controls on large scale LGD patterns | 3.2.2 Topographic indices as controls on large scale LGD patterns |
| 2.3.3 Groundwater flow field as control on large scale LGD patterns | 3.2.3 Groundwater flow field as control on large scale LGD patterns |
|  | 3.2.4 Linear regression models between LGD and far-field predictors |

Similarly, the figures, in particular the presented maps, are very busy. Consider breaking up the figures into smaller sub-figures rather rhan overlying too many things on one graph. I encourage the authors to carefully consider which level of information and especially which data are really required to support the main findings and eliminate those that aren't. For example, data or analyses that have not yielded conclusive results could be removed, but mentioned in one sentence that this was tried, moved to an appendix etc.

Thank you for this general suggestion. Concerning the suggestion of reducing data and data analysis, detailed responses are given on specific suggestions and comments below. Concerning the maps and figures we followed your suggestion and checked each figure and map concerning the level of information. We agree with the reviewer that some maps are busy and therefore removed the radon information. To increase the clarity in the figures, we furthermore modified the figures as follows:

- Figure 1b+c: bathymetry contour labels will be included, radon removed from all subfigures
- Figure 2 b-c: different background colours will be used for the southern and northern part and x labels will be inserted in figure 2b.
- Figure 3: y label will be changed from "|ρ|" to "autocorrelation coefficient (|ρ|)", two correlation plots of correlations between neighbouring LGD measurement locations (distance 10m) in the northern part and southern part
- Figure 5: NEW: Sediment temperatures at 30 cm below the sediment lake interface measured during two different VTP surveys are plotted against LGD rates estimated from VTPs.
- Current Figure 6b: Radon removed
- Previous Figure 6 removed
- Figure 7: labels are removed and presentation of the distances of measurement locations will be improved by grouping the measurement locations into two groups: first group: 0 m -190 m south from the northern tip of the lake, second group: 480 m – 520 m south from the northern tip of the lake. For each group a different colour ramp will be used. A corresponding legend will be added to the figure.
- Figure 8 and Figure 9: Equations of regression models are included Figure 10: legend of subfigure (a) will be placed right of the figure, subfigures (b-d) will be placed below figure (a). Subfigure (e) will be removed

**2)** I suggest to remove the radon part from the study. Data is very sparse, and does not contribute significantly to the final interpretation. It is argued that low Radon concentrations in the lake's center

are a result of low LGD, but they could equally well be a result of (a) not taking into account greater depth (ie inventories were not calculated) (b) wind-driven radon loss to the atmosphere. In short, there is not sufficient information to adequately interpret the radon data.

We moved methods, results and discussion concerning the radon data set to the appendix and extended the discussion there to address your concerns.

We used the radon measurements in order to localise groundwater inflow in the offshore part of the lake. Our approach was orientated on the approach of Shawn et al., 2013 or Ono et al., 2012 measuring radon activities at the lake bottom at different locations and relating significant differences in radon activities to spatial patterns in groundwater inflow. Shawn et al., 2013 took samples when lake was frozen and could therefore neglect the effect of wind losses. However, by taking samples from the lake bottom, we assume that the effects of wind-driven losses are low. Kluge et al. measured several vertical profiles in a lake and related increasing radon activities at the bottom of the lake to local groundwater inflow. Furthermore, our test of the ability of radon as an indicator for groundwater inflow at Lake Hinnensee showed significant stronger radon activities at locations with groundwater inflow (P12 L16-17), even though wind losses were not considered and lake depths were shallow (< 1 m) at these locations. Thus we assume, that measurements of radon activities from the lake bottom at Lake Hinnensee could be used to identify locations with significant groundwater inflow. To consider the comment, we would change the discussion as follows:

We are not sure what is meant by "not taking into account greater depth". Indeed, we did not calculate inventories, as we were only interested on identifying presence or absence of groundwater inflow in the central part of the lake, not in calculating groundwater inflow rates. For the purpose of calculating groundwater inflow rates, indeed, sufficient information is not available, but this was as well beyond the scope of these measurements, which were only used as simple qualitative indicators. We furthermore agree that sampling locations were sparse as the number of samples was limited, but the sampling design covered the entire lake, including the deep parts in the south. As the radon data provide us with information from this southern area, where no other data are available, we would prefer to leave the data in the manuscript, but agree to move it to the appendix.

**3)** It appears that the fibre-optics part of this study has previously been published (Blume et al 2013), or that the data presented here does not add new insights to those found previously. Please explain in detail what is new and / or consider reducing the part on FO-DTS in this ms.

The FO-DTS data were not presented before. Blume et al. (2013) conducted a FO-DTS study at Lake Hinnensee three years before our FO-DTS campaign, but the study focused only on a small shore section of 20 m length and 4 m width and the objectives were different, in case of Blume et al. (2013) methodical-driven ("The objective of the present study is to test whether FO-DTS-based upscaling of point measurements of lacustrine groundwater discharge rates is an adequate and feasible approach to represent the spatial heterogeneity of LGD rates." (Blume et al., 2013)). The experimental designs differ strongly from each other. While Blume et al. (2013) arranged the DTS cable in the near shore part on a small section of 20 m length and 4 m width, we installed the cable along 6 transects across the lake. Beside the methodical insight from the study of Blume et al. (2013), they mentioned spatial variability in LGD and a decrease of LGD with increasing distance from shore within the focus area. As we found as well spatial variability in LGD and a decrease of LGD with increasing distance from shore, we compared our results with the previous study within the discussion part. To clarify the different approaches and foci of the two studies, we change the following sentence:

"The study from Blume et al. (2013), conducted at a small shore line section of 20 m length and 4 m width in the northern part of Lake Hinnensee, indicated that the strongest decrease of LGD occurred in the first 1.5 m distance from shore."

**4)** It is in times difficult to follow the heavy reliance on statistical treatment. For example, what is the point of the autocorrelation analysis of LGD values (e.g. page 5)? Is there a physical process associated with LGD that requires an understanding of autocorrelation of LGD along a shoreline (there is no use made of this analysis in the discussion unless I overlooked it?).

Thank you for the advice. We have reduced the statistical methods to increase ease of reading. The following changes were made:

- removing MAD as an additional measure of spatial variability (P7&P13),
- replacing RMSE by median differences (P7&P13) and
- removing the description of grain size distribution (P 15).

Concerning the comment on autocorrelation: this method has the advantage that it clearly addresses the question, how similar neighboured measurement locations are and up to which distance a correlation between measurement locations exist. This is exactly what we wanted to show. To reduce statistical methods, we could alternatively present correlation plots and correlation coefficients of neighboured LGD measurement locations (Lag distance 10 m, spatial resolution of VTP measurements), as presented in the figures below. We would prefer to leave the autocorrelation analysis in the manuscript, but leave the decision to the editor. However, we will include the result of correlation between LGD measurement locations within the discussion and have clarified the introduction into the method of autocorrelation in the method part as follows:

P7 L1: Spatial variability and correlation of LGD along different distances along the shoreline were analysed using autocorrelation plots and autocorrelation values ($|\rho|$) as described in Caruso et al. (2016). High autocorrelation ($|\rho|$) indicate that LGDs along a given stretch of the shoreline are correlated, whereas $|\rho| < 0.2$, indicate that LGDs are uncorrelated and strong spatial variability exists.

[Figure]

Figure 2: Correlation between neighbouring LGD measurement locations in the northern part (left) and southern part (right).

**5)** A linear mixed effects model should be applied instead of independent linear relationships with explanatory far field predictors.

We applied simple regression models to examine the relation between single far field predictors and LGD and applied multiple regression models to examine the combined relation of all far field predictors with LGD. But the multiple linear regression models revealed no significant relationship (or improvements over the individual regressions) between LGD and the combined use of far field predictors (P 14 L19-21). As we see no advantage by using linear mixed models instead of multiple linear regression models for our dataset and want to keep the statistical treatment as simple as possible, we would therefore prefer to use the multiple regression models instead of linear mixed models.

**6)** Data distribution is very irregular, in particular the largest coverage is in the northern section of the lake where you already know that the highest LGD is found from a previous paper. In addition, the lake is connected to a larger lake to the south (of which you make no mention). Could this affect LGD patterns, in particular with respect to far field predictors?

We do not expect an effect on LGD patterns at Lake Hinnensee as a result of the open connection to Lake Fürstenseer See in the south. From topography and the regional flow field, we expect in general smaller groundwater inflow rates at Lake Fürstenseer See and negative LGD rates in the south of Lake Fürstenseer See. One process might influence temporally LGD patterns at Lake Hinnensee: if wind direction is from south and wind is strong, wind- driven water movements might increase water table towards the north of Lake Hinnensee, which might reduce groundwater inflow rates in the northern part.

We added a sentence explaining that the connection to Lake Fürstenseer See is unlikely to affect LGD patterns in Lake Hinnensee.

P 5: "The connection to Lake Fürstenseer See is not assumed to influence LGD patterns of Lake Hinnensee, as the general flow direction of the groundwater flow field is from north to south with water leaving the lake system at the southern end of Lake Fürstenseer See."

We furthermore did not know the highest rates would be in the northern section. We just knew that high rates occur here. The previous study was limited to a very small section of the shore line (20m).

**7)** LGD flux was estimated / calculated / modelled using different methods, and it is somewhat difficult to reconcile the results obtained by different methods in quantitative terms. How do the flux estimates compare, and are they consistent with each other?

LGD rates were only estimated from VTPs. Subsequently regression models were used to model these LGD rates based on various predictors. FO-DTS and radon measurements were only used as qualitative indicators for offshore LGD. A comparison between measured and modelled LGD rates is given on P13 L17 -20 and P14 L23-26. A subsection with a comparison of the applied methods (VTPs, FO-DTS, radon) could be given in the discussion part. In order to make clearer that FO-DTS and radon measurements were only used as qualitative tracers, we insert the following sentence in the objectives:

P 4 "To identify LGD patterns, we measured VTPs in the near shore area and used FO-DTS measurements and radon sampling in the off-shore area. VTPs were used to quantify LGD rates, whereas FO-DTS measurements and radon sampling were used as qualitative tracers to detect the presence or absence of off-shore LGD. As potential controls of LGD patterns…"

**8)** It was found that LGD correlated with grain size but not with slug test results (hydraulic conductivity). This is surprising (in particular in a large data set), and would require more detailed discussion. Were the slug tests done correctly, correct equations applied (there are a few for different experimental configurations), is there a possibility that there is a problem with the slug test results?

We checked the equations again, but found no problems, we also checked the correlation applying the equation of Hvorslev (1951) without the shape correction factor introduced by Chapuis (1989) and a second adaption of the Hvorslev equation even though both equation fit less well to our experimental setup. However, with both equations absolute values of estimated ksat changed slightly, but calculated correlation coefficients between LGD and $k_{sat}$ values were the same ($\rho$=-0.36). Thus conclusion from measurements would be the same even by applying different equations. Slug tests were carried out carefully in the field. However, as we discussed on P 15 L31 – P16 L5, we assume that the uncertainty and errors of the slug test method are larger than the differences in hydraulic conductivity in the sandy sediment at Lake Hinnensee.

In the end I am a little lost what key message to take away from the ms. I believe this is likely due to 'too much data, too much analyses'. In this light, I suggest to include a detailed account of the value of each of the applied methods (incl a comparison and appraisal of the different tools to estimate flux on different scales), and to provide a 'recipe' how to design a 'good' study on LGD variability in the future.

Thank you for these constructive suggestions, we will include a subsection on evaluating the applied methods in the discussion part (P 23) and discuss the problems of using piezometers to identify LGD rates and patterns due to uncertainties in $k_{sat}$ estimations. Furthermore we will give a recipe how to design a good study. In this part we would also emphasize the advantages of using a needle for measuring sediment temperatures, because this instrument allows precise sediment temperature measurements without disturbing the sediment and the flow. Furthermore, we will evaluate the use of temperature measurements from one single sediment depth (instead of the entire profile) to get a fast impression of LGD patterns and will therefore present correlations between LGD rates estimated from VTPs and sediment temperatures from different depths of the of the profile. Sediment temperatures from the top of the sediment down to a depth of 10 cm were not well correlated with LGD rates, but strong correlations were found between LGD rates and sediment temperatures measured 20 cm below surface and deeper (correlation coefficients range between 0.46 and 0.96 and see figure below). Measuring sediment temperatures only in one sediment depth instead of complete VTPS would save a lot of time in the field. A discussion of these points will be included in the revised manuscript.

[Figure]

Figure 3: Sediment temperature measured 30 cm below the sediment lake interface measured during different VTP surveys are plotted against LGD rates estimated from VTPs.

In summary, whilst some aspects of the study provide new general insights into LGD variability, overall this ms appears to be primarily of regional interest and as such I wonder if it would not be better placed in a journal with a more regional focus e.g. Journal of Hydrology – Regional Studies.

As our study bridges the observational gap between detailed local and low resolution regional investigations of previous LGD studies and highlighted correlation between LGD patterns and external controls, which are transferable to other landscapes with similar settings and provides methodological recommendations, we feel that our study does provide new insights which are beyond the regional interest. The recommendations and transferability and novelty of our results are more strongly highlighted in the revised manuscript (see also our general reponse at the very beginning).

Specific comments

Abstract – include relevant quantitative data on results (order of LGD fluxes etc). Also, include size of lake in abstract and study site description.

We have revised the abstract completely, and also included the points suggested here.

Page 2 line 15-20 and relevant sections after: previous study results are not contradictory – they just highlight the fact that the effect of sediment structure on discharge is highly site-specific. Are your findings of a more general nature and as such applicable to other sites?

Thank you for your suggestion, we agree with the reviewer, the effect of sediment characteristics on LGD patterns is site-specific. Thus we will rephrase the section as follows:

"P2 L26: There is no clear picture of the role of sediment characteristics in controlling LGD patterns and observations seem to be very site specific. For example, Kidmose et al. (2013) found that low permeable lacustrine sediments can completely prevent groundwater upwelling, whereas Vainu et al. (2015) observed LGD through low permeable lacustrine sediments. Kishel and Gerla (2002) associated small scale variabilities in LGD with small scale heterogeneities in hydraulic conductivities (Kishel & Gerla, 2002), but in contrast, and Schneider et al., (2005) found no correlation between seepage rates and sediment characteristics."

We cannot say how well our results can be transferred to other sites as this hasn't been tested yet, however, we generally assume similar relationships at other sites in this type of geology and landuse. This is now also discussed in the revised version of our manuscript.

Page 3 methods: include site coordinates and a larger overview map.

Site coordinates are now included, a general description of the location of the study site is given in the subsection 2.1 "Study site". To reduce the number of figures we did not include a map of Germany.

Page 5 line 23: show example data with and without SGD in a graph.

We are sorry, but we are not sure what is meant with this suggestion. The sentence on Page 5 in line 23 does not seem to fit to this suggestion and what is SGD?

Page 6: line2: why compare 2 datasets with RMS / statistics? Why not just subtract one from another and analyse the difference if you have data from the same locations? Use statistical methods only where they are required.

Thank you for the comment, RMSE will be replaced by the median of the differences.

P7: "In order to analyse the temporal stability of spatial patterns we analysed the differences between the LGD rates measured in different surveys and calculated the correlation between the surveys using the Spearman's rank correlation coefficient ($\rho$). Correlations were regarded as significant for p-values smaller than 0.05. Differences between LGD rates measured in different years were quantified with the RMSE. "

P13: "The differences between LGD rates measured in different years were lowest comparing rates from summer 2011 and summer 2012 (median difference = -6 L m-2 d-1) and strongest comparing rates from summer 2011 and winter 2013 (median difference = 27 L m-2 d-1)."

Page 5 line 14: provide temperature values of calibration points.

This information is now included

Page 6 line 29: include month (August).

Is now included

Page 10 line 21: VTP could not be taken in areas with high stone content – however, these are areas with potentially high bulk hydraulic conductivity and could present a preferential flow path. Please acknowledge that you might have missed important entry points or explain why they are not considered important.

Thank you for this helpful suggestion. We agree that areas with high stone contents could potentially be areas of high LGD. Thus we checked the measurement protocols again and noted that all locations, where VTPs were not measured because sediment was unsuitable, were characterized by thick muddy organic materials. Stones in the sediment only caused single depth measurement gaps within the VTPs, but estimation of LGD rates from VTPs at these locations were still possible. Thus, measurement locations characterized by high stone contents are included in the LGD distribution. Most of the gaps occurred in the lake section 150 m to 290 m south from the northern tip at western shore (Figure 2). At this section, the lake was difficult to access and VTP were difficult to measure as sediment was very muddy. Therefore, we reduced the spatial resolution within this section to 20 m. However, we assume that we still covered the spatial variability of LGD, because lake sediment in this lake section was very similar between the measurement locations. All other 11 locations, where VTP measurements were not possible, were irregularly distributed so that we do not expect strong influence of these measurement gaps on observed spatial patterns. We will  correct the sentences on P10 L21 in the revised manuscript as follows:

P12: "At 216 locations along the shoreline of Lake Hinnensee (Figure 1) a total of 520 VTPs were measured to analyse a) spatial patterns of near shore LGD, b) the trend of LGD with increasing distance from shore and c) the temporal stability of LGD patterns. These 520 profiles thus include repeated measurements in time as well as measurements at two distances to shore. At the western lake section, 150 m to 290 m from the northern tip, VTP measurements could only be taken every 20 m instead of every 10 m as the lake shore could either not be accessed or the sediment was unsuitable for measuring due to a thick layer of muddy organic material. However, as lake sediments in this lake section were quite homogeneous, we assume that despite the wider spacing we still captured the spatial variability of LGD. The same reasons also precluded measurements at 11 other locations around the lake. These other 11 locations were irregularly distributed so that gaps were small and we do not expect a strong influence of these gaps on overall spatial patterns. 22 profiles (4%) were excluded from the analyses as no satisfying fit to the heat transport equation could be achieved. The quality of all remaining LGD estimations was satisfying (median(RMSE) = 0.06 °C, n = 498)."

Page 10 line 27 and onwards: the interquartile range is irrelevant as far as I can see. Further on in the section, you statistically analyse the LGD variation along lag distances. What is the point of this analysis, are the results interpreted in the discussion, and what is the physical meaning behind them? Again, use simpler metrics and less statistics where possible.

We used the IQR to describe the variability of measured LGD. As variability is a focus of our study, describing of the data dispersion is important for us and IQR is a common robust statistical measure of variability. As mention above, it was a conscious decision to use autocorrelation analyses as this method exactly fits the aim of describing the spatial variability along the shoreline. But to reduce the statistical methods we will leave out the median absolute differences calculated for different subsections of the lake . And as mentioned above we will interpret the results of the autocorrelation analysis in the discussion and would clarify the introduction into the method of autocorrelation in the method part. We also added two scatter plots to clarify the use of the autocorrelation plot as well as the difference between the north and the south.

Page 12 line 11 – am I mistaken or does this indicate higher LGD rates further offshore than measured with other methods? If so, this would not fit with your overall appraisal.

We only used one method to quantify LGD rates: VTP measurements. We conducted VTP measurements along the DTS cable in the shallow near-shore area to verify temperature anomalies measured with the FO-DTS system and results showed a correlation of temperature anomalies and LGD estimations at each "corner". Indeed, VTP measurements showed strong inflow at the most northern VTP measurement location taken along the cable at corner 2 (Figure 6). The hotspot in LGD is well in correspondence with the temperature anomaly observed at this location with the FO-DTS system. This consistent observation fits very well in our overall appraisal as this location is still in the near shore area at the beginning of a steep step toward the lake centre. To avoid misunderstandings and reduce the amount of data, Fig. 6 was deleted.

Along those lines, please provide a table that summarises the LDG rate measurements with different tools including error bars and discuss advantages and disadvantages of different methods.

As stated above, this is a misunderstanding as we only used one method to estimate LGD rates: VTPs. FO-DTS and radon results were only interpreted qualitatively. A subsection with a comparison of the applied methods (VTPs, FO-DTS) will be included in the discussion part.

Page 13 line 5 – provide a grain size map, remove most of the text in this paragraph describing the grain size distribution and include only one or two sentences with the main information on grain size distribution that is required to understand the context.

We feel that the important point -the relationship of grainsize and LGD- is shown in figure 8. More detailed information of spatial patterns of grainsize would only distract from this main point. To further reduce the distraction from this main point we will remove –the lengthy description on P15.

Page 13 line 15: include the data in table 3 in figure 4.

If we interpret this comment in the right way, equations of regression models should be included in the corresponding figures (would be figure 8 and figure 9, not figure 4). This will be implemented.

The grain size model is nice, although I wonder a little about applicability given the large residual errors? In my view the most interesting message of this paper is shown in Figure 8: a variation of LGD of up to a factor 3 can be due to grain size alone.

As residual errors can indeed not be neglected, we don't recommend using the equation for LGD prediction, but, as the results showing a strong impact of grain size on LGD patterns, to use grain size observations to develop an effective and efficient experimental design. We will strengthen the outcome of the correlation in the text by including the information that LGD could vary up to a factor of three only by changes in grain size.

Page 13 line 32 and onwards – remove ordinary kriging from the ms (and remove this paragraph which is then obsolete) and use only the method that works best. If you can, justify the use of regression kriging. Similarly, remove the discussion section  page 15 lines 5-9 – it is a circular argument to say that regression kriging is the more appropriate method because results fit better.

In the context of the question whether or not the system is topography controlled, the comparison of ordinary kriging and regression kriging provides useful information and we would prefer not to remove this point from the manuscript entirely. However, we did remove the subfigure concerning ordinary kriging from Figure 10 to reduce the density of information as suggested above.

Page 14 line 11 and onwards: applying individual linear regressions between far field predictors and LGD rates assumes the simplistic view that parameters can be isolated. Instead, use a linear mixed model to comprehensively analyse the combined effect of topographic far field predictors on LGD.

We entirely agree that applying individual linear regression models assume the simplistic view that parameters can be isolated. As mentioned above, we wanted both: to examine the relation between single far field predictors and LGD and therefore applied simple regression models and examined the relation of all far field predictors and LGD and also applied multiple regression models. But the multiple linear regression models revealed no significant relations (or improvements over the individual regressions) between LGD and the combined use of far field predictors. As we see no advantage by using linear mixed models instead of multiple linear regression models for our dataset and want to keep the statistical treatment as simple as possible and would therefore stick to the use of multiple regression models instead of linear mixed models.

Page 15 line 16-20 : can flow reversals be a result of measurement errors?

Following your suggestion, we checked the data of the VTP indicating a flow reversal and compared the measurements with a theoretical profile if LGD would be zero, At the VTPs measurement locations with -4 L $m^{-2}$ $d^{-1}$ it is possible that measurement errors caused negative rates, but at the locations with -8 and -11 L $m^{-2}$ $d^{-1}$ we trust our LGD estimations as at these locations differences between measured sediment temperatures and the modelled temperature profile assuming -8 and -11 L $m^{-2}$ $d^{-1}$ were on average 0.02°C, but significant larger between measured sediment temperatures and a theoretical profile if flow would be zero (0.23°C). As the quality of LGD estimation is much better assuming negative rates than zero exchange and temperature differences between a theoretical profile assuming zero exchange were larger than the measurement accuracy (0.03°C), we conclude that the reversal is not the result of a measurement error. To clarify the subsection about LGD reversals, we changed the subsection as follows:

Even though the interpolated groundwater surface showed groundwater flow towards Lake Hinnensee from all directions (Figure 10), we measured negative LGD rates at one small subsection of the lake (Figure 2). Reasons for this flow reversal are unclear. However, the neighbouring stretches of shoreline were characterized by very low LGD rates (Figure 2), even though ksat values at this section were comparably high (Figure 7) and thus we assume that very low hydraulic gradients are the cause for the low LGD rates. While transpiration is likely to cause diurnal fluctuations in groundwater levels all around the lake, it can result in a temporary local inversion of the groundwater – lake gradients at locations where these gradients are very low (Winter et al., 1998). This could be a potential explanation for the negative LGD rates measured at this location.

Page 16 line 11: give % variations.

Are now included

Page 16 line 21: it is unfortunate that this important data is missing. This would have considerable strengthened the ms.

We agree, it is indeed unfortunate, but the number of sediment cores was already large with 30 samples, analysing more cores was simply beyond our capacity. .

Conclusions – only include FO-DTS results that provide new insights compared to those already published by your group.

As explained before, the FO-DTS results are all new.

Fig 9 – would be better presented in a plot modelled vs measured. #

We would prefer to leave figure 9 as it is. A plot with modelled vs measured data would not present the ability of the regression models to roughly represent the observed patterns. As our focus is on spatial patterns, we would like to keep this information in the plots.

Figures – include bathymetry contour labels in maps.
Are now included

---

## Author Response (AR2)

Dear Editor,

we are glad to see that you and the referees were happy with our revisions. We agree with all suggestions by Referee #3 and have sharpened the introduction accordingly.

We have uploaded two versions of the manuscript, the second with tracked changes so that you can easily identify the last modifications.

Best regards,

Christina Tecklenburg

Response to comments by Referee #3

We would like to thank the Referee #3 for the very constructive comments to further focus the paper. Please find detailed responses to the reviewers' suggestions below.

I commend the authors on a very detailed analysis of local and non-local groundwater discharge in a lacustrine setting. For my review I paid particular attention to the author's responses and edits to the initial set of reviews. Overall, I believe the manuscript is novel based upon the methods applied and the amount of data that was collected. I do believe the authors provide a context that is important for future research and applications that seek to understand the influence of local topography versus groundwater recharge controlled systems. They also provide evidence for the role of sediment permeability and its ability to mediate topographic drivers. The authors have revised the manuscript substantially in terms of structure and the story line. While the manuscript does not provide major new insights, it provides a great analysis and data set that tests/confirms what others have shown – I believe this is important and a solid iteration for hydrologic sciences.
I have two specific comments:
1. The study has too many general/vague research questions. I'd suggest being more specific to help the reader. For example
a. combine question 1 and 2 to "How does LGD vary in space and time?"
b. combine question 2 and 3 to "What are the relative roles of regional groundwater discharge, local topography and sediment permeability for spatial patterns of LGD?
c. Delete "Can we predict LGD patterns"

Thank you for this constructive advice. We changed the research questions as suggested.

2. The introduction section 1.2 is dense and a lot of this could be saved for the methods section. I'd suggest condensing this and focusing on the major point that is highlighted starting on line 18. Previous studies have focused either on large or small scale LGD patterns. You use multiple techniques and lines of inference to evaluate both local and non local controls on LGD. This is the new contribution.

Thank you for this comment. We have sharpened the introduction accordingly.